# Coreset for rational functions

## Abstract

We consider the problem of fitting a rational function $g : \mathbb{R} \to \mathbb{R}$ to a time-series $f : \{1, \cdots, n\} \to \mathbb{R}$. This is by minimizing the sum of distances (loss function) $\ell(g) := \sum_{i=1}^{n} |f(i) - g(i)|$, possibly with additional constraints and regularization terms that may depend on $g$. Our main motivation is to approximate such a time-series by a recursive sequence model $G_n = \sum_{i=1}^{k} \theta_i G_{n-i}$, e.g. a Fibonacci sequence, where $\theta \in \mathbb{R}^k$ are the model parameters, and $k \geq 1$ is constant. For $\varepsilon \in (0, 1)$, an $\varepsilon$-coreset for this problem is a data structure that approximates $\ell(g)$ up to $1 \pm \varepsilon$ multiplicative factor, for every rational function $g$ of constant degree. We suggest a coreset construction that runs in $O(n^{1+o(1)})$ time and returns such a coreset that uses $O(n^{o(1)}/\varepsilon^2)$ memory words. We provide open source code as well as extensive experimental results, on both real and synthetic datasets, which compare our method to existing solvers from Numpy and Scipy.

## 1 Background

The original motivation for this work was to suggest provable and efficient approximation algorithms for fitting input data by a stochastic model or its variants, such as Hidden-Markov Models (HMM) Basu et al. (2001); McCallum et al. (2000); Murphy (2002); Park et al. (2012); Sassi et al. (2020); Yu et al. (2010) and reference therein, Baysian Networks Acar et al. (2007); Murphy (2002); Nikolova et al. (2010); Rudzicz (2010) and reference therein, auto-regression Ghosh et al. (2013), and Decision Markov Process Shanahan & den Poel (2010). Informally, and in the context of this work, a model defines a *time series* (sequence, discrete signal, time series) $F : [n] \to \mathbb{R}$ where $[n] := \{1, \cdots, n\}$, and the value $F(t)$ at time (integer) $t \geq 1$ is a function of only the previous (constant) $k \geq 1$ past values $F(t-1), \cdots, F(t-k)$ in the sequence, and the model's parameter $\theta$.

### 1.1 Auto-regression

Unfortunately, most existing results seem to be based on heuristics with little provable approximation guarantees. We thus investigate a simplified but fundamental version, called *auto-regression*, which has a provable but not so efficient solution using polynomial system solvers, after applying the technique of *generating functions*. This technique is strongly related to the Fourier, Laplace and $z$-Transform, as explained below. We define an auto-regression inspired by Ghosh et al. (2013); Eshragh et al. (2019); Yuan (2009), as follows:

**Definition 1.** *A time-series $F : [n] \to \mathbb{R}$ is an auto-regression (AR for short) of degree $k$, if there exist a vector of coefficients $\theta = (\theta_1, \cdots, \theta_k) \in \mathbb{R}^k$ such that $F(t) = \theta_1 F(t-1) + \cdots + \theta_k F(t-k)$. The polynomial $P(x) = x^k - \theta_k x^{k-1} - \cdots - \theta_1$ is called the characteristic polynomial of $F$.*

Substituting $k = 2$, $\theta = (1, 1)$ and $F(1) = F(2) = 1$ in Definition 1 yields the *Fibonacci sequence*, i.e. $F(t) = F(t-1) + F(t-2)$, where $F(1) = F(2) = 1$.

**From Auto-regression to Rational functions.** In the corresponding "data science version" of Fibonacci's sequence, the input is the time-series $G(1), G(2), \cdots$, which is based on $F$ with additional noise. A straight forward method to recover the original model is by directly minimizing of the squared error between the given noisy time-series and the fitted values, as done e.g. in Eshragh et al. (2019)) using simple linear regression. However, this has a major drawback; AR time-series usually grows exponentially, like geometric sequences, and thus the loss will be dominated by the last few terms in the time-series. Moreover, small changes in the time domain have exponential

affect over time, it makes more sense to assume added noise in the frequency or generative function domain. Intuitively, noise in analog signals such as audio/video signals from an analog radio/tv is added in the frequency domain, such as aliasing of channels Nyquist (1928); Karras et al. (2021); Shani & Brafman (2004), and not in the time domain, such as the volume.

To solve these issues, the fitting is done for the corresponding generation functions as follows.

**Proposition 1** (generative function Yuan (2009)). *Consider an AR time-series $F$ and its characteristic polynomial $P(x)$ of degree $k$. Let $Q(x) = x^k P(\frac{1}{x})$ be the polynomial whose coefficients are the coefficients of $P$ in reverse order. Then, there is a polynomial $R(x)$ of degree less than $k$ such that the* generative function *of $F$ is $f(x) := \sum_{i=1}^{\infty} F(i)x^{i-1} = \frac{R(x)}{Q(x)}$, for every $x \in \mathbb{R}$.*

Inspired by the motivation above, we define the following loss function for the AR recovery problem

**Problem 1** (RFF). *Given a time-series $g : [n] \to \mathbb{R}$ and an integer $k \geq 1$, find a rational function $f : [n] \to \mathbb{R}$ whose numerator and denominator are polynomials of degree at most $k$ that minimizes $\sum_{x=1}^{n} |f(x) - g(x)|$.*

Note that the loss above is for fitting samples from the generative function of a noisy AR as done in Section 3.1. While we will focus on sum of errors (distances), we expect easy generalization to squared-distances, robust M-estimators and any other loss function that satisfies the triangle inequality, up to a constant factor, as in other coreset constructions Feldman (2020).

## 1.2 CORESETS

Informally, an input signal $P$ consists of 2-dimensional points, a set $Q$ of models, an approximation error $\epsilon \in (0, 1)$, and a loss function $\ell$, a coreset $C$ is a data structure that approximates the loss $\ell(P, q)$ for every model $q \in Q$, up to a multiplicative factor of $1 \pm \epsilon$, in time that depends only on $|C|$. Hence, ideally, $C$ is also much smaller than the original input $P$.

**Coreset for rational functions.** Unfortunately, similarly to Rosman et al. (2014), the RFF problem with general input has no coreset which is weighed subset of the input; see Claim 1. This was also the case e.g., in Jubran et al. (2021). Hence, we solve this problem similarly to Rosman et al. (2014), which requires us to assume that our input signal's first coordinate is simply a set of $n$ consecutive integers, rather than a general set of reals. Even under this assumption there is no coreset which is a weighed subset of the input, or even a small weighed set of points; see Claim 1. We solve this problem by constructing a "representational" coreset that allows efficient storage and evaluation but dues not immediately yield an efficient solution to the problem as more commonly Feldman (2020). For more explanation on the components of this coreset see Section 1.4.

**Why such coreset?** A trivial use of such coreset is data compression for efficient transmission and storage. While there are many properties of coresets as mentioned at Feldman (2020), some of them are non-immediate from our coreset; see Feldman (2020) for a general overview that was skipped due to space limitations. Nonetheless, since optimization over the coreset reduces the number of parameter, we hope that in the future there would be an efficient guaranteed solution (or approximation) over the coreset. Moreover, since this coreset does support efficient evaluation, we hope this coreset would yield an improvement for heuristics by utilizing this fast evaluation.

## 1.3 RELATED WORK

**Polynomial approximations.** While polynomials are usually simple and easy to handle, they do not suffice to accurately approximate non-smooth or non-Lipschitz functions; in such cases, high order polynomials are required, which leads to severe oscillations and numerical instabilities Peiris et al. (2021). To overcome this problem, one might try to utilize piecewise polynomials or polynomial splines Northrop (2016); Nürnberger (1989); Sukhorukova (2010). However, this results in a very complex optimization problem Meinardus et al. (1989); Sukhorukova & Ugon (2017).

**Rational function approximation.** A more promising direction would be to utilize rational functions for approximating the input signals, an example for this is Runge's phenomenon Epperson (1987) that resonates with Figure 4 in the appendix. Rational function approximation is a straight forward extension for polynomial approximations Trefethen (2019), yet are much more expressive

due to polynomial in the denominator. A motivation for this can be found in the popular book Bulirsch et al. (2002). A close relation between such functions and spline approximations has been demonstrated e.g., in Petrushev & Popov (2011). Given an input signal consisting of $n$ pairs of points in $\mathbb{R}^2$, the rational function fitting (or RFF) problem aims to recover a rational function that best fits this input, as to minimize some given loss function.

**Hardness of rational function approximation.** To the best of our knowledge, rational function approximation has only been solved for the max deviation case (where the loss is the maximum over all the pairwise distances between the input signal and the approximation function) in Peiris et al. (2021). Various heuristics have been suggested for other instances of the problem, see Peiris et al. (2021) with references therein.

### 1.4 NOVELTY

The suggested coreset in this paper is very different from previous coreset papers. To our knowledge, this is the first coreset for stochastic signals. Most existing coresets are motivated by problems in computational geometry or linear algebra, especially clustering, and subspace approximation. Their input is usually a set of points (and not a time series), with few exceptions e.g. coresets for linear regression Dasgupta et al. (2009) that can be considered as a type of hyperplane approximation.

Our main challenges were also very different from existing coreset constructions. A typical coreset construction begins with what is known as an $\alpha$-approximation or $(\alpha, \beta)$-approximation for the optimal solution that can be easily constructed, e.g. using Arthur & Vassilvitskii (2007); Feldman & Langberg (2011); Cohen et al. (2015). From this point the main change in these papers is to compute the sensitivity or importance of each point using the Feldman-Langberg framework Feldman & Langberg (2011). The coreset is then a non-uniform random sample from the input set, based on the distribution of sensitivities.

However, in this paper, the sensitivity of a point is simply proportional to its distance from our $(\alpha, \beta)$-approximation. Therefore, the main challenge in this paper is to use our solver for rational functions, which takes time polynomial in $n$, to compute an efficient $(\alpha, \beta)$-approximation. We cannot use the existing sample techniques as in Feldman & Langberg (2011) since it might create too many "holes" (non-consecutive sub-signals) in the input signal. A solution for computing bicriteria for $k$-linear segments was suggested in Feldman et al. (2012); Jubran et al. (2021) by partitioning the input signal into consecutive equal $2k$ sub-signals, so that at least half of them would contain only a single segment, for every $k$-segmentation. In our case, a $k$ rational function cannot be partitioned into, say, $O(k)$ linear or even polynomial functions.

Instead, we computed a "weak coreset", which guarantees a desired approximation for a constrained version of the problem; in terms of constraints on both its input signal and feasible rational functions. We then combine this weaker coreset with a merge-reduce tree for maintaining an $(\alpha, \beta)$-approximation, which is very different from the classic merge-reduce tree that is used to construct coresets for streaming data. This tree of height at most $\lceil \log \log n \rceil$ is also related to the not-so-common running time and space complexity of our suggested coreset.

We hope that this coreset technique will be generalized in the future for more involved stochastic models, such as graphical models, Baysian networks, and HMMs.

## 2 CORESET FOR RATIONAL FUNCTION FITTING

### 2.1 DEFINITIONS

We assume that $k \geq 1$ is an integer, and denote by $\mathbb{R}^k$ the union of $k$-dimensional real column vectors. For every integer $n \geq 0$, we denote $[n] = \{1, \cdots, n\}$. For every vector $c = (c_1, \cdots, c_k) \in \mathbb{R}^k$ and a real number $x \in \mathbb{R}$, we denote by $\mathrm{poly}(c, x) = \sum_{i=1}^{k} c_i \cdot x^{i-1}$ the value of the polynomial of degree less than $k$ whose coefficients are the entries of $c$ at $x \in \mathbb{R}$. For simplicity, we assume $\log(x) := \log_2(x)$. A *weighted set* is a pair $(P, w)$ where $P \subseteq \mathbb{R}^2$ and $w : P \to [0, \infty)$ is a weights function. A partition $\{P_1, \cdots, P_\theta\}$ of a set $P \subset \mathbb{R}^2$ is called *consecutive* if for every $i \in [\theta - 1]$ we have $\min \{x \mid (x, y) \in P_{i+1}\} > \max \{x \mid (x, y) \in P_i\}$. A query $q \in (\mathbb{R}^k)^2$ is a pair of k-dimensional coefficients vectors. For any integer $n \geq 1$ *an n-signal* is a set $P$ of

pairs $\{(1, y_1), \cdots, (n, y_n)\}$ in $\mathbb{R}^2$. Such an $n$-signal corresponds to an ordered set of $n$ reals, a discrete signal, or to the graph $(1, g(1)), ..(n, g(n))$ of a function $g : [n] \to \mathbb{R}$. A set $P \subset \mathbb{R}^2$ is *an interval of an $n$-signal* if $P := \{(a, y_a), (a+1, y_{a+1}), \cdots, (b, y_b)\}$ for some $a, b \in [n]$ such that $a < b$. The sets $\{a, a+1, \cdots, b\}$ and $\{y_a, y_{a+1}, \cdots, y_b\}$ are the interval's *first and second coordinates*, respectively. Given a function $f : \mathbb{R} \to \mathbb{R}$, the *projection* of $P$ onto $f$ is defined as $\{(a, f(a)), \cdots, (b, f(b))\}$. The RFF problem is defined as follows. Note that the following definition of the rational function is inspired by Proposition 1.

**Definition 2** (RFF). *We define* ratio $: (\mathbb{R}^k)^2 \times \mathbb{R} \to \mathbb{R}$ *to be the function that maps every pair* $q = (c, c') \in (\mathbb{R}^k)^2$ *and any $x \in \mathbb{R}$ to*

$$\mathrm{ratio}(q, x) = \mathrm{ratio}((c, c'), x) := \begin{cases} \dfrac{\mathrm{poly}(c, x)}{1 + x \cdot \mathrm{poly}(c', x)} & \text{if } 1 + x \cdot \mathrm{poly}(c', x) \neq 0 \\ \infty & \text{otherwise.} \end{cases}$$

*For every pair $(x, y) \in \mathbb{R}^2$, the loss of approximating $(x, y)$ via a rational function $q$ is defined as* $D(q, (x, y)) := |y - \mathrm{ratio}(q, x)|$. *For a finite set $P \subset \mathbb{R}^2$ we define the* RFF *loss of fitting $q$ to $P$ as*

$$\ell(P, q) = \sum_{p \in P} D(q, p) = \sum_{(x,y) \in P} |y - \mathrm{ratio}(q, x)|.$$

*More generally, for a weighted set $(P, w)$ we define the* RFF *loss of fitting $q$ to $(P, w)$ as*

$$\ell((P, w), q) = \sum_{p \in P} w(p) D(q, p) = \sum_{(x,y) \in P} w((x, y)) |y - \mathrm{ratio}(q, x)|. \tag{1}$$

A coreset construction usually requires some rough approximation to the optimal solution as its input; see Section 1.4. Unfortunately, we do not know how to compute even a constant factor approximation to the RFF problem in Definition 2 in near-linear time, but only in $(2kn)^{O(k)}$ time; see Lemma 8. Instead, our work is mostly focused on efficiently computing an $(\alpha, \beta)$ or bicriteria approximation as Feldman & Langberg (2011) defined at Section 4.2.

**Definition 3** (($\alpha, \beta$)-approximation). *Let $P$ be an interval of an $n$-signal. Let $\{P_1, \cdots, P_\beta\}$ be a consecutive partition of $P$, and $q_1, \cdots, q_\beta \in (\mathbb{R}^k)^2$. The set $\{(P_1, q_1) \cdots, (P_\beta, q_\beta)\}$ of $\beta$ pairs is an $(\alpha, \beta)$-approximation of $P$ if*

$$\sum_{i=1}^{\beta} \ell(P_i, q_i) \leq \alpha \cdot \min_{q \in (\mathbb{R}^k)^2} \ell(P, q).$$

*If $\beta = 1$ the $(\alpha, 1)$-approximation $B = \{(P_1, q_1)\}$ is called an $\alpha$-approximation. For every $i \in [\beta]$ we denote by $P_i'$ the projection of $P_i$ onto $q_i$, and by $\bigcup_{i=1}^{\beta} P_i'$ the projection of $P$ onto $B$. We define the set of bicriterias of $P$ as the union of all the $(\alpha', \beta')$-approximations of $P$.*

A coreset for the RFF problem is defined as follows. Similarly to Rosman et al. (2014) it includes an approximation to allow a coreset construction despite the lower bound from Claim 1.

**Definition 4** ($\epsilon$-coreset). *Let $P \subseteq \mathbb{R}^2$ be an $n$-signal, and let $\epsilon > 0$ be an error parameter. Let $B := \{(P_1, q_1), \cdots, (P_\beta, q_\beta)\}$ be a bicriteria approximation of $P$ (see Definition 3), and let $(C, w)$ be a weighted set. The tuple $(B, C, w)$ is an $\epsilon$-coreset for $P$, if for every $q \in (\mathbb{R}^k)^2$ we have*

$$\left| \ell(P, q) - \ell((C, w), q) - \ell(P', q) \right| \leq \epsilon \cdot \ell(P, q),$$

*where $P' := \{(x, \mathrm{ratio}(q_i, x)) \mid i \in [\beta], (x, y) \in P_i\}$ is the projection of $P$ onto $B$. The storage space required for representing such coreset is in $O(|C| + \beta k)$.*

## 2.2 ALGORITHMS OVERVIEW

For less technical intuition for the algorithms see Section J.4 at the appendix.

For simplicity, in the following we consider $k$ to be a constant. The input to Algorithm 1, which is the main algorithm, is an $n$-signal $P$, and input parameters $\epsilon, \delta \in (0, 1/10]$. Its output, with probability at least $1 - \delta$, is an $\epsilon$-coreset $(B, C, w)$ of $P$ with size in $O\left(n^{o(1)}/\epsilon^2\right)$ for constant $\delta$.

**Algorithm 2: coreset construction.** As in Definition 4, the coreset consists of an $(\alpha, \beta)$-approximation $B := \{(P_1, q_1), \cdots, (P_\beta, q_\beta)\}$ of the input set $P$, and a weighted set $(C, w)$. The weighted set $(C, w)$ consists of a small sample $C \subseteq P$ and its weights function $w : C \to (0, \infty)$. As in Feldman et al. (2012), for every $i \in [\beta]$ the probability of placing a point $(x, y) \in P_i$ into the sampled set $C$ is $\mathrm{Dist}(q, p)$. Hence, a point $(x, y) \in P$ whose $y$-value is far (not approximated well) from $B$ would be sampled with high probability, but a point that is close to $B$ will probably not be sampled. The weight $w(p)$ of a point is inverse proportional to its probability to be chosen, so that the sum of distances to any query (rational function) is the same as its expectation.

The rest of the algorithms aim to compute the $(\alpha, \beta)$-approximation $B$.

**Algorithm 1: bi-criteria tree construction.** We compute $B$ via a balanced $\beta$-tree, which is similar to the classic merge-reduce tree that is usually used to compute coresets for streaming data Braverman et al. (2020). However, the merge and reduce steps are different, as well as the number of children in each node. Each leaf of this tree corresponds to a 1-approximation (i.e. optimal solution) for a consecutive set of $\beta := \Theta\left((n^{1/\log\log(n)}\right)$ input points, which is computed via Algorithm 5. Hence, there are $\Theta(n/\beta)$ leaves. An inner node in the $i$th level corresponds to an $(\alpha_i, \beta_i)$-approximation of its $\beta$ child nodes, it has $O\left(\beta^i\right)$ leaves and $O\left(\beta^{i+1}\right)$ input points of its sub-tree, for every $i \in [\ell]$, where $\ell = \lceil \log\log(n) \rceil - 1$ is the number of levels in the tree; follows since $\left(n^{1/\log\log(n)}\right)^{\log\log(n)} = n$. Here, $\alpha_i = 3^i$, $\beta_i = O(1)^i$, and thus $(\alpha, \beta) = (\alpha_\ell, \beta_\ell) \in \left(O(\log(n)), \log(n)^{O(1)}\right)$.

**Algorithm 3: the merge-reduce step.** This step is computed in each inner node of the tree. For an inner node in the $i$th level, where $i \in \{2, \cdots, \ell\}$, the input is a set $B$ of size $\beta$, where each $B_j \in B$ is an $(0, r_j)$-approximation of $P_j$ (i.e. $P_j$ is projected unto $B_j$), where $P_1, \cdots, P_\beta$ is an equally-sized consecutive partition of an interval of an $n$-signal $P$, and the output is a bicriteria for $P$; see Algorithm 3 and Fig. 1. This is by computing the following for every possible subset $G \subseteq [\beta]$ of size $\beta - 6k + 3$: (i) Compute $\{(\cdot, q_G)\}$ an $\alpha$-approximation, for some $\alpha \geq 1$, for $\bigcup_{j\in g} P_j$. (ii) For each $j \in G$ set $\ell_j := \ell(P_j, q_G)$ as the loss of $q_G$ for $P_j$. (iii) Set $G' \subset G$ to be the union of the $6k - 3$ indices $j \in G$ with the largest value $\ell_j$. (iv) Set $s_G := \sum_{j\in G\setminus G'} \ell_j$; the sum of the losses $\ell_j$ excluding the largest $|g'|$.

The final bicriteria approximation for the inner node is the one that minimizes $s_G$, among every subset $g \subset B$ of child nodes above. More precisely, we take its part that approximates the union of $|B| - 2|G'|$ children whose approximation error is $s_g$, and take its union with the $2|G'|$ original (input) bicriterias that correspond to the child nodes $(B \setminus g) \cup g'$. The final approximation error in the inner node is thus the minimum of $s_G$ over every $G \subseteq B$ of size $\beta - 6k + 3$.

**Algorithm 4: restricted coresets for rational functions.** Computing the $\alpha$-approximation in Step (i) above can be done by computing the optimal solution, with details shown in Lemma 8. However, this would take $n^{O(1)}$ and not quasi-linear time as desired. To this end, Algorithm 4 constructs a restricted coreset for rational functions that is restricted in the following two senses: (i) It assumes that the input signal is projected onto a bicriteria approximation, which is indeed the case in Step (i) of the *merge-reduce step* above. (ii) It approximates only specific rational functions, which are $2^k$-bounded over the first coordinate of the input signal; see Definition 5.

We handle the second assumption by removing the $O(k)$ sets where the second assumption does not hold (via the exhaustive search described above). It should be emphasized that the final coreset construction has no such restrictions or assumptions for either its input or queries.

This restricted coreset is computed on each child node, so that in Step (i) above we compute the $\alpha$-approximation only on the union of $|C| - |C'|$ coresets that corresponds to the chosen $|C| - |C'|$ internal nodes. The size of the coreset, that fail with probability at most $\delta \in (0, 1/10]$, is $m \in O\left((\log(\beta n/\delta))^2\right)$ and thus the running time of our approximation algorithm (Algorithm 3) is $m^{O(1)}$, which is in $O\left(n^{o(1)}\right)$, for every constant $\delta > 0$. Algorithm 4 computes this restricted coreset by: (i) Partitioning the input into chunks of exponentially increasing sizes; see Fig. 5. (ii)

Computing a sensitivity based sub-sample for each chunk. (iii) Returning the union of those coresets after appropriate re-weighting.

As we show in the proof, the sensitivity for the RFF inside each chunk can be reduced to the sensitivity of the polynomial fitting problem, which was previously tackled; see Corollary 2.

### 2.3 ALGORITHMS

**From bicriteria to sampling-based coresets.** In Feldman et al. (2012) a sampling based coreset construction algorithm was suggested for the $k$-segments problem. This algorithm requires as input an $(\alpha, \beta)$-approximation as defined above. With some modifications to this algorithm, we can efficiently construct a coreset for the RFF problem. The missing part is a bicriteria approximation for the RFF, which is the main focus and main contribution of this work.

**Optimal solution ($\alpha = \beta = 1$) in polynomial time.** Using polynomial solvers, and previous work (see e.g., Marom & Feldman (2019) with references therein) it can be proven that given a set $P$ of $n$ points on the plane, we can compute the optimal fitting rational function to the point, i.e., the rational function that minimizes Equation 1 at Definition 2 in $(2kn)^{O(k)}$ time; see Lemma 8.

**Efficient $(1, \beta)$-approximation for large $\beta$.** Using the polynomial time optimal solution above, we can compute an $(1, \beta)$-approximation to a $n$-signal $P$, for a large $\beta$. This is by partitioning the input into many ($\beta$) small subsets, and apply the optimal solution to each subset, which is relatively fast as each set is very small, more precisely at $n \cdot (2k\beta)^{O(k)}$; see Algorithm 5 for suggested implementation, note that there $\beta$ corresponds to $n/\beta$ at this paragraph.

---

**Algorithm 1:** CORESET$(P, k, \epsilon, \delta)$; see Theorem 1.

**Input** : An $n$-signal $P$, an integer $k \geq 1$, and constants $\epsilon, \delta \in (0, 1/10]$.
**Output:** A tuple $(B, C, w)$, where $B$ is a bicriteria approximation of $P$, and $(C, w)$ is a weighted set.

1 $\beta := \left\lceil n^{1/\log(\log(n))} \right\rceil; \tilde{\beta} = \lceil n/\beta \rceil; \Lambda = 6k - 3$

2 Set $c^* \geq 1$ to be a constant that can be determined from the proof of Theorem 1.

3 $\lambda_1 := \left\lceil c^*(4^{k+1}k^2 + 1)\left(k^2 \log(4^{k+1}k^2 + 1) + \log\left(\frac{kn}{\delta}\right)\right)\right\rceil$

4 $B := (B_1, \cdots, B_{|B|}) := $ BATCH-APPROX$(P, \tilde{\beta})$// see Algorithm 5.

5 **while** $|X| > 2\beta$ **do**

6 $\quad$ Set $\left\{B'_1, \cdots, B'_{|B|/\psi}\right\}$ to be a partition of $B$ into equally sized consecutive sets, each of size $\psi \in [\beta, 2\beta]$, a power of 2; i.e. $B'_i := \{B_a, \cdots, B_{a+\psi}\}$ for some $a \in [|B|]$ and every $i \in [|B|/\psi]$.

7 $\quad$ Set $B_i := $ REDUCE$(B'_i, \lambda_1, \Lambda)$ for every $i \in \{1, \cdots, \psi\}$; see Algorithm 3.
$\quad$ // We reduce every set in the partitions of $B$.

8 $\quad$ $B := (B_1, \cdots, B_\psi)$

9 $B' := \left\{B_1, \cdots, B_{|B|}\right\}$ // $B'$ is an (un-ordered) set.

10 $B := $ REDUCE$(B', \lambda_1, \Lambda)$.

11 $\lambda_2 := \left\lceil \frac{c^*}{\epsilon^2} \cdot \log(n) \cdot \left(k^2 \log\log(n) + \log\left(\frac{n}{\delta}\right)\right)\right\rceil$

12 $(B, C, w) := $ SAMPLE-CORESET$(B, \lambda_2)$ // see Algorithm 2.

13 **return** $(B, C, w)$.

---

The following theorem proves the correctness of Algorithm 1. See Theorem 5 for a full proof.

**Theorem 1.** *Let $P$ be an $n$-signal, for $n$ that is a power of 2, and put $\epsilon, \delta \in (0, 1/10]$. Let $(B, C, w)$ be the output of a call to* CORESET$(P, k, \epsilon, \delta)$*; see Algorithm 1. With probability at least $1 - \delta$, $(B, C, w)$ is an $\epsilon$-coreset of $P$; see Definition 4. Moreover, the computation time of $(B, C, w)$ is in*

$$2^{O(k^2)} \cdot n \cdot n^{O(k)/\log\log(n)} \cdot \log(n)^{O(k \log(k))} \cdot \log(1/\delta)^{O(k)},$$

*and the memory words required to store $(B, C, w)$ are in $(2k)^{O(1)} \cdot \log(n)^{O(1)+\log(k)} \cdot \log(1/\delta)/\epsilon^2$. In particular, if $k$ and $\delta$ are constants a running time of $O\left(n^{1+o(1)}\right)$ and the space is in $O\left(n^{o(1)}/\epsilon^2\right)$.*

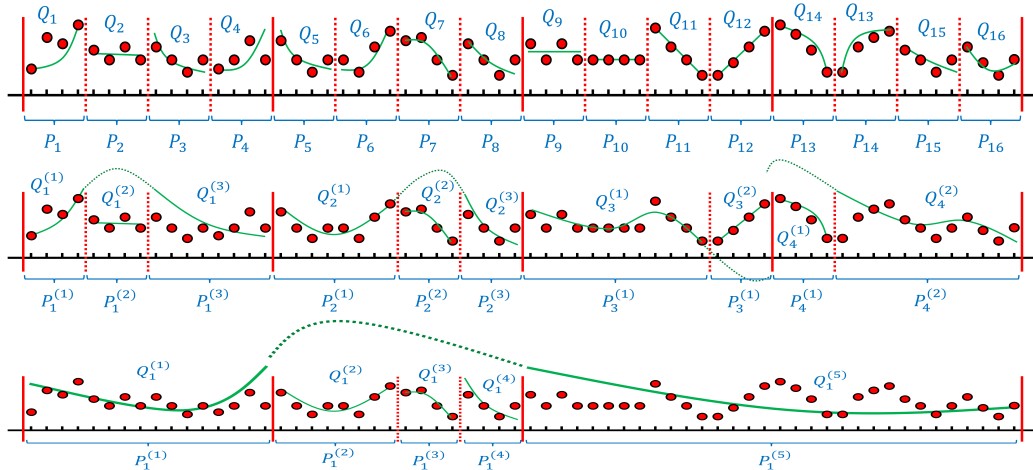

Figure 1: Illustration for Algorithm 1. **(Top)** In Line 4, an input $n$-signal $P$ (black ticks and red dots) and its partition $\{P_1, \cdots, P_{16}\}$ into $\psi = 16$ sets via a call to Algorithm 5. This call also computes $(1, 1)$-approximation $B_i = \{(P_i, Q_i)\}$ for every $P_i$ (green curves), see Definition 3. In Line 6, the set $\{B_1, \cdots, B_{16}\}$ is partitioned into $B = B_1' \cup B_2' \cup B_3' \cup B_4'$, where each such set contains 4 elements from $B$. **(Middle)** In Line 7, for every $i \in [4]$ we set $B_i = \left\{ \left(P_i^{(1)}, Q_i^{(1)}\right), \cdots, \left(P_i^{(|B_i|)}, Q_i^{|B_i|}\right) \right\}$ as the output of a call to REDUCE$(B_i')$. For every $i \in [3]$ the projection of every $\bigcup_{i=1}^{|B_i|} P_i^{(j)}$ onto $B_i$ is denoted in green. **(Bottom)** The process above is repeated, with $B := \{B_1, B_2, B_3, B_4\}$. $B$ is therefore partitioned into sets of size $\beta = 4$, i.e., only 1 such set $B$, and then REDUCE$(B)$ is called, which reduces the size of $B$ to 5.

## 3 EXPERIMENTAL RESULTS

We implemented our coreset construction from Algorithm 1 in Python 3.7 and in this section we evaluate its empirical results, both on synthetic and real-world datasets. More results are placed in the supplementary material; see Section I. Open-source code can be found in Code (2022).

Note that since it is non trivial how to accelerate the computation of the optimal solution using the coreset we will focus on the quality of the compression, similarly to video compression analysis.

**Hardware.** We used a PC with an Intel Core i7-10700, NVIDIA GeForce GTX 1660 SUPER (GPU), and 16GB of RAM.

**The gap between theory and practice.** In practice, we apply two minor changes to our theoretically-provable algorithms, which have seemingly neglectable effect on their output quality, but aim to improve their running times. Those changes are: **(i):** Line 11 of Algorithm 3 computes the query with the smallest cost $\ell((S, w), r)$ over every $r \in \mathbb{R}^k \times \mathbb{R}^k$. Instead, to reduce the computational time, we iterate only over the queries in FAST-CENTROID-SET$(S, 64)$; see Algorithm 6. **(ii):** Line 2 of Algorithm 5 computes, for every $P_i \subset \mathbb{R}^2$ defined in Algorithm 5, the query with the smallest cost $\ell(P_i, r)$ over every $r \in \mathbb{R}^k \times \mathbb{R}^k$. Instead, to reduce the computational time, for every such $Pi_i$, we iterate only over the queries in FAST-CENTROID-SET$(P_i, 64)$; see Algorithm 6. **(iii)** At Line 11 of Algorithm 4, the set $S_i^j$ was sampled from $P_i^j$ an interval of an $n$-signal, where each point $(x, y) \in P_i^j$ was sampled with probability $s'(x)$. We observed, in practice, that the probabilities assigned by $s'$ were sufficiently close to $1/|P_i^j|$ for most of the points. Hence, to reduce the computational time, we sampled the set $S_i^j$ uniformly from $P_i^j$.

**Global parameters:** We used the degree $k = 2$ in our experiments, since it seemed as a sufficiently large degree to allow a visually pleasing approximation as seen, for example, in Figure 13.

**Competing methods.** We consider the following compression schemes:
**(i):** RFF-coreset$(P, \lambda)$ - The implementation based on Algorithm 1, where we set $\beta = 32$, $\tilde{\beta} = 32$, $\lambda_1 = 32$, and $\Lambda = 0$.

**(ii):** FRFF-coreset$(P, \lambda)$ - A heuristic modification to RFF-coreset above, where the call to REDUCE at Line 7 of Algorithm 1 is replaced with a call to FAST-CENTROID-SET; see Algorithm 6. This should boost the running time by slightly compromising accuracy.

**(iii):** Gradient$(P, \lambda)$ - A rational function $q \in \left(\mathbb{R}^k\right)^2$ was fitted to the input signal via scipy.optimize.minimize with the default values, where the function minimized is $\ell(P, q)$, and the starting position is $\{0\}^k \times \{0\}^k$. Then, a coreset was constructed using the provable algorithm SAMPLE-CORESET$(\{(P, q)\}, \lambda)$.

**(iv):** $L_\infty$Coreset$(P, \lambda)$ - A rational function $q \in \arg\min_{q' \in (\mathbb{R}^k)^2} \max_{p \in P} D(p, q')$ was computed based on Peiris et al. (2021). Then, a coreset was constructed using the provable algorithm SAMPLE-CORESET$(\{(P, q)\}, \lambda)$.

**(v):** RandomSample$(P, \lambda)$ - returns a uniform random sample of size $1.5 \cdot \lambda$ from $P$.

**(vi):** NearConvexCoreset$(P, \lambda)$ - The coreset construction from Tukan et al. (2020), chosen to be of size $1.5 \cdot \lambda$.

**Coreset size.** In all the following experiments, for fair comparison, the input parameters of all the competing methods above were tuned as to obtain an output coreset of the same desired size. Note that there can be small noise in the sizes due to repeated samples of the same element.

**Repetitions.** Each experiment was repeated 100 times. All the results are averaged over the tests.

**Evaluation.** The quality of a given compression scheme (from the competing methods above) is defined as $\varepsilon(q) = 100 \cdot \frac{|\ell' - \ell|}{\ell}$, where $\ell'$ is the loss in Equation 1 when plugging in the compression and some query $q$, and $\ell$ is the same loss but when plugging the full input signal and the same query $q$. We tested a couple of different options for such a query $q$: **(i):** as it is hard to compute the optimal solution $q^*$ which minimizes Equation 1, we sampled a set $Q$ of 1024 queries using Algorithm 6 and recovered the query $q \in Q$ which has the smallest loss over the full data. This query aims to test the compression accuracy in a near-optimal scenario. We then presented $\varepsilon(q)$ for the q above. **(ii):** For every individual compression scheme, we picked the query in $q \in Q$, which yields the largest value for $\epsilon(q)$; i.e., the least satisfied query.

## 3.1 SYNTHETIC DATA EXPERIMENT

In this experiment, we aim to reconstruct a given noisy homogeneous recurrence sequence's explicit representation, as explained in Section 1 (specifically, see Definition 1 and Proposition 1).

**Dataset.** Following the motivation in Section 1, the data ($n$-signal) $P$ we used in this test is simply a set of $n$ values $\{F(x) \mid x = j/n - 1/2, j \in [n]\}$ of the generating function of a noisy Fibonacci series, i.e., $F(x) = \sum_{i=0}^{99} \left(s_{i+1} \cdot x^i\right)$ where $s_i$ is the $i$'th element in the Fibonacci series with Gaussian noise with zero mean and a standard deviation of $0.25$.

**Results.** Fig. 2 presents the results, along with error bars that present the $25\%$ and $75\%$ percentiles.

## 3.2 REAL-WORLD DATA EXPERIMENT

In this experiment we consider an input signal which contains $n$ readings of some quantity (e.g., temperature or pressure) over time. We aim to fit a rational function to such a signal, with the goal of predicting future readings and uncovering some behavioral pattern.

**Dataset.** In this experiment we consider the **Beijing Air Quality Dataset** Chen (2019) from the public UCI Machine Learning Repository Asuncion & Newman (2007). The dataset contains $n = 7344, 8760, 8760, 8784$ readings, respectively for the years 2013 to 2016. Each reading contains "the temperature (degree Celsius, denoted by TEMP), pressure (hPa, denoted by PRES), and dew point temperature (degree Celsius, denoted by DEWP)". For each year and property individually, we construct an $n$-signal of the readings over time. Missing entries were replaced with the average property value. Fig. 9 presents the dataset readings along with our rational function fitting algorithm and Scipy's rational function fitting. The **Italy Air Quality Dataset** Vito (2016) is also tested; see Section I.2 in the appendix.

**Results.** Fig. 3 presents the results for the year 2016, along error bars that present the $25\%$ and $75\%$ percentiles. Graphs for other years along with results for dataset Vito (2016) are placed in Section I.

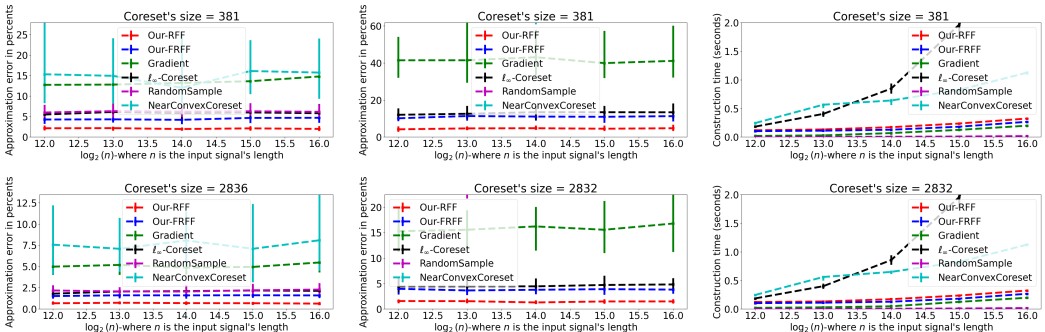

Figure 2: Results of the experiment from Section 3.1. (Left + Middle): The $X$-axis presents, on a logarithmic scale, the size of the input signal, and the $Y$-axis presents the approximation error of each compression scheme, for given compression sizes of $382$ and $2824$ respectively. The upper and lower rows present Evaluations (i) and (ii) respectively. (Right): The computational time for each of the two coreset sizes. The evaluation method does not affect those times.

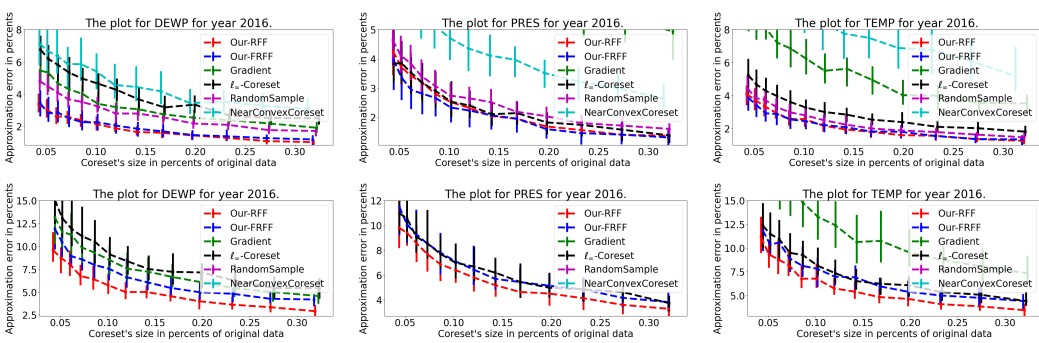

Figure 3: Results of the experiment from Section 3.2. The $X$-axis presents the size of the compression - percentage from the original data, and the $Y$-axis presents the approximation error of each compression scheme. The top and bottom rows present Evaluations (i) and (ii) respectively.

## 3.3 DISCUSSION

Fig. 4 demonstrates that rational function fitting is more suitable than polynomial fitting, for a relatively normal dataset. It also shows that computing any of those fitting functions either on top of the full data or our coreset produces similar results. Lastly, Fig. 2 and 3 demonstrate that our method and its variants achieve, almost consistently, better approximation quality in faster time, as compared to most of the competing methods; the only faster methods were the uniform sample and gradient which yielded significantly worse results. For a more in depth discussion see Section J which is at the appendix due to space limitation.

## 4 FUTURE WORK AND CONCLUSION

This paper provides a coreset construction that gets a time-series and returns a small coreset that approximates its sum of (fitting) distances to any rational functions of constant degree, up to a factor of $1 \pm \epsilon$. The size of the coreset is sub-linear in $n$ and quadratic in $1/\epsilon$. Our main application is fitting to Auto-Regression model, whose generative functions are rational. While we focused on sum of errors (distances), we expect easy generalization to squared-distances, robust M-estimators and any other loss function that satisfies the triangle inequality, up to a constant factor, as in other coreset constructions. We believe that the new suggested technique initializes a line of research that would enable sub-linear time algorithms with provable approximation for more sophisticated stochastic models such mentioned at the start of the paper in Section 1.

## ETHICS STATEMENT

To the best of our knowledge, there are no ethical concerns for our work due to the following:

- The work is of theoretical nature that aims to develop efficient coresets for rational functions (and approximation is a major part of the contribution).
- All the datasets which we tested our methods on where from the public UCI Machine Learning Repository Asuncion & Newman (2007), and more precisely Vito (2016) and Chen (2019).

## REPRODUCIBILITY STATEMENT

In all our tests we included error bars, hardware used, global parameters, dataset (citing existing data or its generalization method) and the paper contains full pseudo-code for all the Algorithms. Where there were assumptions on the data they were stated explicitly.

As stated in Code (2022), the authors commit to publish the code for all of the tests (or part of them) upon acceptance of this paper or reviewer request.

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

# Appendix

## Table of Contents

## A  ADDITIONAL MOTIVATION FOR RFF

In the following figure we demonstrate that in some cases rational functions can yield better approximations than polynomials, this is essentially a variation of the known Runge's phenomenon Epperson (1987). We do not use a rational function, that is commonly used to demonstrate Runge's phenomenon Epperson (1987), since we also want to demonstrate the superiority of our methods upon existing solvers, and if all the points are exactly on a rational function existing rational interpolation methods can solve this case as well; for example Padé approximant as mentioned in Baker Jr (1964) or rational function fitting for max deviation as mentioned in Peiris et al. (2021).

While rational functions can give better fitting than polynomials at some cases the same also holds for polynomials, where at some instances polynomials would yield a significantly better fitting. For an example where the polynomial fitting yielded better results than our methods see Figure 9 and Figure 12 with discussion at Section J.3.1.

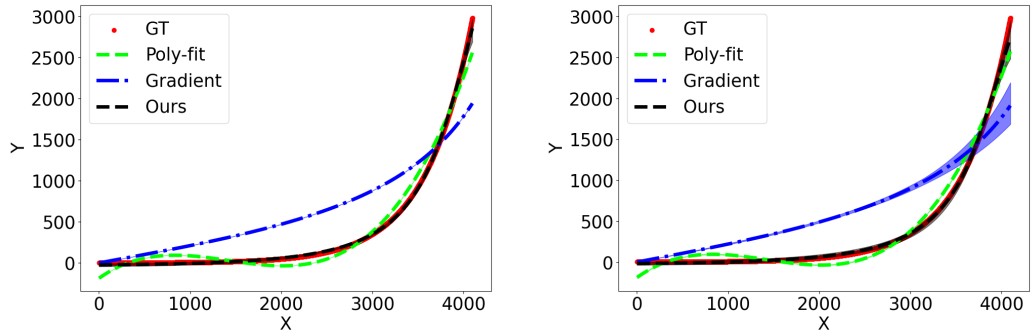

Figure 4: **RFF illustration.**  A time-series $f(x) = e^{x/512}$ over $\{1, \cdots, 2^{12}\}$ in red, which is denoted by GT, an abbreviation of ground truth. The goal is to approximate it via: (black) a rational function computed using our algorithm `FRFF-coreset` from the Experimental Results (see Section 3), (blue) a rational function computed using the `Scipy.optimize.minimize` which aims to minimize the same RFF loss as in Equation 1, and (green) a polynomial of degree 3 computed using the `numpy.polyfit` function, which minimizes the sum of squared distances between the polynomial and the input. For fair comparison, all three methods have been allowed 4 free parameters. (Left): All 3 methods applied to the original signal. (Right): A coreset of size $< 10\%$ of the input was first computed for the given signal via Algorithm 1. Then, all 3 methods were applied on the coreset points only. The error bars are from 10 experiments.

## B  ADDITIONAL ALGORITHMS

### B.1  SAMPLE BASED ON A BICRETIRIA APPROXIMATION ALGORITHM

Algorithm 2 gets as input a bicretiria approximation of the RFF problem for some given input $n$-signal $P$; see Definition 3. The algorithm utilizes this rough approximation in order to compute, in linear time, an $\epsilon$-coreset as in Definition 4 via sensitivity sampling. The formal statement is given in Lemma 1. This algorithm is a modified version of the algorithm presented for the $k$-segments problem in Feldman et al. (2012).

The following lemma states the desired properties of Algorithm 2; see Lemma 6 for its proof.

**Lemma 1.** *Let $B := \{(P_1, q_1), \cdots, (P_\beta, q_\beta)\}$ be an $(\alpha, \beta)$-approximation of some $n$-signal $P$, for some $\alpha > 0$; see Definition 3. Put $\epsilon, \delta \in (0, 1/10]$, and let*

$$\lambda := \left\lceil \frac{c^*}{\epsilon^2}(\alpha + 1)\big(k^2 \log(\alpha + 1) + \log(1/\delta)\big) \right\rceil,$$

*where $c^* \geq 1$ is a constant that can be determined from the proof. Let $(B, C, w)$ be the output of a call to* SAMPLE-CORESET$(B, \lambda)$*; see Algorithm 2. Then, Claims (i)–(ii) hold as follows:*

*(i)* $(B, C, w)$ *can be stored using* $O(\lambda + \beta k)$ *memory.*

*(ii) With probability at least* $1 - \delta$, *we have that* $(B, C, w)$ *is an* $\epsilon$-*coreset of* $P$; *see Definition 4.*

---

**Algorithm 2:** SAMPLE-CORESET$(B, \lambda)$; see Lemma 1.

---

**Input** : A bicirteria approximation $B := \{(P_1, q_1), \cdots, (P_\beta, q_\beta)\}$ of some $n$-signal $P$; see
Definition 3. An integer $\lambda \geq 1$ for the sample size.
**Output:** A tuple $(B, C, w)$, where $B$ is a bicriteria approximation of $P$, and $(C, w)$ is a
weighted set.

1 $c := \displaystyle\sum_{i=1}^{\beta} \ell(P_i, q_i)$

2 **if** $c \in \{0, \infty\}$ **then**
3 $\quad$ Let $w : \mathbb{R}^2 \to \{0\}$ such that for every $p \in \mathbb{R}^2$ we have $w(p) = 0$.
4 $\quad$ **return** $(B, \emptyset, w)$.
5 $s(p) := D(q_i, p)/c$ for every $i \in [\beta]$ and every $p \in P_i$.
6 Pick a sample $S \subset P$ of $\lambda$ points from $P$, where $S$ is a multi-set and each point $p \in S$ is
sampled i.i.d. with probability $s(p)$; observe that there might be repetitions in $S$.
7 Set $r(p)$ as the number of repetitions of $p$ in the multi-set $S$, for every $p \in P$.
8 $w(p) := r(p)/(\lambda \cdot s(p))$ for every $p \in S$.
9 $S' := \{(x, \mathrm{ratio}(q_i, x)) \mid i \in [\beta], (x, y) \in P_i \cap S\}$ // project the labels of
every set $P_i \cap S$ onto $q_i$ and take their union.
10 **for** *every* $i \in \{1, \cdots, \beta\}$ **do**
11 $\quad$ $w\big((x, \mathrm{ratio}(q_i, x))\big) := -w(p)$ for every $p \in S \cap P_i$
12 $C := S \cup S'$
13 **return** $(B, C, w)$

---

### B.2 FROM LARGER TO SMALLER VALUES OF $\beta$

In this section, we show how, given an $(\alpha, \beta)$-approximation with large $\beta$, we can recover an $(\alpha', \beta')$-approximation with $\beta' < \beta$ but a larger $\alpha' > \alpha$. This is achieved by computing an approximation to the projection of the set of points approximated by the $(\alpha, \beta)$-approximation onto the $(\alpha, \beta)$-approximations. This is implemented in Algorithm 3, which utilizes Algorithm 4.

To efficiently compute the previously stated bicriteria we will utilize restricted coresets. It should be emphasized that the final coreset will not contain such restriction. One of the limitation on the restricted coresets will involve the following definition.

**Definition 5** ($\rho$-bounded function). *For every* $X \subset \mathbb{R}, \rho \in [1, \infty)$, *and any* $(c, c') \in (\mathbb{R}^k)^2$ *we say that* $(c, c')$ *is* $\rho$-*bounded over* $X$ *if and only if*

$$\frac{\max\limits_{x \in X} f(x)}{\min\limits_{x \in X} f(x)} \leq \rho,$$

*where the function* $f : \mathbb{R} \to \mathbb{R}$ *maps every* $x \in \mathbb{R}$ *to* $f(x) = \dfrac{1}{\big|1 + x \cdot \mathrm{poly}(c', x)\big|}$.

**Overview of Algorithm 3** The input for the algorithm is $P$ an interval of $n$-signal which is projected onto some set of $(\alpha, \beta)$-approximations. This projection is represented by the set $B$, where each element $B_i \in B$ is a $(0, \beta)$-approximation for some $P_i$, and $P_1, \cdots, P_{|B|}$ is a consecutive partition of $P$. The algorithm also receives a parameter $\lambda$ which control the size of the output, and a parameter $\delta$ which control the trade-off between the running time and robustness. The algorithm returns $B'$ a bicriterai of $P$, where the size of $B'$ is smaller than $\sum_{i=1}^{|B|} |B_i|$. The algorithm runs in $O(|P|^{1+o(1)})$ time. This $(\alpha, \beta)$-approximation is computed as mentioned in Section 2.2.

---

**Algorithm 3:** REDUCE$(B, \lambda, \Lambda)$; see Lemma 2.

---

**Input** : A set $B := \{B_1, \cdots, B_\beta\}$, where each $B_i \in B$ is an $(0, r_i)$-approximation of $P_i$, i.e. $P_i$ is projected unto $B_i$, and $\{P_1, \cdots, P_\beta\}$ is an equally-sized consecutive partition of $P$, some interval of an $n$-signal; see Figure 6 and Definition 3. Integers $\lambda \geq 1$ and $\Lambda \geq 0$.

**Output:** A bicriteria approximation $B'$ of $P$; see Definition 3.

---

1   $\ell^* := \infty; B' := \bigcup\limits_{i=1}^{\beta} B_i$

2   **for** *every $B_i \in B$* **do**

3      Identify $\left\{ B_i^{(1)}, \cdots, B_i^{(r_i)} \right\} := B_i$.

4      **for** *every $B_i^{(j)} \in B_i$* **do**

5         $\left( S_i^{(j)}, w_i^{(j)} \right) :=$ MINI-REDUCE $\left( \left\{ B_i^{(j)} \right\}, \lambda \right)$ // see Algorithm 4.

6   **for** *every set $G \subset \{1, \cdots, \beta\}$ of size $|G| = \beta - \Lambda$* **do**

7      $S_G := \emptyset$.

8      **for** *every $i \in G$ and $B_i^{(j)} \in B_i$* **do**

9         Set $w_G(p) := w_i^{(j)}(p)$ for every $p \in S_i^{(j)}$.

10        $S_G := S_G \cup S_i^{(j)}$

11     Set $q_G \in \arg\min\limits_{q \in (\mathbb{R}^k)^2} \ell\big((S_G, w_G), q\big)$; see Definition 2 and Lemma 8.

12     **for** *every $i \in G$* **do**

13       $\ell_i := \ell(P_i, q_G)$

14     Set $G' \subset G$ to be the union of the $6k - 3$ indices $i \in G$ with the largest value $\ell_i$. Ties broken arbitrarily.

15     **if** $\left( \sum_{i \in G \setminus G'} \ell_i \right) < \ell^*$ **then**

16       $\ell^* := \left( \sum_{i \in G \setminus G'} \ell_i \right)$ // update smallest loss

17       Set $\{R_1, \cdots, R_\gamma\}$ to be the smallest partition of $G \setminus G'$ such that for every $i \in [\gamma]$ we have $G' \cap [\min(R_i), \max(R_i)] = \emptyset$, and for any $i, j \in [\gamma]$, where $i \neq j$, we have $R_j \cap [\min(R_i), \max(R_i)] = \emptyset$.
        // Via simple greedy partition; see Fig 5

18       $P_i' := \big\{ \big(x, \text{ratio}(q, x) \mid (x, y) \in P_i\big) \big\}$ for every $i \in G$ // the projection of $P_i$ onto $q$.

19       $P_i^* := \bigcup\limits_{\psi \in R_i} P_i'$, for every $i \in [\gamma]$
        // Union of all the sets $P_\psi'$ with index $\psi$ in $R_i$.

20       $B' := \big\{ \big(P_1^*, q_G\big), \cdots, \big(P_\gamma^*, q_G\big) \big\} \cup \left\{ B_i^{(j)} \in B_i \mid i \in G' \cup \big([\beta] \setminus G\big) \right\}$

21   **return** $B'$.

---

Note that in the following lemma $\lambda$ is different than in Lemma 3. This is since Algorithm 4 would be called as a subroutine at most $n$ times, and as such we need to adjust the failure probability in this use of Lemma 3 to $\delta/n$.

The following lemma states the desired properties of Algorithm 3; see Lemma 14 for its proof.

**Lemma 2.** *Let $B := \{B_1, \cdots, B_\beta\}$ such that there is $\{P_1, \cdots, P_\beta\}$ an equally-sized consecutive partition of some interval of $n$-signal $P$, $|P| \geq 2k$, where each $B_i \in B$ is an $(0, r_i)$-approximation of $P_i$, i.e. $P_i$ is projected unto $B_i$; see Figure 6 and Definition 3. Put $\epsilon, \delta \in (0, 1/10]$, and let*

$$\lambda := \left\lceil \frac{c^*}{\epsilon^2} (4^{k+1} k^2 + 1) \left( k^2 \log_2(4^{k+1} k^2 + 1) + \log_2 \left( \frac{nk}{\delta} \right) \right) \right\rceil,$$

*be an integer, where $c^* \geq 1$ is a constant that can be determined from the proof. Let $B'$ be the output of REDUCE$(B, \lambda, 6k - 3)$; see Algorithm 3. With probability at least $1 - \delta$, we have that $B'$ is a $(1 + 10\epsilon, \beta^*)$-approximation of $P$ for some $\beta^* \geq 1$; see Definition 3.*

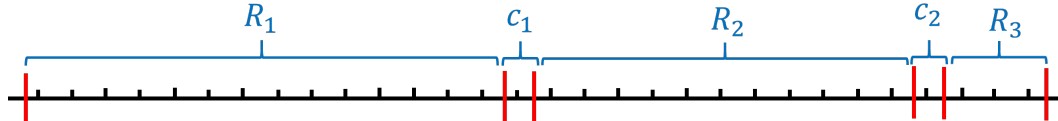

Figure 5: A set $[\beta]$ for $\beta = 30$ of indices (black ticks), a set $C = \{c_1, c_2\} \subseteq [\beta]$ of 2 indices, and a partition $R_1 \cup R_2 \cup R_3 = [\beta] \setminus C$ of the indices not in $C$, as described in Line 17 of Algorithm 3.

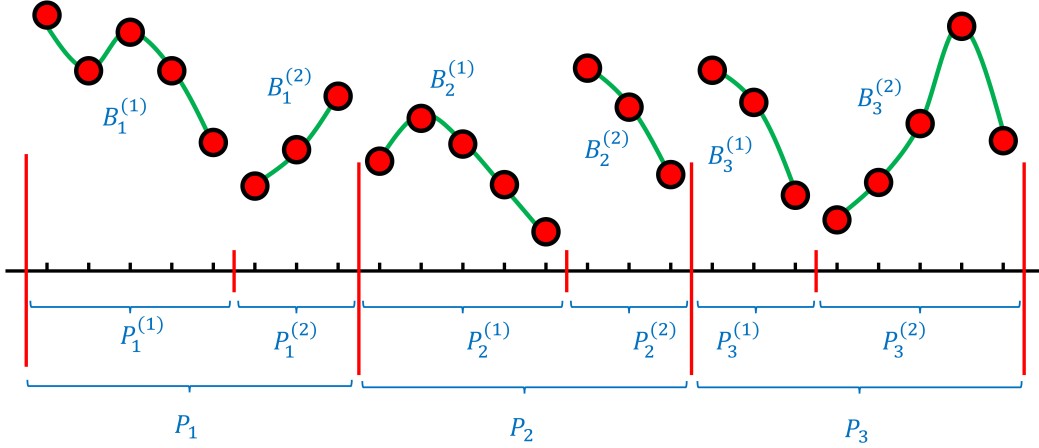

Figure 6: Illustration for the input to Algorithm 3. The set $\{P_1, P_2, P_3\}$ is an equally-sized consecutive partition of $P$ an interval of an $n$-signal, where for every $i \in \{1, 2, 3\}$ the set $P_i$ is projected onto a bicriteria of size 2 which is denoted by $\left\{B_i^{(1)}, B_i^{(2)}\right\}$.

**Overview of Algorithm 4.** The input for the algorithm is the projection of an interval of an $n$-signal onto some rational function. This projection is represented by an $(0, 1)$-approximation; see Definition 3. The algorithm also receives an integer $\lambda \geq 1$ which controls the size of the output. The algorithm computes a (restricted) coreset for the projected input, for which the approximation guarantees hold only for the subset of the set of rational functions that are $2^k$-bounded over the first coordinate of the input interval; see Definition 5 and Lemma 3.

Algorithm 4 computes this restricted coreset as mentioned in section 2.2.

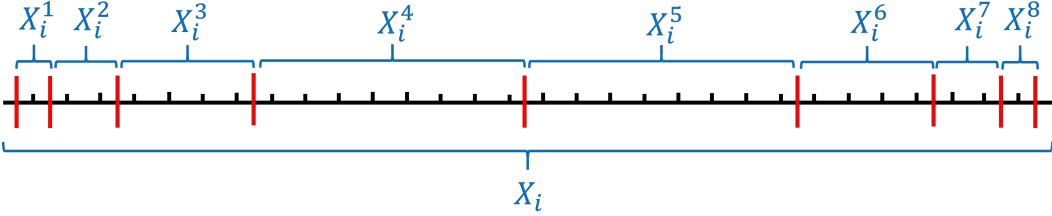

Figure 7: An exponential partition $X_i^1, \cdots, X_i^8$ of a set $X_i = [30]$; see Line 6 of Algorithm 4.

We note that in our code we substitute the precise computation of the partition in Line 3 of Algorithm 4 presented in Lemma 11 by an "approximate partition", where the roots of the polynomials in Lemma 11 are approximated by numeric methods; done using the method `roots` from the library numpy, which to the best of our knowledge utilizes Horn & Johnson (1999).

---

**Algorithm 4:** MINI-REDUCE$(B, \lambda)$; see Lemma 3.

---

**Input** : An interval of an $n$-signal $P$ which is projected unto some $q \in \left(\mathbb{R}^k\right)^2$, i.e., $\ell(P, q) = 0$. This is represent by $B := \{(P, q)\}$, which is a $(0, 1)$-approximation $B$ of $P$; see Definition 3. An integer $\lambda \geq 1$

**Output:** A weighted set $(S, w)$, i.e., $S \subset \mathbb{R}^2$ and $w : S \to \mathbb{R}$; see Section 2.1.

1 $X := \{x \mid (x, y) \in P\}$, i.e., $X$ is the union over the first coordinate of every pair in $P$.

2 $(c, c') := q \in \left(\mathbb{R}^k\right)^2$.

3 Let $\{X_1, \cdots, X_\eta\}$ be a partition of $X$ into $\eta \in O(k)$ sets, such that for every $i \in [\eta]$ the function $f(x) = |1 + x \cdot \mathrm{poly}(c', x)|$ is monotonic over $\left[\min(X_i), \max(X_i)\right)$, and for every $i, j \in [\eta]$, where $i \neq j$ we have $X_i \cap \left[\min(X_j), \max(X_j)\right] = \emptyset$; see Lemma 11.

4 $S := \emptyset$

5 **for** *every* $i \in \{1, \cdots, \eta\}$ **do**

6     Let $\left\{X_i^1, \cdots, X_i^{m_i}\right\}$ be a consecutive partition of $X_i$ into $m_i \in \Theta\left(\log(|X_i|)\right)$ sets such that for every $j \in [m_i]$ we have $|X_i^j| = 2^{\min\{j-1, m_i-j\}}$.
    `// See Figure 7 for illustration.`

7     **for** *every* $j \in [m_i]$ **do**

8        Let $s : X_i^j \to (0, \infty)$ such that $s(x) \geq \sup_c \dfrac{|\mathrm{poly}(c, x)|}{\sum_{x' \in X_i^j} |\mathrm{poly}(c, x')|}$ for every $x \in X_i^j$, and $\sum_{x \in X_i^j} s(x) \in O(k^2)$ where the supremum is over $c \in \mathbb{R}^{2k+1}$ such that $|\mathrm{poly}(c, x)| > 0$; see Corollary 2.

9        Set $s'(x) := \dfrac{s(x)}{\sum_{x' \in X_i^j} s(x')}$ for every $x \in X_i^j$.

10        $P_i^j := \left\{\left(x, \mathrm{ratio}(q, x)\right) \mid x \in X_i^j\right\}$ `// see Definition 2.`

11        Pick a sample $S_i^j$ of $\lambda$ i.i.d. points from $P_i^j$, where each $(x, y) \in P_i^j$ is sampled with probability $s'(x)$.

12        $S := S \cup S_i^j$

13        Set $w(p) := 1/\left(\lambda \cdot s'(x)\right)$ for every $p = (x, y) \in S$.

14 **return** $(S, w)$

---

The following lemma states the desired properties of Algorithm 4; see Lemma 13 for its proof.

**Lemma 3.** *Let $P$ be an interval of a $n$-signal which is projected unto some $q \in \left(\mathbb{R}^k\right)^2$, i.e., $\ell(P, q) = 0$. Let $B := \{(P, q)\}$, which is a $(0, 1)$-approximation $B$ of $P$; see Definition 3. Let $X$ be the first coordinate of $P$, i.e., $X := \{x \mid (x, y) \in P\}$. Put $\epsilon, \delta \in (0, 1/10]$, and let*

$$\lambda \geq \frac{c^*}{\epsilon^2} \cdot (4^{k+1}k^2 + 1)\left(k^2 \log(4^{k+1}k^2 + 1) + \log\left(\frac{k \log n}{\delta}\right)\right),$$

*be an integer, where $c^* > 1$ is a constant that can that can be determined from the proof. Let $(S, w)$ be the weighted set that is returned by a call to MINI-REDUCE$(B, \lambda)$; see Algorithm 4. Then $|S| \in O\left(k\lambda \cdot \log n\right)$ and, with probability at least $1 - \delta$, for every $q' \in \left(\mathbb{R}^k\right)^2$ that is $2^k$-bounded over $X$ (see Definition 5), we have*

$$|\ell(P, q') - \ell\left((S, w), q'\right)| \leq \epsilon \cdot \ell(P, q'). \tag{2}$$

### B.3 COMPUTING AN $(\alpha, \beta)$-APPROXIMATION WITH LARGE $\beta$ ALGORITHM

**Overview of Algorithm 5.** Algorithm 5 takes as input an $n$-signal $P$ and an integer $\beta \geq 1$. It aims to partition $P$ into $\psi \in \Theta(\beta)$ sets $P_1, P_2, \cdots, P_\psi$ of the same size, and to compute, for every such set $P_i$, the query $q_i \in \left(\mathbb{R}^k\right)^2$ that minimizes the RFF loss $\ell(P_i, q)$ for this set. The algorithm outputs the sets $P_i$ in the partition of $P$, each equipped with its optimal query $q_i$.

As the time to compute the optimal query for each set in the partition of $P$ depends polynomially on the size of the set, we need those sets to be small. Unfortunately, this implies that, we must plug

a large value of $\beta$. To this end, this algorithm on its own does not suffice in order to compute the desired $(\alpha, \beta)$-approximation with small values of both $\alpha$ and $\beta$. However, this algorithm is still utilized in Algorithm 1 as some sort of initialization.

---

**Algorithm 5:** BATCH-APPROX$(P, \beta)$; see Lemma 4.

---

**Input** : An $n$-signal $P$ where $n \geq 2k$ is a power of 2, and a positive integer $\beta$.
**Output:** An ordered set $B$ that contains $\psi \in O(\beta)$ $(B_1, \cdots, B_\psi)$, where every $B_i$ is an
$(1, 1)$-approximation of a set $P_i$ in some consecutive partition $\{P_1, \cdots, P_\psi\}$ of $P$; see
Definition 3.

1 Compute an equally-size partition $\{P_1, \cdots, P_\psi\}$, where $\psi \in \left[\lfloor \beta/2 \rfloor, \beta\right]$, of $P$ whose size is
$|P_1| = 2^m$, for some integer $m \geq 1$.
2 For every $i \in [\psi]$, let $q_i$ be the optimal fitting rational function for $P_i$ for every $i \in [\psi]$, i.e.
$q_i \in \arg\min_{q \in (\mathbb{R}^k)^2} \ell(P_i, q)$; see Lemma 8 for an implementation.
3 Set $B := (B_1, \cdots, B_\psi)$, where $B_i := \{(P_i, q_i)\}$, for every $i \in [\psi]$.
4 **return** $B$.

---

**Lemma 4.** *Let $P$ be an $n$-signal, where $n$ is a power of 2. Let $\beta$ be a positive integer. Let $B := (B_1, \cdots, B_\psi)$, where $\psi \in \left[\lfloor \beta/2 \rfloor, \beta\right]$, be the output of a call to BATCH-APPROX$(P, \beta)$; see Algorithm 5. Put $\{(P_i, q_i)\} := B_i$, for every $i \in [\psi]$. Then, $B' := \{(P_1, q_1), \cdots, (P_\psi, q_\psi)\}$ is an $(1, \beta)$-approximation of $P$; see Definition 3. Moreover, the output of the call to BATCH-APPROX$(P, \beta)$ can compute in $n \cdot (2kn/\beta)^{O(k)}$ time.*

*Proof.* By its construction in Algorithm 5 we have that $B'$ is an $(1, \beta)$-approximation of $P$. By Lemma 8, for every $i \in [\psi]$, the computation time of every $q_i$ in Line 2 of Algorithm 5 is in $\left(2k|P_i|\right)^{O(k)}$. Combining this with the construction of Algorithm 5 proves the lemma. $\square$

### B.4 FAST PRACTICAL HEURISTIC

Unfortunately the running time of the algorithms is still large. Therefore, we suggest a heuristic to run on top of our coreset. We later prove that, under some assumptions, this heuristic gives a constant factor approximation. For this heuristic we need the following definition.

**Definition 6.** *Let $S$ be a set of $2k$ points on the plane. We define SOLVER$(S)$ as an arbitrary $(c, c') \in (\mathbb{R}^k)^2$ that satisfies $\ell(S, q) = 0$ if there is such a pair, otherwise it is empty.*

In Lemma 20, we prove, that if $\left|\{x \cdot y \mid (x, y) \in S\}\right| = 2k$, then SOLVER is never empty and that it can be computed in $O(k^3)$ time. In our companion code, we sample $G$ directly from $P$.

---

**Algorithm 6:** FAST-CENTROID-SET$(P, \beta)$; see Lemma 21.

---

**Input** : A finite set $P \subset \mathbb{R}^2$ of at least $2k$ points, where
$\forall S \subset P, |S| = 2k : \left|\{x \cdot y \mid (x, y) \in S\}\right| = 2k$, and an integer $\beta \geq 1$.
**Output:** A set $G \subset (\mathbb{R}^k)^2$ of size $|G| \leq \beta$.

1 $G := \{S \subseteq P \mid |S| = 2k\}$.

2 **if** $|G| \geq \beta$ **then** // $|G| := \binom{|P|}{2k}$

3 $\quad$ **return** $\bigcup_{S \in G}$ SOLVER$(S)$// see Definition 6.

4 Pick a sample $G' \subset G$ of $|G'| = \beta$ sets of points, where each set $S \in G'$ of points is sampled
i.i.d. and uniformly at random from $G$.

5 **return** $\bigcup_{S \in G'}$ SOLVER$(S)$// see Definition 6.

---

## C  ALGORITHM 2: CORESET GIVEN AN $(\alpha, \beta)$-APPROXIMATION

The coreset construction that we use in Algorithm 2 is a non-uniform sample from a distribution, which is known as sensitivity, that is based on the $(\alpha, \beta)$-*approximation* defined in Definition 3. To apply the generic coreset construction we need two ingredients:

(i) A bound on the dimension induced by the query space ("complexity") that corresponds to our problem as formally stated and bounded in subsection C.1. This bound on the dimension induced by the query space determines the required size of the random sample picked in Algorithm 2.

(ii) A bound on the sensitivity as formally stated and bounded in the proof of Lemma 6. This bound on the sensitivity determines the required size of the random sample that is picked in Algorithm 2.

### C.1  BOUND ON THE DIMENSION OF THE QUERY SPACE

We first define the classic notion of VC-dimension, which is used in Theorem 8.14 in Anthony & Bartlett (2009), and is usually related to the PAC-learning theory Li et al. (2001).

**Definition** (VC-dimension Lucic et al. (2017)). *Let $F \subset \{\mathbb{R}^d \to \{0,1\}\}$ and let $X \subset \mathbb{R}^d$. Fix a set $S = \{x_1, \cdots, x_n\} \subset X$ and a function $f \in F$. We call $S_f = \{x_i \in S \mid f(x_i) = 1\}$ the induced subset of $S$ by $f$. A subset $S = \{x_1, \cdots, x_n\}$ of $X$ is shattered by $F$ if $|\{S_f \mid f \in F\}| = 2^n$. The VC-dimension of $F$ is the size of the largest subset of $X$ shattered by $F$.*

**Theorem 2.** *Let $h$ be a function from $\mathbb{R}^m \times \mathbb{R}^d$ to $\{0,1\}$, and let*

$$\mathcal{H} = \{h_\theta : \mathbb{R}^d \to \{0,1\} \mid \theta \in \mathbb{R}^m\}.$$

*Suppose that $h$ can be computed by an algorithm that takes as input the pair $\theta \in \mathbb{R}^m \times \mathbb{R}^d$ and returns $h_\theta(x)$ after no more than $t$ of the following operations:*

- *the arithmetic operations $+, -, \times$, and $/$ on real numbers,*

- *jumps conditioned on $>, \leq, <, \geq, =$, and $\neq$ comparisons of real numbers, and*

- *output 0, 1.*

*Then the VC-dimension of $\mathcal{H}$ is $O(m^2 + mt)$.*

For the sample mentioned in the start of Section C we utilize the following generalization of the previous definition of VC-dimension. This is commonly referred to VC-dimension, but to differentiate this definition from the previous, and to be in line with the notations in Feldman et al. (2019) we abbreviate it to *dimension*. This is the dimension induced by the query space which would be assigned in Theorem 3 to obtain the proof of Algorithm 2.

**Definition** (range space Feldman et al. (2013)). *A range space is a pair $(L, \text{ranges})$ where $L$ is a set, called ground set and $\text{ranges}$ is a family (set) of subsets of $L$.*

**Definition** (dimension of range spaces Feldman et al. (2013)). *The dimension of a range space $(L, \text{ranges})$ is the size $|S|$ of the largest subset $S \subseteq F$ such that*

$$|\{S \cap \text{range} \mid \text{range} \in \text{ranges}\}| = 2^{|S|}.$$

**Definition 7** (range space of functions Feldman et al. (2013); Har-Peled & Sharir (2009); Feldman & Langberg (2011)). *Let $F$ be a finite set of functions from a set $\mathcal{Q}$ to $[0, \infty)$. For every $Q \in \mathcal{Q}$ and $r \geq 0$, let $\text{range}(F, Q, r) = \{f \in F \mid f(Q) \geq r\}$.*
*Let $\text{ranges}(F) = \{\text{range}(F, Q, r) \mid Q \in \mathcal{Q}, r \geq 0\}$.*
*Finally, let $\mathbb{R}_{\mathcal{Q},F} = (F, \text{ranges}(F))$ be the range space induced by $\mathcal{Q}$ and $F$.*

In the following lemma, which is inspired by Theorem 12 in Lucic et al. (2017), we bound the VC-dimension which would be assigned in Theorem 3 to obtain the proof of Algorithm 2.

**Lemma 5.** *Let $B = \{(P_1, q_1), \cdots, (P_\beta, q_\beta)\}$ be an $(\alpha, \beta)$-approximation of some $n$-signal $P = \{(1, y_1), (2, y_2), \cdots, n, y_n)\}$; see Definition 3. Let $f : P \times (\mathbb{R}^k)^2 \to [0, \infty)$, be the function that*

*maps every* $p = (x, y) \in P$, *where* $p \in P_i$, *and any* $q \in (\mathbb{R}^k)^2$ *to* $f(x) = D(q, \text{ratio}(q_i, x))$. *For every* $i \in [n]$ *let* $f_i : (\mathbb{R}^k)^2 \to [0, \infty)$ *denote the function that maps every* $q \in (\mathbb{R}^k)^2$ *to* $f_i(q) = f(q, (i, y_i))$. *Let* $F = \{f_1, \ldots, f_n\}$. *The dimension of the range space* $\mathbb{R}_{(\mathbb{R}^k)^2, F}$ *that is induced by* $(\mathbb{R}^k)^2$ *and* $F$ *is in* $O(k^2)$.

*Proof.* For every $(q, r) = ((c, c'), r) \in (\mathbb{R}^k)^2 \times \mathbb{R}$, let $h_{(c|c'|r)} : \mathbb{R} \to \{0, 1\}$ that maps every $x \in [n]$ to $h_{(c|c'|r)}(x) = 1$ if and only if $f_i(q) \leq r$, and every $x \in \mathbb{R} \setminus [n]$ to $h_{(c|c'|r)}(x) = 0$. Let $\mathcal{H} = \{h_\theta \mid \theta \in \mathbb{R}^{2k+1}\}$. For every $c \in \mathbb{R}^k$ and any $x \in \mathbb{R}$ we can compute $\text{poly}(c, x)$ with $O(k)$ arithmetic operations on real numbers and jumps conditioned on comparisons of real numbers; see, for example, Horner's scheme Neumaier (2001), which is used in numpy's implementation of the method polyval Harris et al. (2020). Therefore, for every $i \in [n]$ and any $\theta \in \mathbb{R}^{2k+1}$, by the definition of $D$, we can calculate $h_\theta(i)$ with $O(k)$ arithmetic operations on real numbers and jumps conditioned on comparisons of real numbers. Hence, substituting $d := n$, $m := 2k + 1$, $h := h$, $\mathcal{H} := \mathcal{H}$, and $t \in O(k)$ in Theorem 2 yields that the VC-dimension of $\mathcal{H}$ is in $O(k^2)$. Hence, by the construction of $\mathcal{H}$ and the definition of range spaces in Definition 7, we have that the dimension of the range space $\mathbb{R}_{P,F}$ that is induced by $P$ and $F$ is in $O(k^2)$. $\square$

## C.2 SENSITIVITY OF FUNCTIONS

For the self containment of the work we state previous work on sensitivity of functions.

Observe that the following is stated in a more general form than required in this section. This is since we would re-use the stated results in later parts for the restricted coreset, while in this section we bound the sensitivity to the projection unto a bicretiria; see Section 4 and Definition 3.

**Definition 8** (query space Feldman et al. (2019)). *Let* $P \subset \mathbb{R}^2$ *be a finite non empty set. Let* $f : P \times (\mathbb{R}^k)^2 \to [0, \infty)$ *and* $\text{loss} : \mathbb{R}^{|P|} \to [0, \infty)$ *be a function. The tuple* $(P, (\mathbb{R}^k)^2, f, \text{loss})$ *is called a* query space. *For every* $q \in (\mathbb{R}^k)^2$ *we define the overall fitting error of* $P$ *to* $q$ *by*

$$f_{\text{loss}}(P, q) := \text{loss}\left((f(p, q))_{p \in P}\right) = \text{loss}\left(f(p_1, q), \ldots, f(p_{|P|}, q)\right).$$

To emphasize that the following coreset is a subset of the input set, in contrast to $\epsilon$-coreset as in Definition 4, we call it subset-$\epsilon$-coreset. In Section C.4 we prove that there is no such coreset for the RFF problem; see Definition 2.

**Definition 9** (subset-$\epsilon$-coreset Feldman et al. (2019)). *Let* $(P, (\mathbb{R}^k)^2, f, \text{loss})$ *be a query space as in Definition 8. For an approximation error* $\epsilon > 0$, *the pair* $S' = (S, u)$ *is called an* subset-$\epsilon$-coreset *for the query space* $(P, (\mathbb{R}^k)^2, f, \text{loss})$, *if* $S \subseteq P, u : S \to [0, \infty)$, *and for every* $q \in (\mathbb{R}^k)^2$ *we have*

$$(1 - \epsilon) f_{\text{loss}}(P, q) \leq f_{\text{loss}}(S', q) \leq (1 + \epsilon) f_{\text{loss}}(P, q).$$

**Definition 10** (sensitivity of functions). *Let* $P \subset \mathbb{R}^2$ *be a finite and non empty set, and let* $F \subset \{P \to [0, \infty]\}$ *be a possibly infinite set of functions. The* sensitivity *of every point* $p \in P$ *is*

$$S^*_{(P,F)}(p) = \sup_{f \in F} \frac{f(p)}{\sum_{p \in P} f(p)}, \tag{3}$$

*where* sup *is over every* $f \in F$ *such that the denominator is positive. The* total sensitivity *given a sensitivity is defined to be the sum over these sensitivities,* $S^*_F(P) = \sum_{p \in P} S^*_{(P,F)}(p)$. *The function* $S_{(P,F)} : P \to [0, \infty)$ *is a* sensitivity bound *for* $S^*_{(P,F)}$, *if for every* $p \in P$ *we have* $S_{(P,F)}(p) \geq S^*_{(P,F)}(p)$. *The* total sensitivity bound *is then defined to be* $S_{(P,F)}(P) = \sum_{p \in P} S_{(P,F)}(p)$.

The following theorem proves that a coreset can be computed by sampling according to sensitivity of functions. The size of the coreset depends on the total sensitivity and the complexity (VC-dimension) of the query space, as well as the desired error $\epsilon$ and probability $\delta$ of failure.

**Theorem 3** (coreset construction Feldman et al. (2019)). *Let*

- $P = \{p_1, \cdots, p_n\} \subset \mathbb{R}^2$ *be a finite and non empty set, and* $f : P \times \left(\mathbb{R}^k\right)^2 \to [0, \infty)$.

- $F = \{f_1, \ldots, f_n\}$, *where* $f_i(q) = f(p_i, q)$ *for every* $i \in [n]$ *and* $q \in \left(\mathbb{R}^k\right)^2$.

- $d'$ *be the dimension of the range space that is induced by* $\left(\mathbb{R}^k\right)^2$ *and* $F$.

- $s^* : P \to [0, \infty)$ *such that* $s^*(p)$ *is the sensitivity of every* $p \in P$, *after substituting* $P = P$ *and* $F = \left\{ f' : P \to [0, \infty] \mid \forall p \in P, q \in \left(\mathbb{R}^k\right)^2 : f'(p) := f(p, q) \right\}$ *in Definition 10, and* $s : P \to [0, \infty)$ *be the sensitivity bound of* $s^*$.

- $t = \sum_{p \in P} s(p)$.

- $\epsilon, \delta \in (0, 1)$.

- $c > 0$ *be a universal constant that can be determined from the proof.*

- $\lambda \geq c(t + 1)\left(d' \log(t + 1) + \log(1/\delta)\right)/\epsilon^2$.

- $w : P \to \{1\}$, *i.e. a function such that for every* $p \in P$ *we have* $w(p) = 1$.

- $(S, u)$ *be the output of a call to* CORESET-FRAMEWORK$(P, w, s, \lambda)$ *(Algorithm 1 in Feldman et al. (2019)).*

*Then the following holds*

- *With probability at least* $1 - \delta$, $(S, w)$ *is an subset-$\epsilon$-coreset of size* $|S| \leq \lambda$ *for the query space* $\left(F, \left(\mathbb{R}^k\right)^2, f, \|\cdot\|_1\right)$; *see Definition 9.*

### C.3 ANALYSIS OF ALGORITHM 2: SAMPLE-CORESET

In the following lemma we prove Lemma 1, which proves that, given values that satisfy specific properties, we have that Algorithm 2 yields an $\epsilon$-coreset; see Definition 4.

**Lemma 6.** *Let* $B := \left\{(P_1, q_1), \cdots, (P_\beta, q_\beta)\right\}$ *be an* $(\alpha, \beta)$-*approximation of some n-signal* $P$, *for some* $\alpha > 0$; *see Definition 3. Put* $\epsilon, \delta \in (0, 1/10]$, *and let*

$$\lambda := \left\lceil \frac{c^*}{\epsilon^2}(\alpha + 1)\left(k^2 \log(\alpha + 1) + \log(1/\delta)\right) \right\rceil,$$

*where* $c^* \geq 1$ *is a constant that can be determined from the proof. Let* $(B, C, w)$ *be the output of a call to* SAMPLE-CORESET$(B, \lambda)$; *see Algorithm 2. Then, Claims (i)–(ii) hold as follows:*

(i) $(B, C, w)$ *can be stored using* $O(\lambda + \beta k)$ *memory.*

(ii) *With probability at least* $1 - \delta$, *we have that* $(B, C, w)$ *is an* $\epsilon$-*coreset of* $P$; *see Definition 4.*

*Proof.* We have (i) by the construction of Algorithm 2 and the definitions in the theorem. Let $c$ as computed in the call to SAMPLE-CORESET$(B, \lambda)$; see Algorithm 2. Since $B$ is an $(\alpha, \beta)$-approximation of $P$ we have that $c \neq \infty$. If $c = 0$, then the theorem holds by the construction of Algorithm 2. Hence, we assume this is not the case. Let $P'$ be the projection of $P$ unto $B$, i.e., $P' := \left\{\left(x, \text{ratio}(q_i, x)\right) \mid i \in [\beta], (x, y) \in P\right\}$; see Definition 3. Let $\mathcal{Q}$ be the union of $q = (c, c') \in \left(\mathbb{R}^k\right)^2$ such that $1 + \text{poly}(c', x) \neq 0$ for every $(x, y) \in P$. Let $q = (c, c') \in \mathcal{Q}$. We have

$$\left| \sum_{p \in P} D(q, p) - \sum_{p \in C} \left(w(p) \cdot D(q, p)\right) - \sum_{p \in P'} D(q, p) \right| =$$

$$\sum_{p \in P} D(q, p) \cdot \left| \frac{\sum_{p \in P} D(q, p) - \sum_{p \in P'} D(q, p)}{\sum_{p \in P} D(q, p)} - \frac{\sum_{p \in C} \left(w(p) \cdot D(q, p)\right)}{\sum_{p \in P} D(q, p)} \right|, \quad (4)$$

where the previous equality is by taking $\sum_{p \in P} D(q, p)$ out of the sum. Let $s : P \to [0, \infty)$ as defined in Line 8 of Algorithm 2 in the call to SAMPLE-CORESET$(B, \lambda)$, i.e., for every $i \in [\beta]$ and any $p \in P_i$ we have

$$s(p) = \frac{D(q_i, p)}{\sum_{i=1}^{\beta} \ell(P_i, q_i)}. \tag{5}$$

Let $i \in [\beta]$ and $s_i^* : P_i \to [0, \infty)$ such that for every point $p = (x, y) \in P_i$ we have

$$s_i^*(p) = \frac{\left| D(q, p) - \left| f_{q_i}(x) - f_q(x) \right| \right|}{\sum_{p \in P} D(q, p)}, \tag{6}$$

which, due to the definition of $P'$ as projection of $P$ unto $B$, is an upper bound on the contribution of every point in $P_i$ to the sum in Equation 4. Let $p = (x, y) \in P_i$, so that

$$s_i^*(p) = \frac{\left| D(q, p) - \left| f_{q_i}(x) - f_q(x) \right| \right|}{\sum_{p \in P} D(q, p)} =$$

$$\frac{\left| \left| y - f_q(x) \right| - \left| f_{q_i}(x) - f_q(x) \right| \right|}{\sum_{p \in P} D(q, p)} \leq \frac{\left| y - f_{q_i}(x) \right|}{\sum_{p \in P} D(q, p)} = \frac{D(q_i, p)}{\sum_{p \in P} D(q, p)} \leq \alpha \cdot s(p), \tag{7}$$

where the first equality is by the definition of $s^*$ from Equation 6, the second equality is by the definition of $D$, the inequality is by the reverse triangle inequality, the third equality is by the definition of $D$, and the last equality is by Equation 5 and the definition of $B$ as an $(\alpha, \beta)$-approximation of $P$.

Let $\tilde{s} : P \to [0, \infty)$ such that for every $p \in P$ we have $\tilde{s}(p) = \alpha \cdot s(p)$. By Equation 7, for every $i \in [\beta]$, we have that $\tilde{s}$ is a sensitivity bound for $s_i^*$.

For every $p = (x, y) \in P_i, i \in [\beta]$ and any $q \in \left( \mathbb{R}^k \right)^2$ let $f(q, p) = D\big(q, \text{ratio}(q_i, x)\big)$. Let $F = \{f_1, \ldots, f_n\}$, where $f_i(y) = f(q, p)$ for every $p \in P_i, i \in [\beta]$ and $q \in \left( \mathbb{R}^k \right)^2$. Let $k^* \in O(k^2)$ be the dimension of the range space $\mathbb{R}_{P, F}$ from Lemma 5 when assigning $P$ and $B$.

Substituting $\epsilon := \epsilon, \delta := \delta, \lambda := \lambda$, the query space $(P, \mathcal{Q}, F, \| \cdot \|_1), d' \in O\left( k^2 \right)$ the VC-dimension induced by $\left( \mathbb{R}^k \right)^2$ and $F$ from Lemma 5, the sensitivity bound $\tilde{s}$, and the total sensitivity $t = \alpha \sum_{p \in P} s(p) = \alpha$ in Theorem 3, combined with the construction of Algorithm 2, yields that with probability at least $1 - \delta$, for every $q \in \mathcal{Q}$, we have

$$\left| \frac{\sum_{p \in P} D(q, p) - \sum_{p \in P'} D(q, p)}{\sum_{p \in P} D(q, p)} - \frac{\sum_{p \in C} \big( w(p) \cdot D(q, p) \big)}{\sum_{p \in P} D(q, p)} \right| \leq \epsilon. \tag{8}$$

Combining Equation 8 and Equation 4 proves the theorem. $\qquad \square$

## C.4 LOWER BOUND.

In this section we prove that there is no subset-$\epsilon$-coreset for the query space $\left( P, \left( \mathbb{R}^k \right)^2, D, \| \cdot \|_1 \right)$, where $P \subset \mathbb{R}^2$ is an $n$-signal $P$ and $D$ is as Definition 2, as defined in Definition 8. This justifies our Definition 4 of coreset for RFF. The main idea in the following claim is illustrated by figure 8.

**Claim 1** (Minor modification of Claim 5 in Rosman et al. (2014)). *For every integer $n \geq 2$ there is an $n$-signal $P$ such that the following holds. For every $C \subseteq \mathbb{R}^2$, where $|C| < n$, there is $q \in \left( \mathbb{R}^k \right)^2$ such that $\sum_{p \in C} D(q, p) \in [0, \infty)$ and $\sum_{p \in P} D(q, p) = \infty$.*

*Proof.* Let $P = \{(1,0), \cdots, (n,0)\}$ and $C \subseteq \mathbb{R}^2$, where $|C| < n$. Put $(a,0) \in P, a > 0$ such that $C \cap \{(a,y) \mid y \in \mathbb{R}\} = \emptyset$; i.e., there is no point in $C$ with $x$-value equal $a$. There is such a point since $|C| < n = |P|$. Let $c = (1, 0, 0, \cdots, 0), c' = \left(-\dfrac{1}{a}, 0, 0, \cdots, 0\right)$, and $q = (c, c') \in \left(\mathbb{R}^k\right)^2$. Since $D\big((a,0), q\big) = \big|\mathrm{ratio}(q,a) - 0\big| = \infty$ (observe that $1 + a\mathrm{poly}(c', a) = 1 - a/a = 0$), and $(a,0) \in P$, we obtain $\displaystyle\sum_{p \in P} D(q,p) \geq D\big((a,0), q\big) = \infty$. On the other hand $1 - \dfrac{1}{a} \cdot x = 0$ only for $x = a$, therefore $\forall p \in C : D(q,p) \in [0, \infty)$, and since $C$ is a finite set we obtain $\displaystyle\sum_{p \in C} D(q,p) \in [0, \infty)$. $\qquad\square$

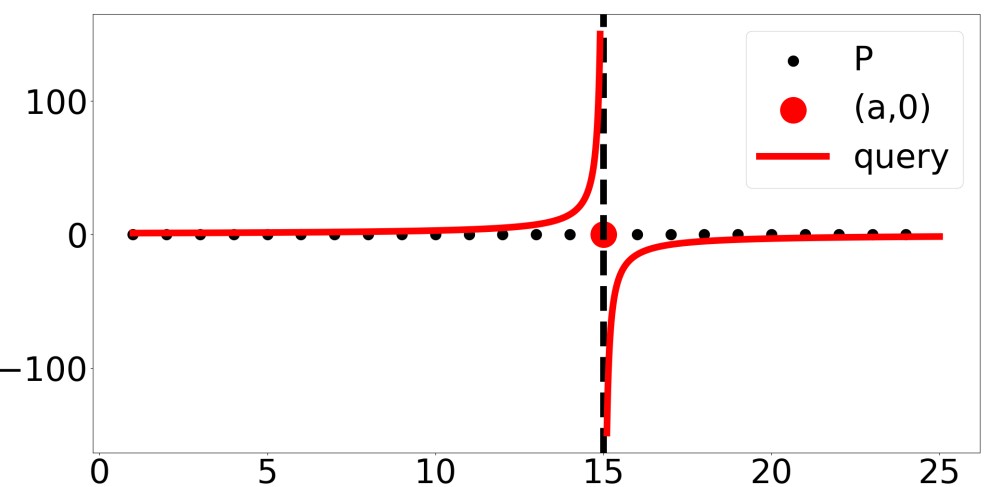

Figure 8: Visual illustration of the main idea behind Claim 1. The $n$-signal $P = \{(1,0), \cdots, (25,0)\}$ as in Claim 1 for $n = 25$ (black dots). Consider a subset $C$ that contains all these points except a single one, say, the red point $(15, 0)$. We can always find a rational function (query) whose sum of distances is close to zero for $P$, but close to $\infty$ for $C$, due to due its high change at the point $(15, 0)$ that was not selected to $C$.

## D INEFFICIENT SOLVER FOR THE RFF PROBLEM

In the following section we will prove that we can solve the RFF problem from Definition 2 in $(2kn)^{O(k)}$ time as previously mentioned in Section 2.1.

The main idea is that, given an assignment of whether each point is below or above the best fitting rational function (to the loss in Equation 1 from Definitions 2) the problem can be written as a fractional polynomial programming problem and solved in polynomial (in the input size) time; see Pizzo et al. (2018). Using previous work Marom & Feldman (2019) we can bound the number of candidate assignments mentioned above to be polynomial in the input size. Hence, by constructing a semi tree we can find all the satisfiable assignments in time polynomial in the input size, which enables us to compute an optimal solution in polynomial time.

In the following section, for every $x \in \mathbb{R}$ let $\mathrm{sign}(x) = 1$ if $x > 0$ and $\mathrm{sign}(x) = -1$ otherwise.

**Theorem 4** (Theorem 23 in Marom & Feldman (2019)). *Let $f_1, \cdots, f_m$ be real polynomials in $k < m$ variables, each of degree at most $b \geq 1$. Then the number of sign sequences $(\mathrm{sign}(f_1(x)), \cdots, \mathrm{sign}(f_m(x)))$ over $x \in \mathbb{R}^d$ that consist of the terms $1, -1$ is at most $\left(\dfrac{4ebm}{k}\right)^k$.*

To utilize this previous bound we state the following observation.

**Observation 1.** *Let $P = \{(x_1, y_1), \cdots, (x_n, y_n)\} \subset \mathbb{R}^2$. Let $\mathcal{Q}$ be the union of $(c, c') \in (\mathbb{R}^k)^2$ such that $1 + x \cdot \text{poly}(c', x) \neq 0$ for every $(x, y) \in P$. There are $g_1, \cdots, g_n : \mathbb{R}^{2k} \to \mathbb{R}$ polynomial of degree in $O(n)$ with $2k$ variables such that for every $(c, c') \in \mathcal{Q}$ we have*

$$g_i(c \mid c') = \big(1 + x \cdot \text{poly}(c', x)\big) \cdot \big(\text{poly}(c, x) - y - y \cdot x \cdot \text{poly}(c', x)\big).$$

*Proof.* Let $q := (c, c') \in \mathcal{Q}$. For every $(x, y) \in P$, by reorganizing the expression we have

$$\text{sign}\big(\text{ratio}((c, c'), x) - y\big) = \text{sign}\Big(\big(1 + x \cdot \text{poly}(c', x)\big) \cdot \big(\text{poly}(c, x) - y - y \cdot x \cdot \text{poly}(c', x)\big)\Big). \quad (9)$$

For every $i \in [n]$, let $g_i : \mathbb{R}^{2k} \to \mathbb{R}$ be a polynomial of degree in $O(n)$ with $2k$ variables that maps every $(c, c') \in (\mathbb{R}^k)^2$ to

$$g_i(c \mid c') = \big(1 + x \cdot \text{poly}(c', x)\big) \cdot \big(\text{poly}(c, x) - y - y \cdot x \cdot \text{poly}(c', x)\big).$$

By Equation 9 we have that $g_1, \cdots, g_n : \mathbb{R}^{2k} \to \mathbb{R}$ satisfy the observation. $\qquad\square$

Using this observation and Theorem 4 we obtain the following result.

**Corollary 1.** *Let $P = \{(x_1, y_1), \cdots, (x_n, y_n)\} \subset \mathbb{R}^2$. Let $\mathcal{Q}$ be the union of $(c, c') \in (\mathbb{R}^k)^2$ such that $1 + x \cdot \text{poly}(c', x) \neq 0$ for every $(x, y) \in P$. The number of sign sequences $\Big( \text{sign}\big(\text{ratio}(q, x_1) - y_1\big), \cdots, \text{sign}\big(\text{ratio}(q, x_n) - y_n\big)\Big)$ over every $q \in \mathcal{Q}$ is in $(2kn)^{O(k)}$.*

*Proof.* By Observation 1, let $g_1, \cdots, g_n : \mathbb{R}^{2k} \to \mathbb{R}$ be real polynomials of degree in $O(n)$ with $2k$ variables such that for every $q := (c, c') \in \mathcal{Q}$ we have

$$\text{sign}\big(\text{ratio}(q, x_1) - y_i\big) = \text{sign}\big(g_i(c \mid c')\big). \quad (10)$$

By Claim 4, the number of sign sequences $(\text{sign}(g_1(x)), \cdots, \text{sign}(g_n(x))$ over $x \in \mathbb{R}^{2k}$ that consist of the terms $1, -1$ is in $(2kn)^{O(k)}$. Combining this with Equation 10 proves the Corollary. $\qquad\square$

While, as stated before, the number of possible positions of every points is indeed polynomial, there is the problem of computing this set in polynomial time. We will solve this by utilizing previous work on polynomial programming mentioned in Pizzo et al. (2018).

**Observation 2.** *Let $P = \{(x_1, y_1), \cdots, (x_n, y_n)\} \subset \mathbb{R}^2$. Let $\mathcal{Q} \subset (\mathbb{R}^k)^2$ be the union of every $(c, c') \in (\mathbb{R}^k)^2$ such that $1 + x \cdot \text{poly}(c', x) \neq 0$ for every $(x, y) \in P$. For every vector $S \in \{0, 1\}^n$ we can check in $(2kn)^{O(k)}$ the existence of $q \in \mathcal{Q}$ that satisfies $S = \Big( \text{sign}\big(\text{ratio}(q, x_1) - y_1\big), \cdots, \text{sign}\big(\text{ratio}(q, x_n) - y_n\big)\Big)$, i.e., satisfies the assignment by $S$.*

*Proof.* By Observation 1, let $g_1, \cdots, g_n : \mathbb{R}^{2k} \to \mathbb{R}$ be real polynomials of degree in $O(n)$ with $2k$ variables such that for every $q := (c, c') \in \mathcal{Q}$ we have

$$\text{sign}\big(\text{ratio}(q, x_1) - y_i\big) = \text{sign}\big(g_i(c \mid c')\big).$$

Hence, for every assignment $S \in \{0, 1\}^n$, there is $q \in Q$ such that

$$S = \Big( \text{sign}\big(\text{ratio}(q, x_1) - y_1\big), \cdots, \text{sign}\big(\text{ratio}(q, x_n) - y_n\big)\Big)$$

if and only if there is $x \in \mathbb{R}^d$ such that $S = \Big(g_1(x), \cdots, g_n(x)\Big)$, and the later can be written as polynomial programming and thus solved numerically in $(2kn)^{O(k)}$ time; since this is equality can be written as sum of squares (SOS), see Pizzo et al. (2018). $\qquad\square$

Using this, in the following lemma, we prove that we can generate all the satisfiable options for the function to be above the points or below the points in polynomial time and not exponential.

**Lemma 7.** *Let $P = \{(x_1, y_1), \cdots, (x_n, y_n)\} \subset \mathbb{R}^2$. For every non empty $C \subset P$ let $Q(C) \subset \left(\mathbb{R}^k\right)^2$ be the union of every $(c, c') \in \left(\mathbb{R}^k\right)^2$ such that $1 + x \cdot \mathrm{poly}(c', x) \neq 0$ for any $(x, y) \in C$. All the sign sequences $\left(\mathrm{sign}\left(\mathrm{ratio}(q, x_1) - y_1\right), \cdots, \mathrm{sign}\left(\mathrm{ratio}(q, x_n) - y_n\right)\right)$ over every $q \in Q(P)$ can be computed in $(2kn)^{O(k)}$ time.*

*Proof.* By Corollary 1, for every $C \subset P, |C| = m \geq 2k$ there are at most $m^{O(k)}$ options for the sing sequence $\left(\mathrm{sign}\left(\mathrm{ratio}(q, x) - y\right) \mid (x, y) \in C\right)$ over every $q \in Q(C)$.

Let $C_1, C_2 \subset P, C_1 \cap C_2 = \emptyset$ s.t. $|C_1|, |C_2| \geq 1$, where we know all the satisfiable sign sequences $\left(\mathrm{sign}\left(\mathrm{ratio}(q, x)\right) - y\right) \mid (x, y) \in C\right)$ over $q \in Q(C)$ for $C \in \{C_1, C_2\}$. Since $|C_1|, |C_2| \leq n$, by Corollary 1, the size of $\left\{\left(\mathrm{sign}\left(\mathrm{ratio}(q, x)\right) - y\right) \mid (x, y) \in C\right) \mid q \in Q(C)\right\}$ is in $(2kn)^{O(k)}$ for $C \in \{C_1, C_2\}$. Hence, the size of the candidate sets for

$$\left\{\left(\mathrm{sign}\left(\mathrm{ratio}(q, x)\right) - y\right) \mid (x, y) \in C_1 \cup C_2\right) \mid q \in Q(C_1 \cup C_2)\right\}$$

is in $(2kn)^{O(k)}$. Therefore, by utilizing Corollary 2 to validate each candidate, we can compute in $(2kn)^{O(k)}$ the set $\left\{\left(\mathrm{sign}\left(\mathrm{ratio}(q, x)\right) - y\right) \mid (x, y) \in C_1 \cup C_2\right) \mid q \in Q(C_1 \cup C_2)\right\}$.

Thus, partitioning $P$ to sets of size $O(k) \geq 2k$, for each such $C \subset P$ using Corollary 2 to compute in $(2k)^{O(k)}$ the set $\left\{\left(\mathrm{sign}\left(\mathrm{ratio}(q, x)\right) - y\right) \mid (x, y) \in C\right) \mid q \in Q(C)\right\}$, and combining all the options as stated above proves the lemma. $\qquad\square$

Using this observation, and returning to the original problem, we obtain the following solver.

**Lemma 8.** *For every weighed set $(S, w)$ that contains $n = |S|$ points a pair $q$ in $\arg\min\limits_{q \in (\mathbb{R}^k)^2} \ell\left((S, w), q\right)$ can be computed in $(2kn)^{O(k)}$ time; see Definition 2.*

*Proof.* Let $\mathcal{Q} \subset \left(\mathbb{R}^k\right)^2$ be the union of all the pairs $(c, c') \in \left(\mathbb{R}^k\right)^2$ such that $1 + x \cdot \mathrm{poly}(c', x) \neq 0$ for every $(x, y) \in S$. By Lemma 7, in $(2kn)^{O(k)}$ time, compute all the $n^{O(k)}$ possible sign sequences $\left(\mathrm{sign}\left(\mathrm{ratio}(q', x_1) - y_1\right), \cdots, \mathrm{sign}\left(\mathrm{ratio}(q', x_n) - y_n\right)\right)$ over every $q' \in \left(\mathbb{R}^k\right)^2$.

For every such sign sequences $w'$, where $w'(p) \in \{-1, 1\}$ is the sign of every point $p \in P$, observe the following:

For every $q := (c, c') \in \mathcal{Q}$ satisfying the assignment by $w'$ we have

$$\ell\left((S, w), q\right) = \sum_{p = (x, y) \in S} w(p) w'(p) \cdot \left(\frac{\mathrm{poly}(c, x}{1 + x \cdot \mathrm{poly}(c', x)} - y\right).$$

Hence, by taking the common denominator in the right part of the equation above, there are polynomials $f, g : \left(\mathbb{R}^k\right)^2 \to \mathbb{R}$ of degree in $(2kn)^{O(1)}$ with $\left(\mathbb{R}^k\right)^2$ as the variables and $(2kn)^{O(1)}$ parameters, such that for every $q \in \mathcal{Q}$ we have

$$\ell\left((S, w), q\right) = \frac{f_i(q)}{g(q)}.$$

Hence, utilizing Pizzo et al. (2018), this problem can be solved in $(2kn)^{O(k)}$ time; the solution is a numerical solution that can be approximated to arbitrary precision. Hence, by taking the minimum over the $(2kn)^{O(k)}$ candidate solutions, we can compute $q$ as defined in the lemma in $(2kn)^{O(k)}$ time. $\qquad\square$

## E    CORESET UNDER CONSTRAINTS; ANALYSIS OF ALGORITHM 4

In this section we will prove that Algorithm 4 constructs a restricted coreset for rational functions, which, as previously mentioned in Section 2.2, will be utilized to efficiently compute an $(\alpha, \beta)$-approximation to a given $n$-signal $P$, where $\alpha, \beta \in O(\log(n))$. It should be emphasized that the final coreset construction has no such restrictions or assumptions for either its input or queries.

For readability we split the proof into three parts:

(i) Mostly citing previous work, we bound the polynomial-fitting sensitivity.

(ii) Utilizing the previous bound we compute a sensitivity for a restricted case of the RFF fitting problem from Equation 1 of Definition 2 that is formally sated and bounded in Lemma 10.

(iii) We utilize the previous bound in Lemma 13, which proves the previously stated Lemma 3 that summarises the desired properties of Algorithm 4.

### E.1    UPPER BOUND ON THE POLYNOMIAL-FITTING SENSITIVITY

For the polynomial-fitting sensitivity, consider the following lemma that follows from the work on sensitivity of near-convex functions by Murad Tukan Tukan et al. (2020)

**Lemma 9** (Lemma 35 in Tukan et al. (2020)). *Let $Y$ be a set of $n$ points in $\mathbb{R}^d$. A function $s' : Y \rightarrow [0, \infty)$ can be computed in $O\left(n \cdot d^2\right)$ time such that for every $x \in Y$ we have*

$$\sup_{q \in \mathbb{R}^d} \frac{|x^T \cdot q|}{\sum_{y \in Y} |y^T \cdot q|}| \leq s'(x), \text{ where the supremum is over } q \in \mathbb{R}^d \text{ such that } |x^T \cdot q| > 0, \text{ and}$$

$$\sum_{x \in Y} s'(x) \in O\left(d^{3/2}\right).$$

Using this lemma we obtain the following corollary.

**Corollary 2.** *Let $X$ be a set of $n \geq 1$ reals, and $k \geq 1$ be an integer. A function $s : X \rightarrow [0, \infty)$ can be computed in $O\left(n \cdot k^2\right)$ time such that for every $x \in X$ we have $\sup_{c \in \mathbb{R}^k} \dfrac{\left|\mathrm{poly}(c, x)\right|}{\sum_{y \in X} \left|\mathrm{poly}(c, y)\right|} \leq s(x)$, where the supremum is over $q \in \mathbb{R}^d$ such that $\left|\mathrm{poly}(c, x)\right| > 0$, and $\sum_{x \in X} s(x) \in O\left(k^{3/2}\right)$.*

*Proof.* Let $f : X \rightarrow \mathbb{R}^k$ be the function that maps every $x \in X$ to $(1, x, \cdots, x^{k-1})^T$. Let $Y := \{f(x) \mid x \in X\}$ denote the image of $f$, and let $s' : Y \rightarrow [0, \infty)$ as defined in Lemma 9. For every $x \in X, c \in \mathbb{R}^k$, where $\left|\mathrm{poly}(c, x)\right| > 0$, we have

$$\frac{\left|\mathrm{poly}(c, x)\right|}{\sum_{y \in X} \left|\mathrm{poly}(c, y)\right|} = \frac{\left|f(x)^T \cdot c\right|}{\sum_{y \in X} \left|f(y)^T \cdot c\right|} \leq s'\left(f(x)\right),$$

where the first inequality is by the definition of $\mathrm{poly}$, and the second inequality is by the definition of $s'$; see Lemma 9. Let $s : X \rightarrow [0, \infty)$ be the function that maps every $y \in X$ to $s(y) := s'\left(f(y)\right)$. By Lemma 9, $s$ satisfies all the claims in the corollary.

**Computation time of $s$.**    For every $x \in X$ we can compute $f(x)$ in $O\left(k^2\right)$ time. Hence, the computation time of $Y$ is in $O\left(nk^2\right)$. By Lemma 9, the computation time of $s'$ is in $O\left(n \cdot k^2\right)$. Therefore, since, for every $y \in X$ we defined $s(y) = s'\left(f(y)\right)$, we have that the computation time of $s$ is in $O\left(n \cdot k^2\right)$. $\square$

### E.2    UPPER BOUND ON THE RFF SENSITIVITY

In this section we the RFF sensitivity for the restricted case mentioned in property (ii) at the beginning of Section E. For this we we define the following.

**Definition 11** (Lipschitz-function). *Let $r > 0$, and $a, b \in \mathbb{R}$, where $a > b$. A function $f : [a, b] \to [0, \infty)$ is a $r$-Lipschitz if $f$ is non-decreasing over $[a, b]$, and for every $c \geq 1$, and any $x \in [a, b]$ we have $f(c \cdot x) \leq c^r \cdot f(x)$.*

In the following claim we state a known property of polynomial functions.

**Claim 2.** *Let $k \geq 1$ be an integer, and $a, b \in \mathbb{R}$ where $a > b$. Let $f : [a, b] \to (0, \infty)$ be a positive and non-decreasing polynomial over $[a, b]$ of degree at most $k$. Then $f$ is a $k$-Lipschitz function.*

Using the claim above, we obtain the following corollary.

**Corollary 3.** *Let $c' \in \mathbb{R}^k$. Let $G \subseteq \mathbb{R}$ denote the extrema of the function $g$ that maps every $x \in \mathbb{R}$ to $g(x) = |1 + x \cdot \text{poly}(c', x)|$. Let $X = [a, b] \subset \mathbb{R}, |X| > 1$ such that for every $x \in X$ we have*

$$\min_{\gamma \in G} |x - \gamma| \geq \max(X) - \min(X).$$

*Then the function $f : X \to \mathbb{R}$ that maps every $x \in X$ to $f(x) = \dfrac{1}{1 + x \cdot \text{poly}(c', x)}$ is well defined,*

*and satisfies $\dfrac{\max\limits_{x \in X} |f(x)|}{\min\limits_{x \in X} |f(x)|} \leq 2^k$.*

*Proof.* By the definition of $X$, if there is $x \in X$ such that $x \in G$, then by substituting $0 = \min\limits_{\gamma \in G} |x - \gamma| \geq \max(X) - \min(X)$ we have that $\max(X) = \min(X)$ and as such $|X| = 1$, which contradicts the definition of $X$ in the claim. Therefore, we have $X \cap G = \emptyset$, that is $g$ has no extrema over $X$. If there is $x \in X$ such that $g(x) = 0$, by the definition of $g$ we have that $x$ is an extrema of the function $g$. Hence, there is no $x \in X$ such that $g(x) = 0$, which yields that the function $f$ defined in the claim is well defined.

Since $g$ has no extrema over $X$ we can prove that $\dfrac{\max\limits_{x \in X} |f(x)|}{\min\limits_{x \in X} |f(x)|} \leq 2^k$ via the following

**Case (i): $g$ is constant in the range of $X$.**
From the definition of the case and the definition of $f$ it follows that $f$ is constant in the range of $X$, and thus $\max\limits_{x \in X} |f(x)| = \min\limits_{x \in X} |f(x)|$.
**Case (ii): $g$ increases in the range of $X$.**
Let $x, y \in X$ such that $x < y$. By the assumptions (and Claim 2)

$$|1 + x \cdot \text{poly}(c', x)| \leq 2^k \cdot |1 + x \cdot \text{poly}(c', y)|, \text{ and}$$
$$|f(x)| \geq 2^k \cdot |f(y)|,$$

where the first inequality is by Claim 2, and the second follows from the definition of $f$ and dividing both sides by $g(x) \cdot g(y)$. Hence, $\max\limits_{x \in X} |f(x)| \leq 2^k \min\limits_{x \in X} |f(x)|$.
**Case (iii): $g$ decreases in the range of $X$.**
Let $x, y \in X$ such that $x < y$. By the assumptions (and Claim 2)

$$|1 + x \cdot \text{poly}(c', y)| \geq 2^k \cdot |1 + x \cdot \text{poly}(c', x)|, \text{ and}$$
$$|f(y)| \leq 2^k \cdot |f(x)|,$$

where the first inequality is by Claim 2, and the second follows from the definition of $f$ and dividing both sides by $g(x) \cdot g(y)$. Hence, $\max\limits_{x \in X} |f(x)| \leq 2^k \min\limits_{x \in X} |f(x)|$. $\qquad \square$

Using Corollary 3 we obtain the following bound for the RFF sensitivity.

**Lemma 10.** *Let $q_1 = (c_1, c_1') \in (\mathbb{R}^k)^2$. Let $D \subseteq \mathbb{R}$ denote the extrema of the function $g$ that maps every $x \in \mathbb{R}$ to $g(x) = |1 + x \cdot \text{poly}(c_1', x)|$. Let $X \subset \mathbb{R}$ such that for every $x \in [\min(X), \max(X)]$ we have that $\min\limits_{\gamma \in D} |x - \gamma| \geq \max(X) - \min(X)$. Let $q_2 \in (\mathbb{R}^k)^2$ be $2^k$-bounded over $X$; see*

*Definition 5. Let $s : X \to [0, \infty)$ be the sensitivity bound computed in Corollary 2 after substituting $k$ with $2k + 1$ and $Y$ by $X$. For every $x \in X$ we have*

$$\frac{\left|\text{ratio}(q_1, x) - \text{ratio}(q_2, x)\right|}{\displaystyle\sum_{y \in X} \left|\text{ratio}(q_1, y) - \text{ratio}(q_2, y)\right|} \leq 4^k \cdot s(x).$$

*Proof.* If $|X| = 1$, by the construction of $s$ in Corollary 2 we have that the single value $x \in X$ satisfies $s(x) \geq 1$, which yields that the inequality in the lemma holds for this case. Therefore, from now on we assume that this is not the case, i.e., we assume that $|X| > 1$.

Identify $q_2 = (c_2, c_2')$, and let $c \in \mathbb{R}^{2k+1}$ such that for every $x \in \mathbb{R}$ we have

$$\text{poly}(c, x) = \text{poly}(c_1, x) \cdot \left(1 + x \cdot \text{poly}(c_2', x)\right) - \text{poly}(c_2, x) \cdot \left(1 + x \cdot \text{poly}(c_1', x)\right).$$

Let $f_1, f_2 : \mathbb{R} \to [0, \infty)$ denote the functions that map every $y \in X$ to $f_1(y) = \dfrac{1}{1 + y \cdot \text{poly}(c_1', y)}$

and $f_2(y) = \dfrac{1}{1 + y \cdot \text{poly}(c_2', x)}$, respectively. Let $x \in X$. We have

$$\begin{aligned}
\left|\text{ratio}(q_1, x) - \text{ratio}(q_2, x)\right| &= \left|\frac{\text{poly}(c_1, x)}{1 + x \cdot \text{poly}(c_1', x)} - \frac{\text{poly}(c_2, x)}{1 + x \cdot \text{poly}(c_2', x)}\right| = \\
&\left|\left(\frac{\text{poly}(c, x)}{\left(1 + x \cdot \text{poly}(c_1', x)\right) \cdot \left(1 + x \cdot \text{poly}(c_2', x)\right)}\right)\right| = \left|\text{poly}(c, x) \cdot f_1(x) \cdot f_2(x)\right|,
\end{aligned} \tag{11}$$

where the second equality is by assigning the definition of $c$, and the third equality is by assigning the definition of $f_1$ and $f_2$.

Substituting $X$ by $[\min(X), \max(X)]$ and $c'$ by $c_1'$ in Corollary 3 yields that the function $f : X \to \mathbb{R}$ that maps every $x \in X$ to $f(x) = \dfrac{1}{1 + x \cdot \text{poly}(c', x)}$ is well defined, and satisfies $\dfrac{\max\limits_{x \in X} |f(x)|}{\min\limits_{x \in X} |f(x)|} \leq 2^k$. That is, since $q_1 = (c_1, c_1')$, that $q_1$ is $2^k$ well behaved over $X$; see Definition 5.

Hence,

$$\frac{\left|\text{ratio}(q_1, x) - \text{ratio}(q_2, x)\right|}{\displaystyle\sum_{y \in X} \left|\text{ratio}(q_1, y) - \text{ratio}(q_2, y)\right|} = \frac{\left|\text{poly}(c, x) \cdot f_1(x) \cdot f_2(x)\right|}{\displaystyle\sum_{y \in X} \left|\text{poly}(c, y) \cdot f_1(y) \cdot f_2(y)\right|} \tag{12}$$

$$\leq \frac{\max\limits_{y \in X} |f_1(y)|}{\min\limits_{y \in X} |f_1(y)|} \cdot \frac{\max\limits_{y \in X} |f_2(y)|}{\min\limits_{y \in X} |f_2(y)|} \cdot \frac{|\text{poly}(c, x)|}{\displaystyle\sum_{y \in X} |\text{poly}(c, y)|} \tag{13}$$

$$\leq 4^k \cdot \frac{|\text{poly}(c, x)|}{\displaystyle\sum_{y \in X} |\text{poly}(c, y)|} \tag{14}$$

$$\leq 4^k \cdot s(x), \tag{15}$$

where Equation 12 is by Equation 11, Equation 13 follows from assigning that $x \in X$, Equation 14 holds since $q_1$ and $q_2$ are $2^k$-bounded over $X$, and Equation 15 is by assigning the definition of $s$ in the lemma. $\square$

### E.3 CORRECTNESS OF ALGORITHM 4; PROOF OF LEMMA 3

In the following lemma we show a minor result that was used in Algorithm 4.

**Lemma 11.** *Let $c \in \mathbb{R}^k$ and $X = \{a, a + 1 \cdots, b\} \subset [n]$ be a non empty interval of $[n]$. Let $f : \mathbb{R} \to [0, \infty)$ be the function that maps every $x \in \mathbb{R}$ to $f(x) = |1 + x \cdot \text{poly}(c, x)|$. There is a partition $\{X_1, \cdots, X_\eta\}$ of $X$ into $|\eta| \leq 2k - 1$ sets, such that for every $i \in [\eta]$ the function*

$f$ is monotonic over $\left[\min(X_i), \max(X_i)\right)$, and for every $i, j \in [\eta]$, where $i \neq j$ we have $X_i \cap \left[\min(X_j), \max(X_j)\right] = \emptyset$. Moreover, this partition can be computed in $(k+1)^{O(1)} \cdot |X|$ time.

*Proof.* Let $g : \mathbb{R} \to [0, \infty)$ be the function that maps every $x \in \mathbb{R}$ to $g(x) = 1 + x \cdot \text{poly}(c, x)$, which is an polynomial of degree at most $k$. Observe that, by the fundamental theorem of algebra, any non zero polynomial (due to its construction $g$ is non zero) of degree at most $k$ has at most $k$ roots. Thus, since the derivative of $g$ is a polynomial of degree at most $k - 1$, $g$ has at most $k - 1$ extrema. Hence, as any extrema of $f$ is either a root or extrema of $g$, $f$ has at most $2k - 1$ extrema. Partitioning $X$ according to the $2k - 1$ extrema of $f$ yields the partition $\{X_1, \cdots, X_\eta\}$ from the lemma.

**Running time:** Observe that in the root finding presented above (including in the computation of the extrema of $g$) it suffices to only search for roots in the range of $X$, and for the non integer roots only for which integer $a$ they are in $(a, a + 1)$. Each integer candidate for the roots can be validated by a simple assignment in the polynomial in $(k+1)^{O(1)}$ time. By Sturm's-theorem Thomas (1941), we can validate each interval candidate for a root in $(k+1)^{O(1)}$ time. Since there are $|X|$ candidates for roots (intervals and integer), all of the roots can be computed for the sufficient precision in $(k+1)^{O(1)} \cdot |X|$ time. Since $f$ has at most $2k - 1$ extrema, the partition of $X$ can be computed in $O(k \cdot |X|)$ time. Hence, the overall running time is in $(k+1)^{O(1)} \cdot |X|$. $\qquad\square$

Lemma 5 bounds the VC-dimension that corresponds to the function $D(q, p_i)$ over the points $p_1, \cdots, p_n$ in a $n$-signal projected unto its bicriteria; see Definition 2 and Definition 3. We now give a similar bound for the distance function $D(q, (x, y)) = |\text{ratio}(q, x) - y|$ between every rational function $q$ to the point $p_i = (x, y)$, where the points are a general set of points in the plane.

The following lemma, similarly to Lemma 5, is inspired by Theorem 12 in Lucic et al. (2017).

**Lemma 12.** *Let $P = \{(1, y_1), \cdots, (n, y_n)\}$ be an $n$-signal. For every $p_i = (i, y) \in P$ and any $q \in \left(\mathbb{R}^k\right)^2$ let $g_i(q) = D(q, p_i) = |\text{ratio}(q, i) - y|$; see Definition 2. Let $G = \{g_1, \ldots, g_n\}$. The dimension of the range space $\mathbb{R}_{(\mathbb{R}^k)^2, G}$ that is induced by $\left(\mathbb{R}^k\right)^2$ and $G$ is in $O(k^2)$.*

*Proof.* For every $(q, r) = \left((c, c'), r\right) \in \left(\mathbb{R}^k\right)^2 \times \mathbb{R}$, let $h_{(c|c'|r)} : \mathbb{R} \to \{0, 1\}$ that maps every $i \in [n]$ to $h_{(c|c'|r)}(i) = 1$ if and only if $f_i(q) \leq r$, and every $x \in \mathbb{R} \setminus [n]$ to $h_{(c|c'|r)}(x) = 0$. Let $\mathcal{H} = \{h_\theta \mid \theta \in \mathbb{R}^{2k+1}\}$. For every $c \in \mathbb{R}^k$ and any $x \in \mathbb{R}$ we can compute $\text{poly}(c, x)$ with $O(k)$ arithmetic operations on real numbers and jumps conditioned on comparisons of real numbers; see, for example, Horner's scheme Neumaier (2001), which is used in numpy's implementation of the method polyval Harris et al. (2020). Therefore, for every $x \in \mathbb{R}$ and any $\theta \in \mathbb{R}^{2k+1}$, by the definition of $D$, we can calculate $h_\theta(x)$ with $O(k)$ arithmetic operations on real numbers and jumps conditioned on comparisons of real numbers. Hence, substituting $d := n$, $m := 2k + 1$, $h := h$, $\mathcal{H} := \mathcal{H}$ and $t \in O(k)$ in Theorem 2 yields that the VC-dimension of $\mathcal{H}$ is in $O(k^2)$. Hence, by the construction of $\mathcal{H}$ and the definition of range spaces in Definition 7, we have that the dimension of the range space $\mathbb{R}_{P, F}$ that is induced by $P$ and $F$ is in $O(k^2)$. $\qquad\square$

This combined with the previous results yields the following restricted coreset construction that utilizes the previous reduction of the RFF sensitivity to polynomial sensitivity.

**Lemma 13.** *Let an interval of a $n$-signal $P$ which is projected unto some $q \in \left(\mathbb{R}^k\right)^2$, i.e., $\ell(P, q) = 0$. Let $B := \{(P, q)\}$, which is a $(0, 1)$-approximation $B$ of $P$; see Definition 3. Let $X$ be the first coordinate of $P$, i.e., $X := \{x \mid (x, y) \in P\}$. Put $\epsilon, \delta \in (0, 1/10]$, and let*

$$\lambda \geq \frac{c^*}{\epsilon^2} \cdot (4^{k+1} k^2 + 1) \left(k^2 \log(4^{k+1} k^2 + 1) + \log\left(\frac{k \log n}{\delta}\right)\right),$$

*be an integer, where $c^* \geq 1$ is a constant that can that can be determined from the proof. Let $(S, w)$ be the weighted set that is returned by a call to* MINI-REDUCE$(B, \lambda)$; *see Algorithm 4. Then $|S| \in O\left(k\lambda \cdot \log n\right)$ and, with probability at least $1 - \delta$, for every $q' \in \left(\mathbb{R}^k\right)^2$ that is $2^k$-bounded over $X$ (see Definition 5), we have*

$$|\ell(P, q') - \ell((S, w), q')| \leq \epsilon \cdot \ell(P, q'). \tag{16}$$

*Proof.* By the construction of Algorithm 4 we have that $|S| \in O(k\lambda \cdot \log n)$, which follows from the bounds on the order of $\eta$ and every $m_i$ stated in Algorithm 4. Let $\delta' := \lceil c_2^*(k\log n)/\delta \rceil$ for a constant $c_2^* > 0$ that can be determined from the proof, more specifically see Equation 19. Consider the set $X_i^j \subset X$, for $i \in [\eta]$ and $j \in [m_i]$, that was constructed during the execution of the $i$-th iteration of the outer "for" loop and $j$-th iteration of the inner "for" loop in the call to MINI-REDUCE$(B, \lambda)$. Let $(c, c') := q$, and let $D \subseteq \mathbb{R}$ denote the extrema of the function $f$ that maps every $x \in \mathbb{R}$ to $f(x) = |1 + x \cdot \text{poly}(c', x)|$. By the construction of Algorithm 4, the size or diameter of $X_i^j$ is smaller than its distance from the edges of $X_i$, i.e., $|X_j^i| \leq \min\left(|x - \max(X_i^j)|, \|x - \min(x_i^j)\|\right)$. Since $X_i$ has no extreme points, this implies that for every $x \in X_i^j$ we have $\min_{\gamma \in D} |x - \gamma| \geq |X_i^j| = \max(X_i^j) - \min(X_i^j)$. Hence, assigning $X := X_i^j, q_1 := q, q_2 : -q$, and the function $s : X_i^j \to [0, \infty)$ computed in the call to MINI-REDUCE$(B, \lambda)$ for $X_i^j$ in Lemma 10 yields for every $x \in X_i^j$ that

$$\frac{\left|\text{ratio}(q, x) - \text{ratio}(q', x)\right|}{\sum_{y \in X_i^j} \left|\text{ratio}(q, y) - \text{ratio}(q', y)\right|} \leq 4^k \cdot s(x). \tag{17}$$

Let $P_i^j$ and $S_i^j$ be as defined in Line 10 and Line 11, respectively, during the execution of the call to MINI-REDUCE$(B, \lambda)$, and let $w_i^j : S_i^j \to [0, \infty)$ such that for every $p \in S_i^j$ we have $w_i^j(p) = w(p)$. Let $\mathcal{Q}$ denote the union over every $q' \in \left(\mathbb{R}^k\right)^2$ that is $2^k$-bounded over $X$; see Definition 5.

We now prove that $S_i^j$ is an $\epsilon$-subset corset for the query space $\left(P_i^j, \mathcal{Q}, D, \|\cdot\|_1\right)$. Indeed, the corresponding dimension of $\mathbb{R}_{\mathcal{Q}, G}$, where is $G$ is as defined in Lemma 12, is in $O(k^2)$. The sensitivity of every $(x, y) := p \in P_i^j$ is $s(x) = \max_{q \in \mathcal{Q}} D(q, p)/\ell(P_i^j, q) \leq 4^k s(x)$ where the inequality is by Equation 17. The total sensitivity is thus $t = \sum_{x \in X_i^j} 4^k s(x) \in 4^k \cdot O(k^2)$, where the is by the inequality is by definition of $s$ computed in the call to MINI-REDUCE$(B, \lambda)$. Thus, substituting $\delta := \delta'$, the query space, $\left(P_i^j, \mathcal{Q}, D, \|\cdot\|_1\right)$ in Theorem 3 combined with the construction of Algorithm 4 yields that, with probability at least $1 - \delta'$, we have $(S_i^j, w_i^j)$ is an $\epsilon$-subset-coreset for the query space $\left(P_i^j, \mathcal{Q}, D, \|\cdot\|_1\right)$. That is, with probability at least $1 - \delta'$, for every $q' \in \left(\mathbb{R}^k\right)^2$ we have

$$\left|\ell(P_i^j, q') - \ell\left((S_i^j, w_i^j), q'\right)\right| \leq \epsilon \cdot \ell(P_i^j, q'). \tag{18}$$

Taking the union over every $i \in [\eta]$ and $j \in [m_i]$, under the assumption that Equation 18 holds for each pair, yields

$$\left|\ell(P, q') - \ell((S, w), q')\right| = \left|\sum_{i=1}^{\eta} \sum_{j=1}^{m_i} \left(\ell\left(P_i^{(j)}, q'\right) - \ell\left(\left(S_i^{(j)}, w_i^{(j)}\right), q'\right)\right)\right| \leq$$

$$\sum_{i=1}^{\eta} \sum_{j=1}^{m_i} \left|\ell\left(P_i^{(j)}, q'\right) - \ell\left(\left(S_i^{(j)}, w_i^{(j)}\right), q'\right)\right| \leq \sum_{i=1}^{\eta} \sum_{j=1}^{m_i} \epsilon \cdot \ell\left(P_i^{(j)}, q'\right) = \epsilon \cdot \ell(P, q'),$$

where the first equality is by the construction of the weighed set $(S, w)$ and the partition $\left\{P_i^{(j)}\right\}$ of $P$ in Algorithm 4, the first inequality is by the triangle inequality, the second inequality is by assigning Equation 18 (which was assumed to hold for all the values), and the last equality follows since $\left\{P_i^{(j)}\right\}$ computed in Algorithm 4 is a partition of $P$.

Since $i \in [\eta]$ by Line 5 and $j \in [m_i]$ by Line 7 of Algorithm 4, we have at most $O(k \log n)$ sets $P_{i,j}$; follows by assigning the bounds on the order of $\eta$, and every $m_i$ from Algorithm 4. By the union bound, Equation 18 hold simultaneously for every $i \in [\eta]$ and $j \in [m_i]$ with probability at least

$$1 - \delta' \cdot \sum_{i=1}^{\eta} m_i \geq 1 - \delta, \tag{19}$$

which holds for $\delta' := \lceil c_2^*(k \log n)/\delta \rceil$ for some constant $c_2^* > 0$. Hence, with probability at least $1 - \delta$ we have $|\ell(P, q') - \ell((S, w), q')| \leq \epsilon \cdot \ell(P, q')$ as stated in Equation 16 which proves the lemma. □

# F    COMBINING $(\alpha, \beta)$-APPROXIMATIONS; ANALYSIS OF ALGORITHM 3

The input for the algorithm is $P$ an interval of $n$-signal which is projected onto some set of $(\alpha, \beta)$-approximations; see Definition 3. This projection is represented by the set $B$, where each element $B_i \in B$ is a $(0, \beta)$-approximation for some $P_i$, and $\{P_1, \cdots, P_{|B|}\}$ is a consecutive partition of $P$. The algorithm returns $B'$, a bicriteria-approximation of $P$ as in Definition 3, where the size of $B'$ is smaller than $\sum_{i=1}^{|B|} |B_i|$. The algorithm runs in $O(|P|^{1+\epsilon})$ time, for every constant $\epsilon > 0$.

The desired properties of Algorithm 3 are stated and proved in Lemma 14.

For the sake of analysis, we will use the following corollary.

**Corollary 4.** *Let $\{R_1, \cdots, R_\beta\}$ be a set of $\beta \geq 6k$ equally sized distinct partitions of $[n]$ such that for every $i, j \in [\beta], i \neq j$ we have $R_i \cap \left[\min(R_j), \max(R_j)\right] = \emptyset$. For every $q \in \left(\mathbb{R}^k\right)^2$, there is $C \subset [\beta], |C| = \beta - 6k + 3$ such that $q$ is $2^k$-bounded over every $R_i$, for every $i \in C$.*

*Proof.* Let $q = (c, c')$, and let $G$ be the set of the extrema of the function $f : \mathbb{R} \to \mathbb{R}$ that maps every $x \in \mathbb{R}$ to $f(x) = |1 + x \cdot \text{poly}(c', x)|$. Let $r = |R_1| = \cdots = |R_\beta|$. W.l.o.g. assume that for every $i \in [\beta - 1]$ we have $\min(R_{i+1}) > \max(R_i)$; i.e., the sets are ordered in an increasing order. Let $R_0 = \{\min(R_1) - r, \cdots, \min(R_1) - 1\}$ and $R_{\beta+1} = \{\max(R_\beta) + 1, \cdots, \max(R_\beta) + 1 + r\}$. By the proof of Lemma 11, $f$ has at most $2k - 1$ extrema. Therefore, removing from $[\beta]$ every index $i \in [\beta]$ such that $f$ has an extrema in the range of either from $\{R_{i-1}, R_i, R_{i+1}\}$ yields a set $C \subset [\beta], |C| = \beta - 6k + 3$ such that for every $i \in C$ we have $\forall x \in R_i : \min_{g \in G} |x - g| \geq r = \max(R_i) - \min(R_i)$. For every $i \in C$, substituting $c', G$ and $X := R_i$ in Corollary 3 yields that the function $g : X \to \mathbb{R}$ that maps every $x \in R_i$ to $g(x) = \dfrac{1}{1 + x \cdot \text{poly}(c', x)}$ is well defined, and satisfies $\dfrac{\max_{x \in X} |g(x)|}{\min_{x \in X} |g(x)|} \leq 2^k$. Thus, $q$ is $2^k$-bounded over $R_i$ for every $i \in C$; see Definition 5. Hence, $C$ satisfies the corollary. $\qquad\square$

Combining a removal inspired by Rosman et al. (2014), with the coreset construction from the previous section, yields the following lemma.

**Lemma 14.** *Let $B := \{B_1, \cdots, B_\beta\}$, where each $B_i \in B$ is an $(0, r_i)$-approximation of $P_i$, i.e. $P_i$ is projected unto $B_i$, and $\{P_1, \cdots, P_\beta\}$ is an equally-sized consecutive partition of $P$, some interval of an $n$-signal; see Figure 6 and Definition 3. Put $\epsilon, \delta \in (0, 1/10]$, and let*

$$\lambda := \left\lceil \frac{c^*}{\epsilon^2} (4^{k+1} k^2 + 1) \left( k^2 \log_2(4^{k+1} k^2 + 1) + \log_2\left(\frac{kn}{\delta}\right) \right) \right\rceil$$

*be an integer, where $c^* \geq 1$ is a constant that can be determined from the proof. Let $B'$ be the output of $\text{REDUCE}(B, \lambda, 6k - 3)$; see Algorithm 3. With probability at least $1 - \delta$, we have that $B'$ is a $(1 + 10\epsilon, \beta^*)$-approximation of $P$ for some $\beta^* \geq 1$; see Definition 3.*

*Moreover, for $\epsilon = 1/10$ we have that the running time of the call to $\text{REDUCE}(B, \lambda, 6k - 3)$ is in $|P| \cdot \beta^{6k-3} \cdot 4^{O(k^2)} (\log(n/\delta)\beta')^{O(k)}$, where $\beta' = \sum_{i=1}^{\beta} |B_i|$.*

*Proof.* If $\beta \leq 12k - 6$, by the construction of $B'$ in Algorithm 3 in Lines 1 and 20, we have $B' := \bigcup_{i=1}^{\beta} B_i$. Since $B_i$ is a $(0, r_i)$-approximation of $P_i$ for every $i \in [\beta]$ and $\{P_1, \cdots, P_\beta\}$ is a partition of $P$, $B'$ is an $(0, |B'|)$-approximation of $P$ which yields that the lemma trivially holds. Thus, from now on, we assume that this is not the case.

For every $B_i \in B$ identify $\left\{ B_i^{(1)}, \cdots, B_i^{(r_i)} \right\} := B_i$. Let $q \in \arg\min_{q \in (\mathbb{R}^k)^2} \ell(P, q)$. For every $i \in [\beta]$, let $R_i := \{x \mid (x, y) \in P_i\}$, i.e. the first coordinates of the points in $P_i$. Since

$\{P_1, \cdots, P_\beta\}$ is an equally-sized consecutive partition of some interval of an $n$-signal $P$, we have that $\{R_1, \cdots, R_\beta\}$ is a set of equally-sized partitions of $[n]$ such that for every $i, j \in [\beta], i \neq j$ we have $R_i \cap \big[\min(R_j), \max(R_j)\big] = \emptyset$. By Corollary 4, there is $G \subset [\beta], |G| = \beta - 6k + 3$ such that $q$ is $2^k$-bounded over $R_i$ for every $i \in G$. Consider the integration of the "for" loop in the call to Algorithm 3 where $G$ is computed, i.e., $G := G$.

Let $G', S_G, w_G$, and $q_G$ be defined as in the "for" loop iteration in the call to Algorithm 3. Let $P_G$ and $P_{G'}$ be the union of $P_i$ over every $i \in G$ and $i \in G'$, respectively. Let $P_{G \setminus G'}$ denote $P_{G'} \cup (P \setminus P_G)$. The set $\left\{ B_i^{(j)} \mid i \in G' \cup ([\beta] \setminus G), B_i^{(j)} \in B_i \right\}$ is a $\left(0, \sum_{i=1}^{\beta} |B_i|\right)$-approximation for $P_{G \setminus G'}$. Hence, by the construction of $B'$ in Line 20 of Algorithm 3 it is left to prove that

$$\ell(P_{G \setminus G'}, q_G) \leq (1 + 10\epsilon) \cdot \ell(P_G, q) \leq (1 + 10\epsilon) \cdot \ell(P, q). \tag{20}$$

The last inequality trivially holds since $P_G \subseteq P$. It is left to prove the first inequality of Eq. Equation 20. By Corollary 4, where we substitute $\beta$ by $|C|$, there is a set $G^* \subset G$ of size $|G^*| = |G| - 6k + 3$ such that $q_G$ is $2^k$-bounded over $R_i$ for every $i \in G^*$. For every $i \in G$ and any $B_i^{(j)} \in B_i$ identify $\left(P_i^{(j)}, q_i^{(j)}\right) := B_i^{(j)}$. For every $i \in G$ and any $B_i^{(j)} \in B_i$ let $\left(S_i^{(j)}, w_i^{(j)}\right) := \text{MINI-REDUCE}\left(\left\{B_i^{(j)}\right\}, \lambda\right)$ as computed in Line 5 of the call to Algorithm 3; for $B_i^{(j)} := B_i^{(j)}$. Substituting $P = P_i^{(j)}, \delta = \delta/(2n), q' = q_G$, and $X := \left\{x \mid (x, y) \in P_i^{(j)}\right\}$ (the first coordinate of $P_i^{(j)}$) in Lemma 13 for every $i \in G^*$ and $j \in [|B_i|]$ (by the choice of $G^*$, $q_G$ is $2^k$-bounded over $R_i$, which contains the first coordinate of $P_i^{(j)}$) yields that with probability at least $1 - \dfrac{\delta}{2n}$ we have

$$\left| \ell\left(P_i^{(j)}, q_G\right) - \ell\left(\left(S_i^{(j)}, w_i^{(j)}\right), q_G\right) \right| \leq \epsilon \cdot \ell\left(P_i^{(j)}, q_G\right). \tag{21}$$

Let $S^* := \bigcup_{i \in G \setminus G^*}, B_i^{(j)} \in B_i S_i^{(j)}$, and $w^* : S^* \to [0, \infty)$ be the function that maps every $p \in S^*$ to $w^*(p) = w_G(p)$. Applying Equation 21 for every $i \in G^*$ and $j \in [|B_i|]$, along with utilizing the union bound, yields that, with probability at least $\left(1 - \delta/(2n)\right)^n \geq 1 - \delta/2$, we have

$$\left| \ell(P_{G \setminus G^*}, q_G) - \ell((S^*, w^*), q_G) \right| = \left| \sum_{i \in G^*} \sum_{j=1}^{|B_i|} \left( \ell\left(P_i^{(j)}, q_G\right) - \ell\left(\left(S_i^{(j)}, w_i^{(j)}\right), q_G\right) \right) \right| \tag{22}$$

$$\leq \sum_{i \in G^*} \sum_{j=1}^{|B_i|} \left| \ell\left(P_i^{(j)}, q_G\right) - \ell\left(\left(S_i^{(j)}, w_i^{(j)}\right), q_G\right) \right| \tag{23}$$

$$\leq \epsilon \cdot \ell(P_{G \setminus G^*}, q_G), \tag{24}$$

where Equation 22 is by the constructions of $P_{G^*}$ and $S^*$, Equation 23 is by the triangle inequality, and Equation 24 is by Equation 21. Substituting $P = P_i^{(j)}, \delta = \delta/(2n), q' = q$, and $X := \left\{x \mid (x, y) \in P_i^{(j)}\right\}$ (the first coordinate of $P_i^{(j)}$) in Lemma 13 for every $i \in G$ and $B_i^{(j)} \in B_i$ (by the choice of $G$, $q$ is $2^k$-bounded over $R_i$, which contains the first coordinate of $P_i^{(j)}$) yields that with probability at least $1 - \dfrac{\delta}{2n}$ we have

$$\left| \ell\left(P_i^{(j)}, q\right) - \ell\left(\left(S_i^j, w_i^j\right), q\right) \right| \leq \epsilon \cdot \ell\left(P_i^{(j)}, q\right). \tag{25}$$

Applying Equation 25 for every $i \in G$ and $B_i^{(j)} \in B_i$, along with utilizing the union bound, yields that, with probability at least $\left(1 - \delta/(2n)\right)^n \geq 1 - \delta/2$, we have

$$
\left| \ell\left(P_G, q\right) - \ell\left((S_G, w), q\right) \right| = \left| \sum_{i \in G} \sum_{j=1}^{|B_i|} \left( \ell\left(P_i^{(j)}, q\right) - \ell\left(\left(S_i^{(j)}, w_i^{(j)}\right), q\right) \right) \right| \tag{26}
$$

$$
\leq \sum_{i \in G} \sum_{j=1}^{|B_i|} \left| \ell\left(P_i^{(j)}, q\right) - \ell\left(\left(S_i^{(j)}, w_i^{(j)}\right), q\right) \right| \tag{27}
$$

$$
\leq \epsilon \cdot \ell\left(P_G, q\right), \tag{28}
$$

where Equation 26 is by the constructions of the $P_G$ and $S_G$ in Algorithm 3, Equation 27 is by the triangle inequality, and Equation 28 is by Equation 21.

If both Equation 22 to Equation 24 and Equation 26 to Equation 28 holds, which happens with probability at least $1 - \delta$, we have

$$
\ell(P_{G \setminus G'}, q_G) \leq \ell(P_{G \setminus G^*}, q_G) \tag{29}
$$

$$
\leq \frac{1}{1 - \epsilon} \cdot \ell\left((S^*, w^*), q_G\right) \tag{30}
$$

$$
\leq \frac{1}{1 - \epsilon} \cdot \ell\left((S, w), q_G\right) \tag{31}
$$

$$
\leq \frac{1}{1 - \epsilon} \cdot \ell\left((S, w), q^*\right) \tag{32}
$$

$$
\leq \frac{1 + \epsilon}{1 - \epsilon} \cdot \ell(P_G, q^*) \tag{33}
$$

$$
\leq (1 + 10\epsilon) \cdot \ell(P_G, q^*), \tag{34}
$$

where Equation 29 is by the choice of $G'$ in Line 16 of Algorithm 3, Equation 30 is by Equation 22 to Equation 24, Equation 31 is since $S^* \subset S$, Equation 32 is since $q \in \arg\min_{q' \in (\mathbb{R}^k)^2} \ell\left((S, w), q'\right)$ (by the construction of Algorithm 3), Equation 33 is by Equation 26 to Equation 28, and Equation 34 holds since $\epsilon \leq 1/2$.

**Running time:**

Let $\beta' := \sum_{i=1}^{\beta} |B_i|$ and assign $\epsilon = \frac{1}{10}$.

If $\beta < 6k - 3$, then, by the construction of Algorithm 3, the output can be computed in $O(|P| \cdot \beta')$. Hence, from now on we assume this is not the case.

Consider a single "for" iteration over the values of $G \subset [\beta]$ during the execution of Line 6 in the call to Algorithm 3. By the construction of every $\left(S_i^{(j)}, w_i^{(j)}\right)$ in Line 5 of Algorithm 3, we have that $\left|S_i^{(j)}\right| \leq \lambda$, for every $i \in [\beta]$ and $j \in [|B_i|]$. Hence, since $|S_G|$ is the union of $S_i^{(j)}$, over every $i \in G \subset [\beta]$ and $j \in [|B_i|]$, recalling the definition of $\beta'$ trivially yields $|S_G| \in O(\lambda \beta')$. By Lemma 8, Line 11 can be computed in $(2k|S_G|)^{O(k)} \in \left(\lambda \cdot \beta'\right)^{O(k)}$ time. The rest of the lines can be computed in $|P| \cdot (k+1)^{O(1)}$ time, therefore, every iteration can be computed $|P| \cdot (k+1)^{O(1)} + \left(\lambda \cdot \beta'\right)^{O(1)}$ time. Assigning $\epsilon = \frac{1}{10}$ in the definition of $\lambda$ in the lemma yields $\lambda \in 4^{O(k)} \log(n/\delta)$. Since there are $\binom{\beta}{6k-3} \in O\left(\beta^{6k-3}\right)$ iterations of the "for" loop we obtain a total running time of $|P| \cdot \beta^{6k-3} \cdot 4^{O(k^2)} \left(\log(n/\delta)\beta'\right)^{O(k)}$. $\qquad\square$

# G ALGORITHM 1: STREAMING

In this section we prove that Algorithm 1, which gets $P$ an $n$-signal of length $n \geq 2k$ is a power of 2, and $\epsilon, \delta \in (0, 1/10]$, returns an $\epsilon$-approximation of $P$ with failure probability at most $\delta$. The formal statement and its proof is given in Theorem 5.

For the analysis we will use the following corollary, which is inspired by Lemma 3.6 in Braverman et al. (2020) and proves that projecting a dataset on a corresponding approximation for some function yields a coreset for this function.

**Corollary 5.** *Let $P$ be an $n$-signal. Let $B := \{(P_1, q_1), \cdots, (P_\beta, q_\beta)\}$ be an $(\alpha, \beta)$-approximation of $P$; see Definition 3. For every $i \in [\beta]$ let $P_i^* := \{(x, \mathrm{ratio}(q_i, x)) \mid (x, y) \in P_i\}$, and let $P^* := \bigcup\limits_{i=1}^{\beta} P_i^*$; i.e., $P^*$ is the projection of $P$ onto $B$. Let $q = (c, c') \in (\mathbb{R}^k)^2$ such that $1 + x \cdot \mathrm{poly}(c', x) \neq 0$ for every $(x, \cdot) \in P$. Then $\ell(P^*, q) \leq (1 + \alpha) \cdot \ell(P, q)$.*

*Proof.* Let $q$ be defined in the corollary. We have

$$\ell(P^*, q) = \sum_{i=1}^{\beta} \sum_{(x,y) \in P_i} \left| \mathrm{ratio}(q, x) - \mathrm{ratio}(q_i, x) \right| \tag{35}$$

$$\leq \sum_{i=1}^{\beta} \sum_{(x,y) \in P_i} \left( \left| \mathrm{ratio}(q_i, x) - y \right| + \left| \mathrm{ratio}(q, x) - y \right| \right) \tag{36}$$

$$= \sum_{i=1}^{\beta} \sum_{p \in P_i} D(q_i, p) + \sum_{p \in P} D(q, p) \tag{37}$$

$$\leq (1 + \alpha) \cdot \ell(P, q), \tag{38}$$

where Equation 35 is by assigning the definition of $D$ and the definitions from the corollary, Equation 36 is by the triangle inequality, Equation 37 by the definition of $D$, and Equation 38 is by the definitions from the corollary. $\qquad\square$

At first glance, it may seem that there would be a significant problem using the previously discussed variant of the Merge-Reduce scheme presented in Braverman et al. (2020). Indeed, using the classic version where combining each pair of coresets would yield that the guarantee on the final approximation would be that the cost of the up to a polynomial in the data-set's size factor from the optimal solution (hence, the sample in SAMPLE-CORESET would be larger than the original dataset).

This can be fixed by combining a significantly larger number of nodes, which by the following observation would yield that height of the tree would be $\lceil \log \log n \rceil$, and as a consequence that the final approximation would be poly-logarithmic in the dataset size.

The following theorem and Algorithm 1 are the main result of this work.

**Theorem 5.** *Let $P$ be an $n$-signal, for $n$ that is a power of 2, and put $\epsilon, \delta \in (0, 1/10]$. Let $(B, C, w)$ be the output of a call to CORESET$(P, k, \epsilon, \delta)$; see Algorithm 1. With probability at least $1 - \delta$, $(B, C, w)$ is an $\epsilon$-coreset of $P$; see Definition 4. Moreover, the computation time of $(B, C, w)$ is in*

$$2^{O(k^2)} \cdot n \cdot n^{O(k)/\log\log(n)} \cdot \log(n)^{O(k\log(k))} \cdot \log(1/\delta)^{O(k)},$$

*and the memory words required to store $(B, C, w)$ are in*

$$(2k)^{O(1)} \cdot \log(n)^{O(1) + \log(k)} \cdot \log(1/\delta)/\epsilon^2.$$

*Which, when considering $k$ and $\log(1/\delta)$ as constants yields that the running time is in $O\left(n^{1+o(1)}\right)$ and the space is in $O\left(n^{o(1)}/\epsilon^2\right)$.*

*Proof.* Let $\beta := \lceil n^{1/\log\log(n)} \rceil$ and $\tilde{\beta} = \lceil n/\beta \rceil$ be the values that are defined in the call to CORESET$(P, k, \epsilon, \delta)$. Let $(B_1, \cdots, B_{\beta'})$ be the output of the call to BATCH-APPROX$(P, \tilde{\beta})$

in Line 4 of Algorithm 1; see Algorithm 5. Let $i \in [\beta']$ and identify $(\tilde{P}_i, q_i) := B_i$. Let $P_i := \left\{ (x,y) \mid (x, \cdot) \in \tilde{P}_i \right\}$, i.e., $\{P_1, \cdots, P_\psi\}$ is the partition computed in the call to BATCH-APPROX$(P, \tilde{\beta})$. By the construction $B_i$ in Line 3 of Algorithm 5 we have $q_i \in \underset{q \in (\mathbb{R}^k)^2}{\arg \min} \ell(P_i, q)$. Let $\mathcal{Q}$ be the union over the pairs $(c, c') \in (\mathbb{R}^k)^2$ such that $1 + x \cdot \mathrm{poly}(c', x) \neq 0$ for every $(x, \cdot) \in P_i$. Hence, for every $q \in \mathcal{Q}$ by assigning $P := P_i, B := \{(P, q)\}, P^* := \tilde{P}_i$ and $\alpha, \beta = 1$ in Corollary 5, we obtain

$$\ell(\tilde{P}_i, q) \leq 2 \cdot \ell(P_i, q). \tag{39}$$

Let $\left\{ B'_1, \cdots, B'_\psi \right\}$ be a set of biciterias, as defined as in Line 6 during the first iteration of the "while" loop in the call to Algorithm 1; see Definition 3. Let $i \in [\psi]$, and $P' := \bigcup_{(Y, q') \in B \in B'_i} Y$, i.e., the union of all the sets of points in the bicritrias in $B'_i$. Let $B_i := \mathrm{REDUCE}\left(B'_i, \lambda_1\right)$ as computed in Algorithm 1. Identify $B_i = \left\{ (\tilde{P}_1, q_1), \cdots, (\tilde{P}_r, q_r) \right\}$, and put $P'_a := \left\{ (x, y) \in P' \mid (x, \cdot) \in \tilde{P}_a \right\}$, for every $a \in [r]$, i.e. every $P'_a$ is the set of points in $P'$ that are approximated by $q_a$. Replacing $\epsilon$ with $1/10$, $B$ with $B'_i$ and $\delta$ with $\delta/(4n)$ in Lemma 2 yields that there is $\lambda_1$ as defined in Line x of Algorithm 1 such that $B_i$ with probability at least $1 - \delta/(4n)$ is an $(2, |B_i|)$-approximation to $P'$. That is, with probability at least $1 - \delta/(4n)$ we have

$$\sum_{a=1}^{r} \ell(P'_a, q_a) \leq 2 \min_{q \in (\mathbb{R}^k)^2} \sum_{a=1}^{r} \ell(P'_a, q). \tag{40}$$

For every $B_a \in B'_i$, let $P_a = \{(x, y) \in P \mid x \in P'_a\}$, i.e., the points in $P$ which are approximated by the biciteria $B_a$; see Definition 3. Plugging Equation 39 in the right side of Equation 40 yields that, with probability at least $1 - \delta/(4n)$, we have

$$\sum_{a=1}^{r} \ell(P_a, q_a^*) \leq 4 \min_{q \in (\mathbb{R}^k)^2} \sum_{a=1}^{r} \ell(P_a, q).$$

That is, with probability at least $1 - \delta/(4n)$, $B_i$ is a $(4, |B_i|)$-approximation to $\bigcup_{a=1}^{r} P_a$. Let $\mathcal{Q}$ be the union over the pairs $(c, c') \in (\mathbb{R}^k)^2$ such that $1 + x \cdot \mathrm{poly}(c', x) \neq 0$ for every $(x, \cdot) \in P_i^*$. Put . Hence, with probability at least $1 - \delta/(4n)$, for every $q \in \mathcal{Q}$ by assigning $P := \bigcup_{a=1}^{r} P_a, B := B_i, P^* := \bigcup_{a=1}^{r} \tilde{P}_a, \alpha = 4$ and $\beta = |B_i|$ in Corollary 5, we obtain

$$\sum_{a=1}^{r} \ell(\tilde{P}_a, q) \leq 5 \sum_{a=1}^{r} \ell(P_a, q).$$

By the construction of Algorithm 1 it trivially holds there are $\lceil \log \log(n) \rceil - 2$ iterations of the "while" loop, and $2n$ calls to REDUCE. Hence, repeating the proof above recursively for every iteration of the "while" loop and the last call to REDUCE in Line 10 yields that with probability at least $(1 - \delta/(4n))^{2n} \geq 1 - \delta/2$ we have that $B$ computed in Line 10 is an $\left(3^{\lceil \log \log(n) \rceil}, \beta^*\right)$-approximation to $P$, for some integer $\beta^* \geq 1$.

By the construction of Algorithms [3,1], and that there are $\lceil \log \log(n) \rceil - 2$ iterations of the "while" loop we have $\beta^* \leq (24k)^{\lceil \log \log(n) \rceil}$. Therefore, combining this with Lemma 1 yields that there is $\lambda_2$ as defined in the call to Algorithm 1, and proves the theorem.

**Space complexity.** Let $\lambda_1$ as defined in Algorithm 1 (conditions were set in the previous part of the proof). From the previous section we have that $B$ is a $O\left(\log(n), \log(n)^{O(1) + \log(k)}\right)$-approximation

of $P$. Hence, by the construction of Algorithm 2 and assigning $\lambda_2$ (conditions were set in the previous section) we have that $(B, C, w)$ can be represented in $(2k)^{O(1)} \cdot \log(n)^{O(1)+k} \cdot \log(1/\delta)/\epsilon^2$ space, which proves the claim for memory size.

**Running time.** Note that for every iteration of the "while" loop in the call to $\text{CORESET}(P, k, \epsilon, \delta)$:

- By Lemma 2 and the previous analysis, we have that each call to $\text{REDUCE}(B_i', \lambda_1)$ in $\text{CORESET}(P, k, \epsilon, \delta)$ takes $n_i \cdot \beta^{O(k)} + (\beta\beta^*\lambda_1)^{O(k)}$, where $n_i = \displaystyle\sum_{B_i \in B_i'} \sum_{(Y, q') \in B_i} |Y|$.

- Put $\psi \leq n$ as the number of sets $B_i'$ computed in this iteration of the "while" loop.

- Hence, since $n = \displaystyle\sum_{i=1}^{\psi} n_i$, we have that the computation time of the iteration is in

$$\sum_{i=1}^{\psi} \left( n' \cdot \beta^{O(k)} + (\beta\beta^*\lambda_1)^{O(k)} \right) = n \cdot \beta^{O(k)} + \psi(\beta\beta^*\lambda_1)^{O(k)} \leq n \cdot \beta^{O(k)} \cdot (\beta^*\lambda_1)^{O(k)}.$$

By the computation of $\lambda_1, \beta$ in the call to $\text{CORESET}(P, k, \epsilon, \delta)$, and that $\beta^* \in \log(n)^{O(1)+\log(k)}$ (from the previous sections) yields that the computation time of the iteration is in

$$2^{O(k^2)} \cdot n \cdot n^{O(k)/\log\log(n)} \cdot \log(n)^{O(k\log(k))} \cdot \log(1/\delta)^{O(k)}.$$

By the construction of Algorithm 1 there are $\lceil \log\log(n) \rceil - 2$ iterations of the "while" loop. Hence, the while-loop takes $2^{O(k^2)} \cdot n \cdot n^{O(k)/\log\log(n)} \cdot \log(n)^{O(k\log(k))} \cdot \log(1/\delta)^{O(k)}$ time. Therefore, we have that the running time of the call to Algorithm 1 is in

$$2^{O(k^2)} \cdot n \cdot n^{O(k)/\log\log(n)} \cdot \log(n)^{O(k\log(k))} \cdot \log(1/\delta)^{O(k)}. \qquad \square$$

## H FAST PRACTICAL HEURISTIC; ANALYSIS OF ALGORITHM 6

Unfortunately, since for the slow solver we used Pizzo et al. (2018) which utilizes polynomial programming which takes significant running time, the running time of our robust algorithms is still large. Therefore, we suggested a heuristic in Algorithm 6 to run on top of our coreset. In this section we prove that, under some assumptions, this heuristic gives a constant factor approximation.

For the sake of the analysis of Algorithm 6 we prove the following result for fitting a hyperplane to a set of points.

### H.1 FITTING PLANE TO POINTS

In Lemma 17 we will prove that a common heuristic for fitting hyperplanes to points does give approximation guarantees for points of bounded coordinates. This result would later be combined with some assumptions to give the desired approximation guarantees for Algorithm 6.

**Lemma 15.** *Let $\{(x_1, y_1) \cdots, (x_n, y_n)\}$ be a set of $n \geq 1$ points on the plane, where $x_i \neq 0$ for every $i \in [n]$, let $a := \dfrac{1}{n} \displaystyle\sum_{i=1}^{n} |x_i|$, and let $k \geq 1$ an integer such that*

$$\forall i \in [n] : \frac{a}{k} \leq |x_i| \leq ka. \tag{41}$$

*For every $i \in [n]$ let $c_i := \dfrac{y_i}{x_i}$, and let $G := \{c_1, \cdots, c_n\}$ be a multi-set of size $n$. Let $c$ be a random item from $G$, sampled uniformly. Then, with probability at least $1/2$ we have*

$$\sum_{(x_i, y_i) \in P} |x_i \cdot c - y_i| \leq \left(2k^2 + 1\right) \cdot \min_{c' \in \mathbb{R}} \sum_{(x_i, y_i) \in P} |x_i \cdot c' - y_i|. \tag{42}$$

*Proof.* Let $c^* \in \arg\min\limits_{c' \in \mathbb{R}} \sum\limits_{(x,y) \in P} |x \cdot c' - y|$. By Markov's inequality, with probability at least $1/2$ we have

$$|c - c^*| \leq \frac{2}{n} \sum_{c' \in G} |c^* - c'| := k'. \tag{43}$$

Suppose this event indeed occurs. For every $(x, y) \in P$ and $c' \in G$ such that $xc' = y$ (there exist such $c'$ by the definition of $G$ in the lemma), we have

$$|xc - y| = |xc - xc'| \tag{44}$$
$$= |x| \cdot |c - c'| \tag{45}$$
$$= |x| \cdot |c - c^* - c' + c^*| \tag{46}$$
$$\leq |x| \cdot (|c^* - c| + |c' - c^*|) \tag{47}$$
$$\leq |x|k' + |x| \cdot |c^* - c'| \tag{48}$$
$$\leq kk'a + |c^*x - y|, \tag{49}$$

where Equation 44 is by the choice of $c' \in G$ as a value satisfying $xc' = y$, Equation 45 and Equation 46 are by reorganizing the expression, Equation 47 is by the triangle inequality, and Equation 48 is by the definition of $k'$ from Equation 43, and Equation 49 is by assigning that $|x| \leq ka$ (from the definition of $P$ in the lemma, substituting $x_i := x$ in Equation 41) and that $xc' = y$ (from the choice of $q \in G$).

Hence,

$$\sum_{(x,y) \in P} |xc - y| - \sum_{(x,y) \in P} |xc^* - y| \leq nkk'a \tag{50}$$

$$= 2k \sum_{i=1}^{n} a|c_i - c^*| \tag{51}$$

$$= 2k \cdot \sum_{i=1}^{n} \frac{a|x_i c_i - x_i c^*|}{|x_i|} \tag{52}$$

$$= 2k \cdot \sum_{(x,y) \in P} \frac{a|y - xc^*|}{|x|} \tag{53}$$

$$\leq 2k \cdot \sum_{(x,y) \in P} k|y - xc^*| \tag{54}$$

$$= (2k^2) \cdot \sum_{(x,y) \in P} |xc^* - y|, \tag{55}$$

where Equation 50 is by summing over every $(x, y) \in P$ utilizing Equation 44 to Equation 49, Equation 51 is by assigning the definition of $k'$ from Equation 43, Equation 52 is by multiplying and dividing the expression inside the sum by $a_i$, Equation 53 follow from the definition of $c_i$ as $\frac{y_i}{x_i}$ in the lemma, Equation 54 is by Equation 41, and Equation 55 is by reorganizing the expression. $\square$

By assuming that similar assumptions hold for every dimension separately we can generalize the previous lemma for higher dimensions, as done in the following lemma.

**Lemma 16.** *Let $P = \left\{ \left( (x_i^{(1)}, x_i^{(2)}, \cdots, x_i^{(d)}), b_i \right) \right\}_{i=1}^{n} \subset \mathbb{R}^d \times \mathbb{R}$ be a set of $n \geq 1$ points, where for every $(i, j) \in [n] \times [d]$ we have $x_i^{(j)} \neq 0$, let $a^{(j)} := \frac{1}{n} \sum\limits_{i=1}^{n} |a_i^{(j)}|$ for every $j \in [d]$, and let $k \geq 1$ be an integer such that*

$$\forall (i, j) \in [n] \times [d] : \frac{a^{(j)}}{k} \leq |a_i^{(j)}| \leq a^{(j)}k.$$

*For every $S \subset P$ of size $d$ let $c_S \in \mathbb{R}^d$ such that $\left|a^T \cdot c_S - y\right| = 0$; there is such value by properties of linear regression. Let $G$ be the multi-set $\{c_S \mid S \subset P, |S| = d\}$. Let $c$ be a random item from $G$, sampled uniformly. Then, with probability at least $2^{-d}$ we have*

$$\sum_{(a,y) \in G} \left|a_i^T \cdot c - y_i\right| \leq \left(2k^2 + 1\right)^d \cdot \min_{c' \in \mathbb{R}^d} \sum_{(a,y) \in G} \left|a_i^T \cdot c' - y_i\right|. \tag{56}$$

*Proof.* Let $c' = (c_1', \cdots, c_d') \in \mathbb{R}^d$, and $j' \in [d]$. For every $i \in [n]$ let $x_i := x_i^{(j')}$, let $y_i' := y_i + \sum_{j=1}^{d} x_i c_i' - x_i c_d'$, and let $c_i := \dfrac{y_i'}{x_i}$. Let $P' := \{(x_1, y_1'), \cdots, (x_n, y_n')\}$, $G := \{c_1, \cdots, c_n\}$ be multi sets, both of size $n$. Let $c$ be a random item from $G'$, sample uniformly, and let $c^*$ be $c'$ were we substitute the $j'$th entry by $c$. By the definition of $P'$ and $c'$ we have

$$\sum_{(x,y) \in P} |x^T \cdot c^* - y| = \sum_{(x,y) \in P'} |x \cdot c - y|. \tag{57}$$

From the definition of $P$ in the lemma, for every $(i,j) \in [n] \times [d]$ we have $\dfrac{a^{(j)}}{k} \leq \left|a_i^{(j)}\right| \leq ka^{(j)}$. Hence, by the construction of $P'$ we have $a^{(j')} := \dfrac{1}{n} \sum_{(x,y) \in P'} |x|$, and for every $(x,y) \in P'$ we have $\dfrac{a^{(j')}}{k} \leq |x| \leq ka^{(j')}$. Thus, by substituting $P$ by $P'$ and $G$ by $G'$ in Lemma 15, with probability at least $1/2$, we have

$$\sum_{(x,y) \in P'} |x \cdot c - y| \leq (2k^2 + 1) \cdot \sum_{(x,y) \in P'} |x \cdot c_i' - y|.$$

Combining this with Equation 57 yields that with probability at least $1/2$ we have

$$\sum_{(x,y) \in P} |x^T \cdot c^* - y| \leq (2k^2 + 1) \cdot \sum_{(x,y) \in P'} |x \cdot c_i' - y| = (2k^2 + 1) \sum_{(x,y) \in P} |x^T \cdot c' - y|,$$

where the equality is by the construction of $P'$.

Let $c' = (c_1', \cdots, c_d') \in \arg\min_{c' \in \mathbb{R}^d} \sum_{(x,y) \in P} |x^T c' - y|$, and let $(c_1, \cdots, c_d) := c$. For every $j \in [d]$ let $c_j^* = (c_1, \cdots, c_j, c_{j+1}', \cdots, c_d')$. Let $c_0^* = c'$. By the proof above, for every $j \in [d]$, with probability at least $1/2$, we have

$$\sum_{(x,y) \in P} |x^T \cdot c_j^* - y| \leq (2k^2 + 1) \cdot \sum_{(x,y) \in P} |x^T \cdot c_{j-1}^* - y|.$$

Assigning $c_0^* = c' \in \arg\min_{c' \in \mathbb{R}^d} \sum_{(x,y) \in P} |x^T c' - y|$ and $c_d^* = c$ in the above, combined with the construction of $G$ in the lemma yields Equation 56. $\square$

By repeatedly taking a sample from the set $G$ in the lemma above we obtain the following lemma that gives the desired approximation guarantees for linear regression under assumptions.

**Lemma 17.** *Let $P = \left\{\left(\left(a_i^{(1)}, a_i^{(2)}, \cdots, a_i^{(d)}\right), b_i\right)\right\}_{i=1}^{n} \subset \mathbb{R}^d \times \mathbb{R}$ be a set of $n \geq 1$ points, where there is an integer $k \geq 1$ such that*

$$\forall(i,j) \in [n] \times [d] : \frac{1}{nk} \sum_{i'=1}^{n} |a_{i'}^{(j)}| \leq |a_i^{(j)}| \leq \frac{k}{n} \sum_{i'=1}^{n} |a_{i'}^{(j)}|. \tag{58}$$

*For every $S \subset P$ of size $d$ let $c_S \in \mathbb{R}^d$ such that $\sum_{(a,y) \in S} \left|a^T \cdot c - y\right|$; there is such value by properties of linear regression. Let $G$ be the multi-set that is the union over every $S \subset P$ of size $|S| = d$. Let*

$\delta \in (0,1), \epsilon = 2^{-d}$, and let $\lambda := \max \left\{ \left\lceil \log_{1-\epsilon}(\delta) \right\rceil, 1 \right\}$. Let $S \subset G, |S| = \lambda$ where each value is sampled i.i.d. and uniformly. With probability at least $1 - \delta$ there is $c \in S$ such that

$$\sum_{(x,y) \in P} |x^T c - y| \le (2k^2 + 1)^d \cdot \min_{c' \in \mathbb{R}^d} \sum_{(x,y) \in P} |x^T c' - y|. \tag{59}$$

*Proof.* By Lemma 16, for a uniformly sampled $c \in G$ with probability most $1 - \epsilon$ we have

$$\sum_{(x,y) \in P} |x^T c - y| > (2k^2 + 1)^d \cdot \min_{c' \in \mathbb{R}^d} \sum_{(x,y) \in P} |x^T c' - y|. \tag{60}$$

Hence, by the union bound, the probability that Equation 60 holds for every $c \in S$ is at most

$$(1 - \epsilon)^\lambda \le (1 - \epsilon)^{\log_{1-\epsilon} \delta} = \delta,$$

where the inequality is since $\lambda \ge \log_{1-\epsilon} \delta$. $\qquad \square$

### H.2 Solver computation; see Definition 6

In this section we prove, for $S \subset \mathbb{R}^2$ of size $2k$ satisfying $\left| \{x \cdot y \mid (x,y) \in S\} \right| = 2k$, that $\text{SOLVER}(S)$ is never empty and that it can be computed in $O\left(k^3\right)$ time; see Definition 6.

This method is a generalization of a technique shown in NIST/SEMATECH (2021). While it is plausible that there are previous work that show the robustness of an equivalent method to ours, this section is still important for the self containment of the work.

For this we define the following global definitions which would be used in the following section.

**Definition 12.** *Let* $S = (x_1, y_1), (x_2, y_2), \cdots, (x_{2k}, y_{2k}) \in \mathbb{R}^2$, *where* $\{x \cdot y \mid (x,y) \in S\}$ *and* $\{x \mid (x, \cdot) \in S\}$ *are both of size* $2k$. *Let* $A_1, A_2 \in \mathbb{R}^{(2k) \times k}$ *as follows*

$$A_1 = \begin{pmatrix} 1, x_1, x_1^2, \cdots, x_1^{k-1} \\ 1, x_2, x_2^2, \cdots, x_2^{k-1} \\ \vdots \\ 1, x_{2d}, x_{2d}^2, \cdots, x_{2k}^{k-1} \end{pmatrix}, A_2 = \begin{pmatrix} -y_1 \cdot x_1, -y_1 \cdot x_1^2, \cdots, -y_1 \cdot x_1^k \\ -y_2 \cdot x_2, -y_2 \cdot x_2^2 \cdots, -y_2 \cdot x_2^k \\ \vdots \\ -y_{2k} \cdot x_{2k}, -y_{2k} \cdot x_{2k}^2 \cdots, -y_{2k} \cdot x_{2k}^k \end{pmatrix}. \tag{61}$$

*Let* $A = (A_1 \mid A_2)$ *be a* $(2k) \times (2k)$ *matrix, and let* $\tilde{x}, \tilde{y} = (x_1, x_2, \cdots, x_{2k}), (y_1, y_2, \cdots, y_{2k})$.

In the following lemma we derive conditions that any value that satisfies them is a candidate for an output to $\text{SOLVER}(S)$.

**Lemma 18.** *Suppose that there is* $b = (b_1, b_2, \cdots, b_{2k}) \in \mathbb{R}^{2k}$ *such that the following holds:*

  (i) *It holds that* $A \cdot b = \tilde{y}$, *i.e.* $b$ *is the solution to the constructed linear equation.*

  (ii) *For every* $(x, y) \in S$ *we have* $1 + x \cdot \text{poly}\left((b_{k+1}, b_{k+2}, \cdots, b_{2k}), x\right) \ne 0$.

*Put* $c = (b_1, b_2, \cdots, b_k)$ *and* $c' = (b_{k+1}, b_{k+2}, \cdots, b_{2k})$. *We have that*
$$\forall p \in S : D(c, c', p) = 0.$$

*Proof.* Using the notation from the lemma we obtain

$$A \cdot b = \tilde{y} \tag{62}$$

$$A_1 \cdot c + A_2 \cdot c' = \tilde{y} \tag{63}$$

$$\forall (x, y) \in S : \text{poly}(c, x) - y \cdot x \cdot \text{poly}(c', x) = y \tag{64}$$

$$\forall (x, y) \in S : \text{poly}(c, x) = y + y \cdot x \cdot \text{poly}(c', x) \tag{65}$$

$$\forall (x, y) \in S : \text{poly}(c, x) = y \cdot \left(1 + x \cdot \text{poly}(c', x)\right) \tag{66}$$

$$\forall (x, y) \in S : \frac{\text{poly}(c, x)}{1 + x \cdot \text{poly}(c', x)} = y, \tag{67}$$

where Equation 62 is by property (i) satisfied by $b$, Equation 63 is by the definition of $c, c'$ as $c = (b_1, b_2, \cdots, b_k)$ and $c' = (b_{k+1}, b_{k+2}, \cdots, b_{2k})$, and by the definition of $A$ as $A = (A_1 \mid A_2)$, Equation 64 is by the construction of $A_1$ and $A_2$ in Equation 61, Equation 65 is by adding $y \cdot x \cdot \text{poly}(c', x)$ to both sides of the equation, Equation 66 is by rewriting the right side of the equation, Equation 67 is by dividing both sides of the equation by $1 + x \cdot \text{poly}(c', x)$, which is a legal operation by property (i) satisfied by $b$. $\qquad\square$

In the following lemma we will prove the existence of $b$ as defined in the previous lemma.

**Lemma 19.** *There is $b = (b_1, b_2, \cdots, b_{2k}) \in \mathbb{R}^{2k}$ such that the following holds:*

(i) *It holds that $A \cdot b = \tilde{y}$, i.e. $b$ is the solution to the constructed linear equation.*

(ii) *For every $(x, y) \in S$ we have $1 + x \cdot \text{poly}\Big((b_{k+1}, b_{k+2}, \cdots, b_{2k}), x\Big) \neq 0$.*

*Proof.* **Proving that $A$ is invertible, i.e., proving property (i):**
Recall the definition of $A_1, A_2$, and $A$ from Definition 12. Let

$$\begin{pmatrix} B_1 \\ B_2 \end{pmatrix} = A_1, \begin{pmatrix} B_3 \\ B_4 \end{pmatrix} = A_2, (B_1, B_2, B_3, B_4) \in \mathbb{R}^{k \times k}.$$

Since $B_1$ and $B_2$ are Vandermonde matrix for different $x$-value $\det(B_1), \det(B_2) \neq 0$; the $x$-values are different by the definition of $S$ from Definition 12. Hence, by block matrix properties for $A$ to be invertible it suffices that $\det\left(B_4 - B_2 B_1^{-1} B_3\right) \neq 0$; see Petersen & Pedersen. Let $\tilde{y}_1, \tilde{y}_2, \tilde{x}_1, \tilde{x}_2 \in \mathbb{R}^k$ such that $(\tilde{y}_1 \mid \tilde{y}_2) = \tilde{y}, (\tilde{x}_1 \mid \tilde{x}_2) = \tilde{x}$; the definition of $\tilde{x}$ and $\tilde{y}$ in Definition 12. For every $c \in \mathbb{R}^k$ let $\text{diag}(c) \in \mathbb{R}^{k \times k}$ be a diagonal matrix, whose diagonal is $c$. Let $Y_1 := -\text{diag}(\tilde{y}_1), Y_2 := -\text{diag}(\tilde{y}_2), X_1 := \text{diag}(\tilde{x}_1)$, and $X_2 = \text{diag}(\tilde{x}_2)$ be $k \times k$ matrices. By the construction of $A$ and the previous definitions we have $(B_3, B_4) = (B_1 Y_1 X_1, B_2 Y_2 X_2)$. Hence,

$$\begin{aligned} B_4 - B_2 \cdot B_1^{-1} \cdot B_3 = B_4 - B_2 \cdot B_1^{-1} \cdot B_1 Y_1 X_1 = \\ B_4 - B_2 Y_1 X_1 = B_2 Y_2 X_2 - B_2 Y_1 X_1 = B_2 \cdot \Big(Y_2 X_2 - Y_1 X_1\Big). \end{aligned} \tag{68}$$

By the definition of $S$ in Definition 12 we have $\det\left(Y_2 \cdot X_2 - Y_1 \cdot X_1\right) \neq 0$. Hence, assigning that $\det(B_2) \neq 0$ yields that $\det\left(B_4 - B_2 \cdot B_1^{-1} \cdot B_3\right) \neq 0$, and consecutively that $A$ is invertible.

**Proving property (ii):**
Let $b = (b_1, b_2, \cdots, b_{2k}) \in \mathbb{R}^{2k}$ such that $A \cdot b = \tilde{y}$; there is such value by the proof that $A$ is invertible in the previous part of the proof. Let $c' = (b_{k+1}, b_{k+2}, \cdots, b_{2k})$.

It can be seen that even if there is $(x, y) \in S$ such that $1 + x \cdot \text{poly}(c', x) = 0$, we can add arbitrarily small noise to the $y$ values (without destroying the condition from the previous part), which would tweak $c'$ such that for every $(x, y) \in S$ we would have $1 + x \cdot \text{poly}(c', x) \neq 0$. $\qquad\square$

In the following lemma we combine the previous lemmas to obtain the previously mentioned desired properties of SOLVER.

**Lemma 20.** *We have that $\text{SOLVER}(S)$ is never empty and can be computed in $O\left(k^3\right)$ time.*

*Proof.* Let $A$ and $\tilde{y}$ as defined in Definition 12. By the proof of Lemma 19 we have $\deg(A) \neq 0$. Let $b \in \mathbb{R}^{2k}$ such that $A \cdot b = \tilde{y}$. Let $c = (b_1, b_2, \cdots, b_d)$ and $c' = (b_{k+1}, b_{k+2}, \cdots, b_{2k})$.

By the of Lemma 19 for every $(x, y) \in S$ we have $1 + x \cdot \text{poly}(c', x) \neq 0$. Therefore, since $b$ satisfies all the conditions in Lemma 18, $(c, c')$ is a candidate for an output for $\text{SOLVER}(S)$.

Let $(c^*, c'^*) \in \left(\mathbb{R}^k\right)^2$ such that for every $p \in S$ we have $D(c, c', p) = 0$. By the proof of Lemma 18 we have that $b^* = (c^* | c'^*)$ satisfies that $A \cdot b^* = \tilde{y}$. Since $\deg(A) \neq 0$ we have that there is a unique $\tilde{b} \in \mathbb{R}^{2k}$ such that $A \cdot \tilde{b} = \tilde{y}$. Therefore, $(c^*, c'^*) = (c, c')$, which yields that $(c, c')$ is the unique candidate for a solution to $\text{SOLVER}(S)$.
**Computation time:**
By the previous part of the proof we have that $\text{SOLVER}(S) = (c, c')$. The constructing of $A$ in

Definition 12 takes $O(k^3)$ time. Solving a system of linear equations with $2k$ equations and $2k$ parameters can be done in $O(k^3)$ time. Hence, we can compute $b$ in $O(k^3)$, which yields that $(c, c')$ can be computed in $O(k^3)$ time. $\qquad\square$

### H.3 ANALYSIS OF ALGORITHM 6; FAST-CENTROID-SET

In the following lemma we combine the previous result with some assumptions to obtain the desired guarantees for Algorithm 6.

**Lemma 21.** *Let $P = \{(x_1, y_1), \cdots, (x_n, y_n)\} \subset \mathbb{R}^2$, where $n \geq 2k$, be a set of points with unique first coordinates with non equal zero, and where for every $S \subset P$ of size $2k$ we have $\big|\{x \cdot y \mid (x, y) \in S\}\big| = 2k$. For every $(x_i, y_i) \in P$ let $\tilde{x}_i = (x^* \mid -(x_i y_i) \cdot x^*) \in \mathbb{R}^{2k}$, where $x^* = (1, x_i, x_i^2, \cdots, x_i^{k-1})$. For every $i \in [n]$ let $\big(a_i^{(1)}, \cdots, a_i^{(2k)}\big) = \tilde{x}_i$. Suppose there is $k' \in [1, \infty)$ such that for every $(i, j) \in [n] \times [2k]$ we have*

$$\frac{1}{nk'} \sum_{\psi \in [n]} |a_\psi^{(j)}| \leq |a_i^{(j)}| \leq \frac{k'}{n} \sum_{\psi \in [n]} |a_\psi^{(j)}|.$$

*Let $\delta \in (0, 1), \epsilon = 4^{-k}$, and let $\lambda := \max\big\{\big\lceil \log_{1-\epsilon}(\delta) \big\rceil, 1\big\}$. Let $G$ be the output of a call to FAST-CENTROID-SET$(P, \lambda)$; see Algorithm 6. Let $(c_1, c_1') \in \arg\min_{q \in (\mathbb{R}^k)^2} \ell(P, q)$. Let $\alpha = (2k'^2 + 1)^{2k}$, and suppose that there is $(c, c') \in G$ such that*

$$\sum_{(x_i, y_i) \in P} \big|\tilde{x}_i^T \cdot (c \mid c') - y_i\big| \leq \alpha \cdot \sum_{(x_i, y_i) \in P} \big|\tilde{x}_i^T \cdot (c_1 \mid c_1') - y_i\big|, \tag{69}$$

*which, by the construction of FAST-CENTROID-SET happens with probability at least $1 - \delta$; follows from assigning $P := \{(\tilde{x}_1, y_1), \cdots, (\tilde{x}_n, y_n)\}$ in Lemma 17. Suppose that there is $\rho \geq 1$ such that for every $(x, y) \in P$ we have*

$$\big|1 + x \cdot \operatorname{poly}(c', x)\big| \leq \rho \cdot \big|1 + x \cdot \operatorname{poly}(c_1', x)\big|, \tag{70}$$

*then, we have that*

$$\ell\big(P, (c, c')\big) \leq \alpha\rho \cdot \ell\big(P, (c_1, c_1')\big).$$

*Proof.* We have that

$$\ell\big(P, (c, c')\big) = \sum_{(x_i, y_i) \in P} \left|\frac{\operatorname{poly}(c, x_i)}{1 + x \cdot \operatorname{poly}(c', x_i)} - y_i\right| \tag{71}$$

$$= \sum_{(x_i, y_i) \in P} \left|\frac{\operatorname{poly}(c, x_i) - y_i \cdot \big(1 + x_i \cdot \operatorname{poly}(c', x_i)\big)}{1 + x_i \cdot \operatorname{poly}(c', x_i)}\right| \tag{72}$$

$$= \sum_{(x_i, y_i) \in P} \left|\frac{|\tilde{x}_i^T \cdot (c \mid c') - y_i|}{1 + x_i \cdot \operatorname{poly}(c', x_i)}\right| \tag{73}$$

$$\leq \alpha \cdot \sum_{(x_i, y_i) \in P} \left|\frac{|\tilde{x}_i^T \cdot (c_1 \mid c_1') - y_i|}{1 + x_i \cdot \operatorname{poly}(c', x_i)}\right| \tag{74}$$

$$\leq \alpha\rho \cdot \sum_{(x_i, y_i) \in P} \left(\left|\frac{1 + x_i \cdot \operatorname{poly}(c', x_i)}{1 + x \cdot \operatorname{poly}(c_1', x_i)}\right| \cdot \left|\frac{|\tilde{x}_i^T \cdot (c_1 \mid c_1') - y_i|}{1 + x_i \cdot \operatorname{poly}(c', x_i)}\right|\right) \tag{75}$$

$$= \alpha\rho \cdot \sum_{(x_i, y_i) \in P} \left|\frac{|\tilde{x}_i^T \cdot (c_1 \mid c_1') - y_i|}{1 + x_i \cdot \operatorname{poly}(c_1', x_i)}\right| \tag{76}$$

$$= \alpha\rho \cdot \sum_{(x_i, y_i) \in P} \left|\frac{\operatorname{poly}(c_1, x_i)}{1 + x \cdot \operatorname{poly}(c_1', x_i)} - y_i\right| \tag{77}$$

$$= \alpha\rho\ell\big(P, (c_1, c_1')\big), \tag{78}$$

where Equation 71 is by the definition of $\ell$ in Definition 2, Equation 72 is by reorganizing the expression (taking common denominator), Equation 73 is by assigning the construction of $\tilde{x}_i$ in the lemma, Equation 74 is by assigning Equation 69, Equation 75 is by assigning Equation 70, Equation 76 is by moving $\left| \frac{1 + x \cdot \text{poly}(g, x)}{1 + x \cdot \text{poly}(c', x)} \right|$ inside the absolute value which is a legal operation since $\forall s, t \in \mathbb{R} \ : |s| \cdot |t| = |st|$, Equation 77 is by reorganizing the expression and assigning the definition of $\tilde{x}_i$, and Equation 78 is by the definition of $\ell$ in Definition 2. $\qquad\square$

## I    FULL RESULTS FOR REAL LIFE DATA TESTS

### I.1    FULL RESULTS FOR THE TEST OVER THE DATASET CHEN (2019)

Fig. 9 presents the dataset readings along with the approximation computed in `FRFF-coreset`, Scipy's rational function fitting computed via `Scipy.optimize.minimize`, and a 3th degree polynomial computed using the `numpy.polyfit` function, which minimizes the sum of squared distances between the polynomial and the input. For fair comparison, all three methods have been allowed 4 free parameters. In particular, our rational function and Scipy's rational function have degree 1 in the enumerator and 2 in the denominator (there are 4 free parameters, since the free variable in the denominator is set to 1), while the polynomial is of degree 3.

Observe that in the following examples the 3th degree polynomial yielded a slightly smaller loss than our method, this is in contrast to the case in the example in Fig. 4 where the 3th degree polynomial yielded significantly worse results. We believe that this occurred due the "un-noisy" data corresponding to a more "smooth" function, while in Fig. 4 the function was "less" smooth due to the data generation.

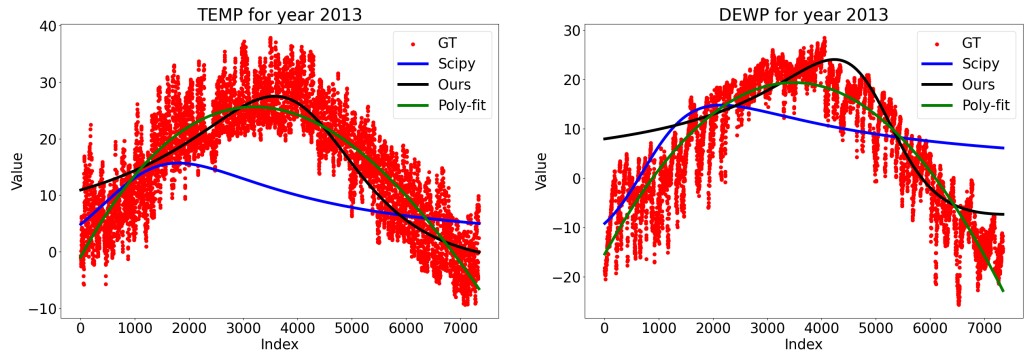

Figure 9: The $n$-signal corresponding to TEMP and DEWP properties from the year 2013 of Dataset Chen (2019), along with three fitted functions: (i) the rational function of degree 2 (i.e., degree 1 in the enumerator and 2 in the denominator) computed in our algorithm `FRFF-coreset`, (ii) the output of a call to `Scipy.optimize.minimize` that aims to fit a rational function of degree 2 to the input signal, and (iii) Polynomial of degree 3, computed using the `numpy.polyfit` function. For fair comparison, all three methods use 4 free parameters.

In this section we include all the results computed for Chen (2019) in section 3.2.

As there is relatively little change in the computational time plot over the sample sizes and features we show the mean results for the year 2016 for the temperature feature in the following table.

| Method | Time (seconds) |
|---|---|
| RFF-coreset | 0.1361 |
| FRFF-coreset | 0.1187 |
| Gradient | 0.0134 |
| $L_\infty$Coreset | 0.8334 |
| RandomSample | 0.0022 |
| NearConvexCoreset | 0.5045 |

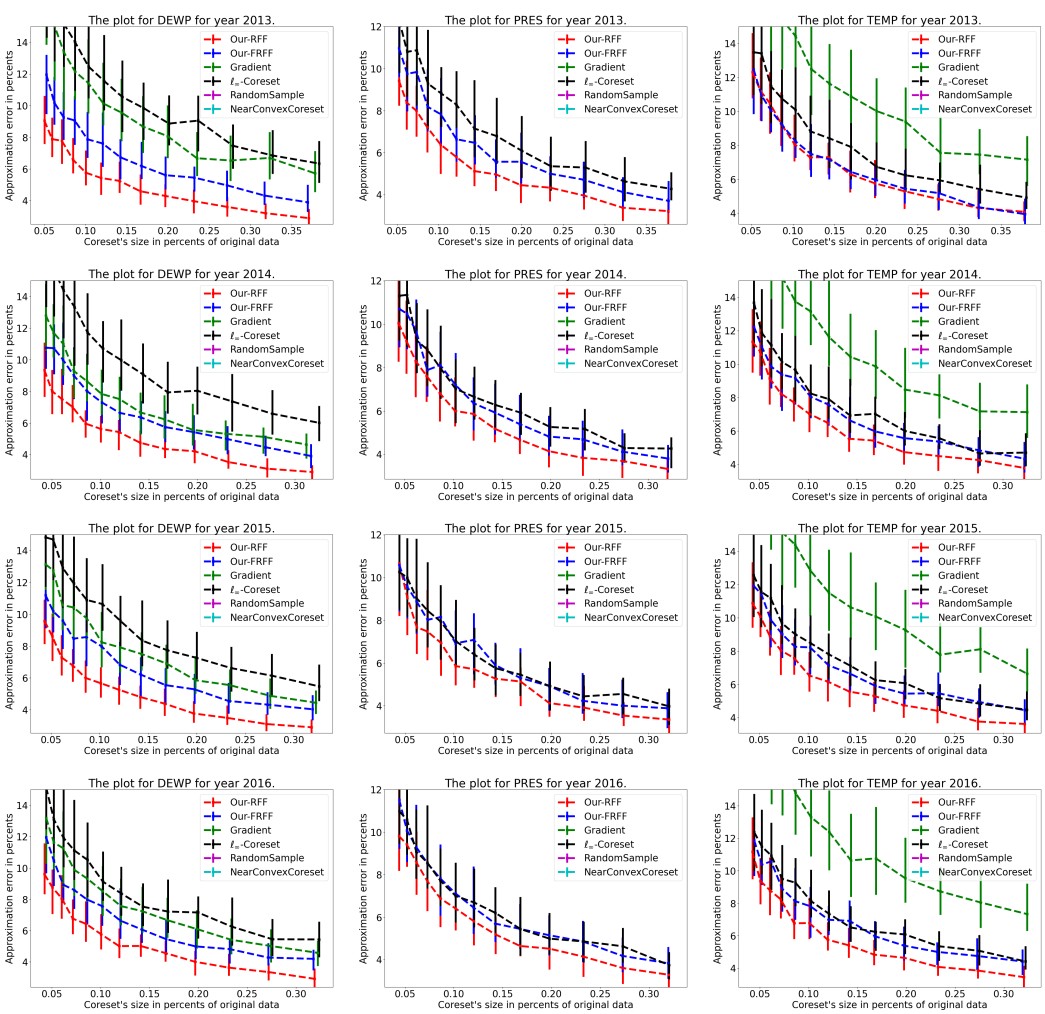

Figure 10: results for the experiment from Section 3.2. The $X$-axis presents, the size of the size of the compression, in percents of the original data, and the $Y$-axis presents the approximation error of each compression scheme, using Evaluation method (i). The plots corresponds: (left to right) to the DEWP, PRESS, and TEMP properties in the dataset Chen (2019), and (top to bottom): to the years $2013, 2014, 2015$, and $2016$. Methods **RandomSample** and **NearConvexCoreset** produced very large errors and are clipped in some cases.

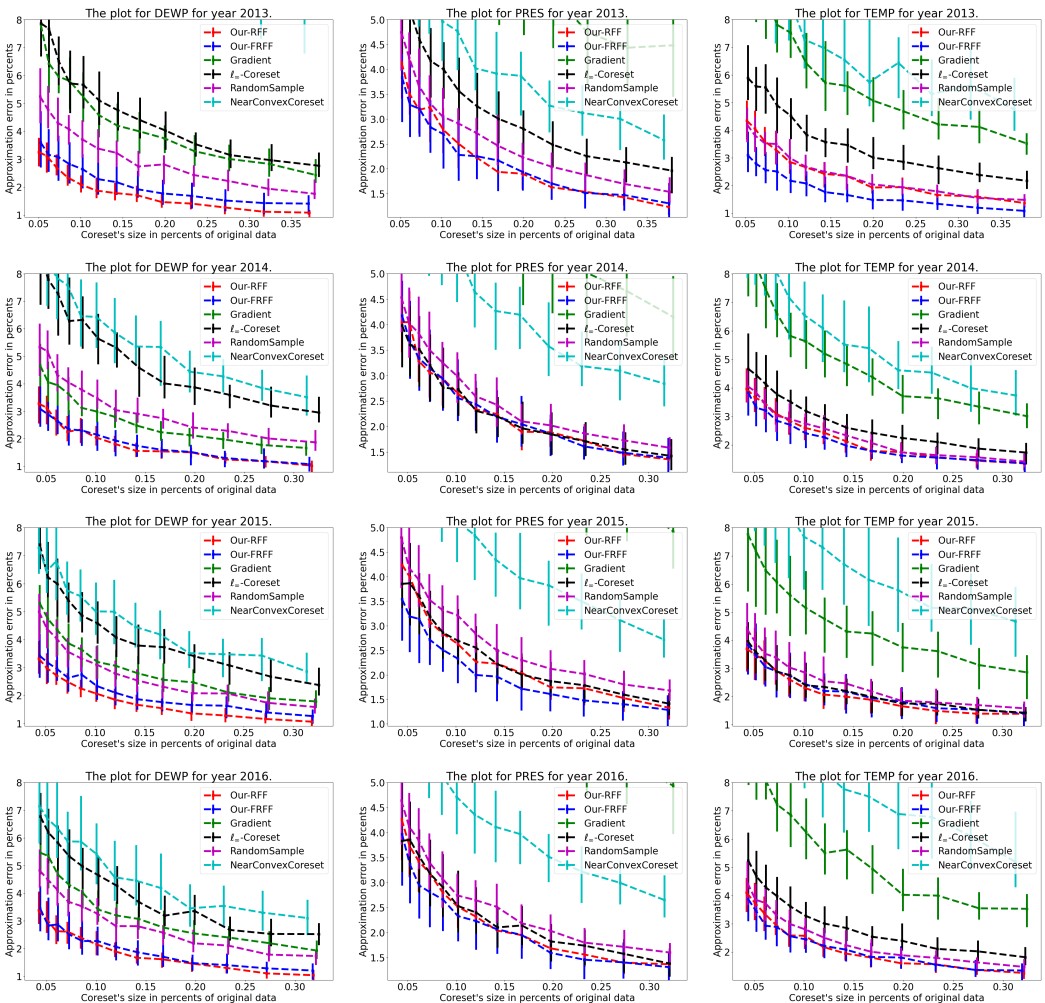

Figure 11: Evaluation of Method (ii), similar to Method(i) in Fig. 10.

## I.2 TEST FOR THE DATASET VITO (2016)

In this test we will use the dataset from Vito (2016) which contains a "yearly measurement of of a gas multisensor device deployed on the field in an Italian city" (cited from Vito (2016)).

Fig. 12 presents the dataset readings along with the approximation computed in `FRFF-coreset`, Scipy's rational function fitting computed via `Scipy.optimize.minimize`, and Polynomial of degree 3, computed using the `numpy.polyfit` function, which minimizes the sum of squared distances between the polynomial and the input. For fair comparison, all three methods have been allowed 4 free parameters. In particular, our rational function and Scipy's rational function have degree 1 in the enumerator and 2 in the denominator (there are 4 free parameters, since the free variable in the denominator is set to 1), while the polynomial is of degree 3.

Observe (as in Fig. 9) that in the following examples the 3th degree polynomial yielded a slightly smaller loss than our method, this is in contrast to the case in the example in Fig. 4 where the 3th degree polynomial yielded significantly worse results. We believe that this occurred due the "un-noisy" data corresponding to a more "smooth" function, while in Fig. 4 the function was "less" smooth due to the data generation.

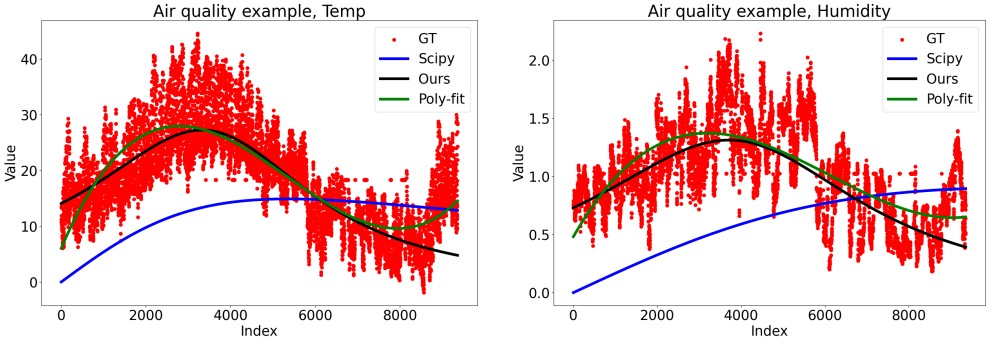

Figure 12: Visual illustration of the $n$-signal $P$ corresponding to Temperature and Absolute Humidity properties of the Dataset Vito (2016), along with two rational function fitting methods: (i) using the approximate rational function computed in our algorithm `FRFF-coreset`, (ii) the output of a call to `Scipy.optimize.minimize` that aims to fit a rational function of degree $2$ to the input signal, and (iii) Polynomial of degree $3$, computed using the `numpy.polyfit` function. For fair comparison, all three methods use $4$ free parameters.

Repeating the test in 3.2 for this data set yields the following plots.

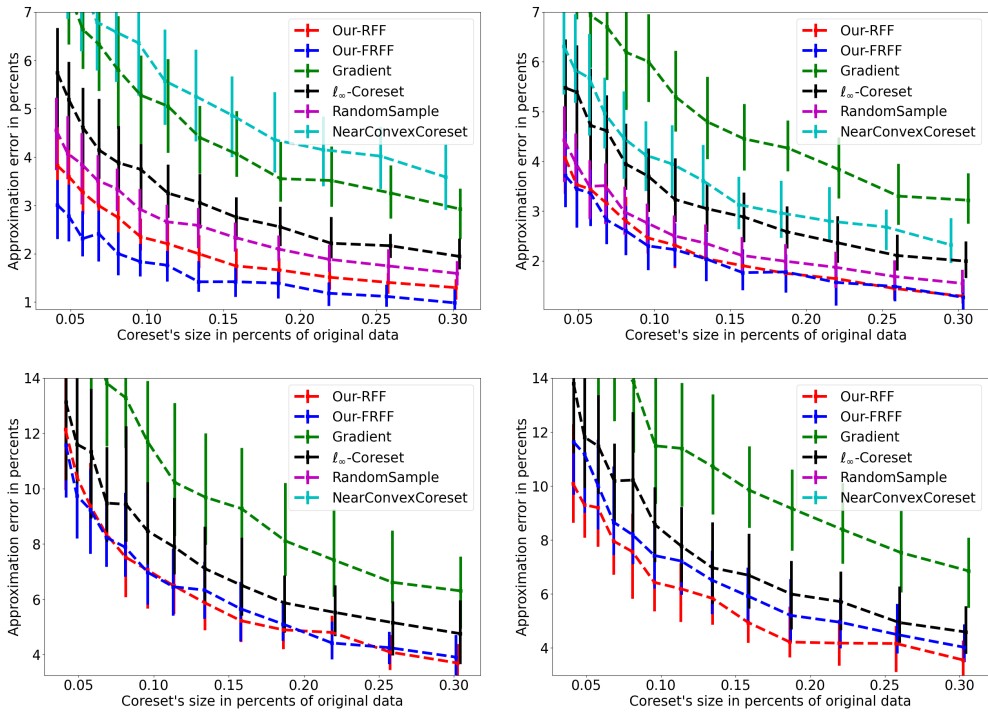

Figure 13: Results for an experiment similar to the the experiment from Section 3.1, but with dataset Vito (2016). The $x$-axis shows the compression ration, with respect to percents of the original data. The $y$-axis shows the approximation error of each compression scheme, for the properties Temperature (left column) and Absolute Humidity (right column). The upper and lower rows present Evaluations (i) and (ii) respectively. The error bars present the $25\%$ and the $75\%$ percentiles. Methods **RandomSample** and **NearConvexCoreset** produced very large errors and are thus in some cases were clipped.

Due to there being a relatively little change in the computational time over the features and sample sizes we show the mean results in the following table, which contains the times only for the Absolute Humidity feature in the dataset.

| Method | Time for the Humidity feature (sec) |
|---|---|
| RFF-coreset | 0.1422 |
| FRFF-coreset | 0.1172 |
| Gradient | 0.0434 |
| $L_\infty$Coreset | 0.7476 |
| RandomSample | 0.0024 |
| NearConvexCoreset | 0.6247 |

## J  ADDITIONAL DISCUSSION

### J.1  FIG. 4

This figure demonstrates that rational function fitting is more suitable than polynomial fitting, for a data relatively normal dataset that is essentially is a set of samples from an exponential function. This also shows that computing any of those fitting functions either on top of the full data or our coreset produces similar results.

### J.2  SECTION 3.1

In this section we demonstrated that our algorithms achieved the best accuracy consistently across the varying signal's length, where RFF-coreset achieved the best performance and is followed by FRFF-coreset. The main results are summarised in Fig. 2. While when considering Evaluation (i) RandomSample, that is essentially a random sample from the input, yielded a good quality, in Evaluation (ii) the values where so large that they where clipped. We, informally, believe that it accrued since the near optimal query considered in Evaluation (i) is not "very far" from any point in the generated data set and as such no point in the data is "very important" for the near optimal query, but as presented in Observation 1 this is not the case for all the queries or even for all the data sets (suppose a dataset where all the points lay on a query beside one point that has $y$-value approaching $\infty$, this single point would be "very important").

In this experiment we obtained significant speed improvement compared to all the other methods beside RandomSample and Gradient, where the later is based on SciPy's library Virtanen et al. (2020) function Scipy.optimize.minimize, and it can be expected since a random sample would obviously be efficient and the function Scipy.optimize.minimize (at least for our parameters and to the best of our knowledge) used optimization methods from Nocedal & Wright (2006) that at some cases can yield very fast a local minimum (but not a global one). Hence, while RandomSample and Gradient had lower running time as observed in Fig. 2 they had significantly worse results.

Observe that while $L_\infty$Coreset which uses the guaranteed approximation for max deviation from Peiris et al. (2021) might seem as a valid heuristic at first glance, our results in Fig. 2 demonstrated that while it was the closest contestant to our methods in terms of quality, it came at the price of a significantly larger running time (it was the only time plot that was clipped). While it is plausible that it follows from our improper parameter tuning, we believe that it comes from the use linear programming solvers which to the best of our knowledge there are no solvers with a running time in $O\left(n^2\right)$; to the best of our knowledge, for the time of writing, the lowest bound is the one in Cohen et al. (2019).

### J.3  SECTION I

In the experiments we obtained very similar results as in the experiment in Section 3.1, which validates our observation above. We note that at some cases FRFF-coreset achieved better quality than RFF-coreset, where this mostly occurred for Evaluation (i). Informally we believe that this occurred since the approximation in FRFF-coreset was very similar to the near optimal query considered, while the BI-criteria in RFF-coreset might had lower loss but was farther from the

query considered. Hence, this difference effected the difference between the results for Evaluation (i) and (ii), where for the later `RFF-coreset` achieved almost consistently better results while in the later lose in many cases. Observe especially the PRES property in Section 3.2, where in Evaluation (i) we had `RFF-coreset` only worse of equivalent results than `FRFF-coreset`, but for Evaluation (ii) we had `RFF-coreset` that gave significantly better results than `FRFF-coreset`.

We note that for TEMP for the year 2013 in Fig. 11, where we used Evaluation (ii), and Fig. 3.1 for the Evaluation (ii) of the Temperature property we had equivalent quality between `RFF-coreset` and `FRFF-coreset`, where the later had better results in some instances.

### J.3.1 FIGURES 9 AND 13 :

In contrast to the example in Fig. 4 in those examples the polynomial fitting with the same number of parameters yielded slightly better results. This is a non surprising result that follows intuitively from considering that as rational functions might yield better fitting for some datasets (especially "non-smooth" Peiris et al. (2021)) for some datasets it is possible to have an opposite effect where polynomial fitting outperforms the rational functions fitting. We believe that this is a specific example of Wolpert (1996) that is related to the well known "No free lunch theorem" Wolpert & Macready (1997).

Nonetheless, even in those examples our approximation outperforms the Scipy's library Virtanen et al. (2020) approximation via `Scipy.optimize.minimize` by a significant margin and gives only slightly worse results than the Numpy's library Harris et al. (2020) polynomial fitting via `numpy.polyfit`.

Observe that this does not invalidate the real-world data experiments in Section 3, since while polynomial fitting yields lower loss we focused on the task of fitting rational functions that is a valid optimization problem on its own. Also we note that our rational function was obtained from `FRFF-coreset` and it is plausible that the optimal rational function (with non constant denominator) would outperform the optimal fitting polynomial function with equal number of free parameters.

### J.4 DISCUSSION ON THE THEORETICAL RESULT

Another contribution of this work is the theoretical results. Our main result which is the basis of our tested methods is Theorem 5, which informally can be summarised as follows.

Given an $n$-signal $P$ we can compress it to a sub-linear size in a quasi-linear time, and the compression with defined probability allows use to compute $\ell(P, q)$, for every query $q \in \left(\mathbb{R}^k\right)^2$, up to a multiplicative factor of $(1 \pm \epsilon)$.

In our eyes another main contribution of this work is the very uncommon framework used in this work, whose overview is in Section 2.2. In particular we used the Merge-Reduce scheme presented in Braverman et al. (2020) to maintain a BI-CRITERIA tree, where it is usually presented to maintain on-line coresets. We also wish to mention the combination of the BI-CRITERIA approximations, where we essentially "trimmed" the approximation to chunks where the data is "well-behaved" and this allowed use to compute a "leaky-coreset" that fails for a bounded number of BI-CRITERIA for each query. this "leaky-coreset" is used in union with an exhaustive search over the "leaks" that are the BI-CRITERIA in the input where the coreset fails.

While those methods might be useful only the researchers in the coreset community, we hope that those novel methods might help build coresets for problems where there was a coreset only for a very specific case, which from a bottom up look can be seen as the focus of this work, i.e., we start with a coreset only for a very restricted case (both in query and form) and we build upon this to obtain an approximation and consecutively an $\epsilon$-coreset without any such limitations.

