# OpenReview forum: "Coreset for Rational Functions"
_ICLR.cc/2023/Conference — Submitted to ICLR 2023_

### Official Review · Reviewer_EJmJ · 2022-10-24

**Confidence:** 4
**Correctness:** 4
**Technical Novelty And Significance:** 4
**Empirical Novelty And Significance:** 4
**Recommendation:** 5

**Clarity, Quality, Novelty And Reproducibility:**

Clarify: Writing needs to be improved, specifically Definition 4 and the empirical section.

Quality: The technical results are solid.

Originality: The considered problem is new and the proposed results are original.

**Strength And Weaknesses:**

Strengths:
- The problem of coreset construction for the rational function is well motivated.
- The idea of combining the construction of a weaker coreset with a merge-reduce tree for maintaining a bi-criteria approximation is novel, and can significantly reduce the total sensitivity of points.

Weaknesses:
- Missing discussion on several related literature. The authors mentioned that "To our knowledge, this is the first coreset for stochastic signals." As I know, time series coreset has been considered before, e.g., [Lingxiao Huang et al., Coresets for Regressions with Panel Data, NeurIPS 2020] and [Lingxiao Huang et al., Coresets for Time Series Clustering, NeurIPS 2021]. I think a comparison with models in these papers is necessary.
- Def 4. The definition comes sudden and lacks explanations. Why including B in the coreset definition? What happens if B is the optimal solution to the fitting problem? How can we use this coreset for acceleration?
- The empirical section is confusing. The authors only refer to figures and do not provide an analysis of the empirical results. At least, the authors should discuss what the advantage of their coresets compared to baselines, provide some concrete numbers or advantage ratios, and summarize the punchlines.
- An advantage of coreset is to accelerate the original optimization algorithm due to the small size. However, I don't see the acceleration analysis in the empirical section.

**Summary Of The Paper:**

The paper considers the problem of fitting a rational function to a time series. The authors provide a coreset construction that gets a time-series and returns a small coreset that approximates its sum of (fitting) distances to any rational functions of constant degree. The size of the coreset is sub-linear in $n$ and quadratic in $\varepsilon^{-1}$. The empirical results on real and synthetic datasets show that the coreset size and the estimation error is smaller compared to the baselines.

**Summary Of The Review:**

The paper considers an interesting problem of fitting a rational function to a time series, and propose a coreset construction framework. The technical results are novel and solid. The writing and clarify can be improved and needs a more careful literature search.

Overall, I do not recommend acceptance now. But I may raise my score if the authors can address my questions well.

---

> ### Author Response · Authors · 2022-11-14
> **Respond to initial review**
>
> # We thank the reviewer for the review.
> ### We will upload an updated version followed by the reviews.
> ### Regarding the weaknesses mentioned:
>
> ## Missing discussion on several related literature:
> It seems that we had bad wording in the sentence "To our knowledge, this is the first coreset for stochastic signals."  that caused this confusion.
>
> The focus was meant to be on the stochastic, since we are aware of many coreset for time series, as mentioned in your answer, while of no coreset inspired from stochastic models.
>
> Regarding the papers mentioned in your review, they seem to fall into clustering and subspace approximation, as mentioned in the line following the quote cited by the reviewer, which is "Most existing coresets are motivated by problems in
> computational geometry or linear algebra, especially clustering, and subspace approximation."
>
> In Section 1.4 (the section from from which the sentence was cited) we included a comparison to previous work on common works from the coreset literature that your suggested works seem to fall into.
>
> **We would be glad to include a more in depth discussion if there seems to you that there is a discussion missing from this section, and for suggestions on the wording of the sentence cited by the reviewer.**
>
> ## Definition 4:
> **Thank you for the comment, we will include additional explanations prior to Definition 4.**
>
> The addition of the approximation B to coreset follows from as mentioned in Section 1.2. that "Unfortunately, similar to Rosman et al. (2014a), the RFF problem with general input has no coreset which is weighed subset of the input; see Claim 1."
>
> Hence, similar to Rosman et al. (2014a) we have included the approximation as part of the coreset; in Rosman et al. (2014a) they managed to obtain a weighed set which is a coreset.
>
> Indeed, after rereading there is a good point for an explicit mentioning and this was addressed.
>
> ## Regarding the questions:
> ### The inclusion of the approximation
> The inclusion of the approximation is to solve the counter example in Claim 1 of the paper; done to allow a generalization of the methods from Rosman et al. (2014a) to allow a coreset construction from the approximation.
>
> **In the case that the approximation is the optimal solution to the fitting problem:**
> There is no particular problem; while the main focus of the paper is to compute this approximation, hence, if we are given the optimal solution the coreset construction would be essentially calling Algorithm 1 that follows from Rosman et al. (2014a).
>
> It is non-trivial how we can use the coreset for provable acceleration, while the evaluation can be done efficiently.
>
> We hope that this result could be improved in future work to compute a solution over the coreset, however, when we aimed to do this we obtained different optimization problems that we are not aware of a polynomial solution for.
>
> ### The empirical section is confusing:
> Indeed, we have "over-trimmed" the discussion due to the space limitation.
>
> We will add a reference to a more in depth discussion at the appendix aiming to to solve the deficiencies raised.
>
> ### Acceleration analysis:
> Since the coreset is aimed for compression of the problem, and we do not solve the problem over the coreset we do not have an acceleration analysis, but rather only compression analysis.
>
> Nonetheless, we hope that this result that yields an optimization problem with fewer coefficient would inspire solutions with guarantees by applying polynomial solution over the coreset.

---

### Official Review · Reviewer_Vhf7 · 2022-10-24

**Confidence:** 3
**Correctness:** 3
**Technical Novelty And Significance:** 3
**Empirical Novelty And Significance:** 2
**Recommendation:** 5

**Clarity, Quality, Novelty And Reproducibility:**

Clarity: The introduction is generally well written. Things get murkier in the technical content, with relatively little intuition and explanation given around the technical definitions, which would have helped a reader such as me who is not familiar with the prior work on this problem (for example, why does the definition of "ratio" in RFF (def 2) increase the degree of the polynomial in the denominator from k to k+1? What is the intuition behind the non-standard definition of coreset in Def 4, which uses both a weighted set and a bicriteria solution to approximate the loss?) The description of the algorithm was also rather on the technical rather than intuitive side, and unfortunately I couldn't glean what are the main insights driving the algorithm.

Small comments:

- Section 1.2 states that: "this is due to the three main coreset properties: merge and reduce [citations]" - what? What are the three properties?

- In Figure 2 there is some mismatch between the coreset sizes stated in the textual caption and the figure captions.

Reproducibility: I wouldn't deem the results reproducible unless code is released, since the algorithms are stated toward analysis rather than implementation (there are parameters and O-notations etc). The description of the experimental design was also not entirely clear to me (see above).

Novelty: I am not sufficiently familiar with the area to say.

**Strength And Weaknesses:**

Strengths: The problem studied seems meaningful and interesting, and the proposed solution also seems to be potentially interesting and contain new ideas. I am not familiar with prior work on this problem, but it seems to me an intriguing direction that restricting the solution space to degree-k rational functions could give rise to coresets of size proportional to only k, and it was interesting to me that the useful notion of bicriteria approximation here was a piecewise approximation (and not, say, a single rational function with degree somewhat larger than k, which would have been my first thought). Also, the problem and the algorithms and proofs are defined rigorously and in detail, which I appreciated.

Weaknesses: the two big ones for me are the usability of the defined notion of coreset, and the experimental design; I hope to see some clarification from the authors on both. In more detail:

1. How is this non-standard notion of coresets useful for approximation?Coresets are usually defined (for example in the context of clustering) as weighted subsets of the original set of points. This is useful since we can apply any exact or approximate (albeit slow) algorithm to the coreset instead of the original problem (for example, solve k-means on a subset instead of the whole set), and obtain a fast approximate algorithm.

Here, the coreset is not a weighted subset, but a more general object (a combination of a weighted set and a piecewise rational function). Do we have algorithms to find an (exactly or approximately) optimal degree-k rational approximation for this object? I am not familiar with this space of problem and possibly I am missing something basic here, but I could not find it mentioned in the paper. This is important, since without such an algorithm it is not clear that this notion of coreset is of any use: once we constructed the coreset, what do we do with it to find a good approximation for the original problem? Typically, a coreset leads as a corollary to a theorem that states "the original problem can be solved up to X approximation in running time Y", the proof being by constructing the coreset and then running on it a known (exact or approximate) algorithm. Can you state such a theorem for RFF as a corollary of Theorem 1?

2. Experimental design:Related to the above, I am not sure what the evaluation in the experiments is actually measuring, and whether it is actually constructing a degree-k rational approximation of the input signals, as Def 2 requires. The "Evaluation" paragraph in section 3 is too unclear and it would help to explain better and in more detail what is going on there. I understand that (as the paragraph mentions) finding the groundtruth solution to each signal is too computationally costly, but this is not necessary: the experiment I would have liked to see is using each of the evaluated coresets to construct a degree-k rational approximation, and measure the approximation loss (sum of differences) between the original signal and the approximation, and then compare the losses across the evaluated methods. This would have also answered my question about using the coresets to actually obtain an approximation.

**Summary Of The Paper:**

The paper is concerned with the problem of approximating a time-series by a rational function of degree at most k (which is the ratio of two polynomials of degree at most k each), and more specifically, with constructing a coreset for this problem. A coreset of a given problem instance is an instance of a much smaller related problem (typically a small weighted subset of the original instance), which approximately preserves the value (or loss) of every solution to the original problem. This is useful since an approximate solution can then be computed on the small coreset instead of the original large problem. This paper suggests a coreset of size polylog(n) * k^(loglog n) / epsilon^2, that preserves the approximation loss up to 1+/-epsilon, though the notion of coreset is non-standard (not a weighted subset).

Technically, the construction relies on the well-known sensitivity framework for constructing coresets. This framework is based on importance sampling, where the coreset is formed by sampling random points from the original problem with probability proportional to a measure called "sensitivity", which measures how important is the point for solving the problem on the given instance. The sensitivities are estimated by an initial coarse (bicriteria) solution to the problem. The majority of technical effort in this paper is constructing the initial bicriteria solution, where the notion of bicriteria here is that the time-series is approximated by a piecewise rational function degree k instead of just one such function.

**Summary Of The Review:**

The paper studies a meaningful problem and seems to contain interesting ideas. However, there are some gaps and unclarities in the overall reasoning (how is the non-standard notion of coreset defined here used for fast approximation) and in the experimental design, that I would like to clarify before recommending to accept. I also found the technical writing rather unapproachable and was unable to verify correctness, though this may be partly because I lack background on the topic.

---

> ### Author Response · Authors · 2022-11-14
> **Respond to initial review**
>
> # We thank the reviewer for the review and glad that you have liked the direction of the paper.
> ## Weaknesses.
> ### Point 1.
> Indeed, it is non-trivial how we can use the coreset for provable acceleration, while the evaluation can be done efficiently.
>
> **We will emphasize this point better to avoid such confusion.**
>
> In other words, in order to solve the problem we are only aware of the method of "uncompromising" the coreset to $O(n)$ points and solve the problem on the "uncompressed" data.
>
> **We hope that since this result in an optimization problem with fewer coefficient this result could be later improved to derive a guaranteed approximation; when we aimed to do this we obtained difficult optimization problems that we are not aware of a polynomial solution for.**
>
> Nonetheless, this compression allows to store and to transmit a compressed representation of the data, e.g., as in video compression where AV1/H.256/H.246 allow small representation of the data but in many tasks the video is uncompressed before applying the solution.
>
> This point was also stated at **answering queries** section 3 of [1].
>
> We believe that this *disadvantage* can also be seen as one of the strengths of the paper, since one of the major novelties (and difficulties) of this work (in our eyes) is the computation of the coreset without relying on such result.
> We hope that this side-stepping would inspire different coreset constructions, even if they would be mostly for efficient storage/transmission.
>
> [1] Feldman, Dan. "Introduction to core-sets: an updated survey." arXiv preprint arXiv:2011.09384 (2020).‏
> ### Point 2.
> The goal of the evaluation is to evaluate the quality of the compression as stated in the evaluation.
>
> **We will update the writing to emphasize this point better, and would be glad if you could check that it answered this problem.**
>
> ## Clarity:
> We will upload in the following days a version aiming to address the clarity problems raised in your answer and other reviewers claims.
>
> ## Small comments:
> ### Section 1.2
> **Thank you for noticing the problem here**
>
> Indeed we had bad writing here that caused this confusion, the citations should have been at the end.
> The correct wording is: "this is due to the three main coreset properties: merge, reduce, and that a coreset approximate every model, and not just the optimal model [citations]".
>
> In other words the properties are  "merge", "reduce", and "the approximation of every model".
>
> **The wording was fixed, and hopefully this clarifies this confusion**
> ### Figure 2
> **Thank you for noticing**
>
> This small difference is since the coreset size is computed after accounting for repetitions, i.e., if we chose a point twice we can simply update its weight and not send it two times, while the construction requires sampling $x$ points.
> Hence, there is a noise in the coreset size, and while it is usually small and the size is averaged across the tests like you have noticed it can still be present at the results.
>
> In other words, there is noise at the coreset size due to repeated samples of the same element.
>
> **We will add clarification that there can be small noise in the sizes due to noise at the size required to store the samples.**
> ## Reproduciblity:
> As mentioned in the **REPRODUCIBILITY STATEMENT**, we commit to publish the code for all of the tests (or part of them) upon acceptance of this paper or upon reviewer request.
>
> **In case you would like us to release the code at the moment we would be glad to.**

---

> > ### Comment · Reviewer_Vhf7 · 2022-11-17
> > **Response**
> >
> > I thank the authors for their response. This helps clarify things, though I still believe that both of the matters I have mentioned are issues that should be considered. I understand now that your result is a compression result, that does lead to improved running time. However, this is quite at odds with the writing of the manuscript: for example, the first sentence introducing coresets in the introduction is "Why coresets? The main goal for constructing a coreset is to be able to compute an optimal model or its approximation much faster", which certainly leaves the readers to think that the paper would be about faster running time, and the rest of the introduction does not seem to correct this impression.
> >
> > Regarding evaluation, I understand what the plots are measuring now. It is perhaps not the best methodology to evaluate the coreset on random solutions instead of approximately optimal solutions, though given that you deem solving the problem too slow (even approximately?) and that the corset does not yield an efficient algorithm, I am also not sure what measure this could be replaced with.

---

> > > ### Author Response · Authors · 2022-11-17
> > > **Fix of the problems raised (thank you for the concrete example)**
> > >
> > > # We thank the reviewer for the respond and the concrete example.
> > > ## Part 1:
> > > Unfortunately we have misjudged the previous point as more of a question than a problem.
> > >
> > > Indeed this point on our coreset should be emphasized.
> > > In the original version we have discussed motivation for coresets in a too general way that due to including non-relevant properties can indeed cause confusion, especially considering the space limitations that would limit our ability to correct such confusion.
> > >
> > > We have rewritten this paragraph, **whose name was changed to why such coreset to reflect this change**,and moved after explanation on our coreset, to emphasis the usefulness of our coresets and not coresets in general.
> > >
> > > Given the minor space gained after this change we have slightly expanded the explanation in **Coreset for rational functions**.
> > >
> > > **We have updated the paper, and hopefully this would solve this point**
> > >
> > > ### We want to emphasis that we have not meant to dismiss this problem, but rather though it was a question.
> > >
> > > **The updated paragraph is as follows:**
> > >
> > > **Why such coreset?**
> > > A trivial use of such coreset is data compression for efficient transmission and storage.
> > > While there are many properties of coresets as mentioned at Feldman (2020), some of them are non-immediate from our coreset; see Feldman (2020) for a general overview that was skipped due to space limitations.
> > > Nonetheless, since optimization over the coreset reduces the number of parameter, we hope that in the future there would be an efficient guaranteed solution (or approximation) over the coreset.
> > > Moreover, since this coreset does support efficient evaluation, we hope this coreset would yield an improvement for heuristics by utilizing this fast evaluation.
> > >
> > > ## Part 2:
> > > We are glad that our explanation made things more clear.
> > >
> > > Regarding this point we wanted to clarify that we have two evaluation methods, where indeed one of them can be seen as random queries, **but the the other evaluation option, Evaluation (i), considers the quality for a query that aims to approximate the optimal query (compute a near-optimal query in the original text)**.
> > >
> > > This seems to us very similar to the evaluation to approximately optimal solutions mentioned.
> > >
> > > **We would be glad if you could clarify if we have missed something from the methodology suggested, or if we should emphasized this more in the Evaluations paragraph/experiments.**
> > >
> > > *We wanted to elaborate that Evaluation (ii) considers the largest discrepancy between the ground truth and the coreset over the random queries in an aim to empirically approximate the bound over least satisfied query, which is inspired by the definition of $\varepsilon$-coreset at the paper.*
> > >
> > > ## We would be glad for your response on whether this clarifies/answers the queries, and to answer additional problems in your eyes.

---

### Official Review · Reviewer_oFkV · 2022-10-27

**Confidence:** 2
**Correctness:** 2
**Technical Novelty And Significance:** 4
**Empirical Novelty And Significance:** 3
**Recommendation:** 3

**Clarity, Quality, Novelty And Reproducibility:**

I've already written a bunch about a lack of clarity leads to a certain lack of quality in this paper.

The paper definitely seems novel and significant enough for publication though. The results as written are compelling.

I devote the rest of this space to a list of typos.

### Typos

1. [Page 1, background paragraph] "the references therein" instead of "reference therein"
1. [Page 2, section 1.2 first paragraph] Say "Informally, given an input signal of d-dimensional points" instead. Also, why not just say 2-dimensional instead of d-dimensional?
1. [Page 2, section 1.2 last paragraph] If there's no coreset which is a weighted subset, then it makes sense to use a bicriteria approximation in addition. But, why restrict yourself to consequtive integers in the first dimension then? There's nothing wrong with assuming that to be the input data given to you, but the flow of this paragraph's logic / justification doesn't quiet stand up.
1. [Figure 4] Not a big deal, but this might read a bit more clearly if we also had an error plot, showing if any approximations are especially accurate or innacurate, and where those accuracies/innaccuracies are.
1. [Throughout the paper] Feldman & Langberg 2011a and 2011b are the same paper. This is cited a lot, so it's helpful to realize they're the same paper.
1. [Page 3, section 1.4 last paragraph] Say "believe" instead of "expect", or some other weaker language. This sentence comes off kinda weird with "expect" there.
1. [Page 3, section 1.4 last paragraph] I have no idea what the "have few dependent recursive functions ..." phrase means. No clue what message you're trying to send with this sentence.
1. [Page 3, section 2.1 first line] I'd probably say "set of k-dimensional" instead of "union of k-dimensional". Not sure what's getting union'ed here.
1. [Page 4, section 2.1 just before definition 2] "Projection of f onto P" makes _a lot_ more sense than the "Projection of P onto f". f exists where P doesn't, so we project f onto P by evaluating f on that interval.
1. [Page 4, definition 2] Why define a rational function this way? Why not just leave it be a ratio of two arbitrary polynomials. Justify this in text.
1. [Page 4, paragraph after definition 2] Remove the comma after "Definition 2"
1. [Page 4, paragraph after definition 2] "approximation _as_ Feldmen & Langberg (2011) defined"
1. [Page 4, paragraph after definition 2] Expliciltly site section 4.2 of Feldmen & Langberg (2011) and explicitly use the language "bicriteria approximation"
1. [Page 4, definition 3] Consider using the language "consecutive intervals" instead of "consecutive sets"
1. [Page 4, definition 3] Missing subscript on $P1$
1. [Page 5, first line] Use the standardized notation $n^{o(1)}/\varepsilon^2$ instead of this $n^{\phi}$ for any phi notation. It's standard in TCS
1. [Page 5, first line] If it's with probability $1-\delta$, then $\delta$ should appear in the space complexity
1. [Page 5, first line] Remove "; see Definition 4", that definition was 2 lines ago.
1. [Page 5, first paragraph] It's algorithm 2, not algorithm 1
1. [Page 5, first paragraph] I'd say something like "probability of placing a point $p \in P$ into the samplet set C"
1. [Page 5, first paragraph] Write out the formal probability proportionality, so it's more clear. It's quick to state something like $Pr[\text{picking } p_i] \propto D(q_i, p_i)$.
1. [Page 5, second paragraph] To clarify the break in the sentence, add "it has" after "$\beta$ child nodes, " and before "$O(\beta^i)$ leaves.
1. [Page 5, third paragraph] Don't use $C$ here. $C$ already means the subsampled points returned in the coreset.
1. [Page 5, third paragraph] Don't change the meaning of $i$ from the second paragraph to the third. Keep $i$ as the level, and let $j$ denote an element $B_j \in B$.
1. [Page 5, third paragraph] The $(0,r_i)$ notation is really unclear. As written, I think means that $B_i$ has $r_i$ subintervals which exactly interpolate the given dataset $y$, which I don't think is the case.
1. [Page 5, fifth paragraph] Lemma 8 isn't anything that the reader knows. It's burried in the appendix. Say somethign like "this can be solved using <brute force / greedy search / whatever>, with details shown in Lemma 8".
1. [Page 5, fifth paragraph] **As far as I understand, the input signals $C$ are arbitrary subsets of $\{B_1,...,B_{\beta}\}$ of size $\beta-6k+3$. Not consecutive. How does this mesh together with point (i) from the third paragraph on this page?**
1. [Page 5, fifth paragraph] Add "approximation" after "bicriteria"
1. [Page 5, last paragraph] Add "time" after "and thus the running"
1. [Page 5, last paragraph] Again, please abandon this "for all $\varepsilon>0$ notation". There's (in my opinion) a better notation to say what you're trying to say.
1. [Page 6, "optimal soloution" paragraph] The phrase "that minimizes 2" -- I've got no idea what 2 is here. There's no equation 2 in the body of the paper.
1. [Page 6, "efficient $(1,\beta)$" paragraph] "relatively fast" doesn't give me a good intuition. Give me a big-oh notation.
1. [Page 8, list of implimentations] Is one of these analogous to Lemma 8's algorithm? If not, would it be possible to add that to the experiments. If so, could you lable which one that is?
1. [Page 8, section 3.1 results] You don't explain what the error bars are here
1. [Page 8, "dataset" paragraph in section 3.2] Missing parentheses on the citations after "Beijing Air Quality Dataset" and "UCI Machine Learning Repository"
1. [Page 8, section 3.2 results] If you use 25th and 75th quantiles, you should probably use median for the central line instead of an average. Probably won't change the data much, but it's a bit more emotionally consistent.
1. [Page 9, section 3.3 results] Figure 4 shows a brutally hard adversarial edge case for polynomial, but no such example for rationals. It's not fair to point at Runge's phenominon and just say that polynomials are worse at fitting time series than rationals. There's arguments that can be made there, but this is unreasonable.

**Strength And Weaknesses:**

[edit: fixed the block quote in _Part 1_ to correctly separate my writing from the paper's]

This paper gave me some whiplash. It's got some sections written really clearly, and some which are brutally opaque. I wrote a long summary for this paper because it took me a long time to just understand the algorithm and construction of the paper. In the places it's opaque, I'm not comfortable saying that I believe it's correct. Because of concern about correctness, I'm not comfortable accepting this paper.

### Part 1: Introduction and Motivation

The paper starts with an interesting and compelling story about why MSE is a bad loss metric for fitting time series data: often time series models like the "auto-regressive" model have variance that grow exponentially, so minimizing the MSE is just fitting the last couple data points. So, we should consider fitting in some other metric that. The authors then define a generating function and seem to fail to give any fundamentally new metric, and end up using $\ell_1$ error instead of MSE (i.e. $\ell_2$) error.

I'd like to know what the authors were going for, because minimizing the $\ell_1$ loss would also just fit the last few data points from an exponentially growing time series.

Regardless, fitting a time series in $\ell_1$ norm is a perfectly well motivated problem, so it's not a huge problem, but it's certainly confusing. I really wished they stuck the landing and suggested a new way to talk about error in time series data though.

This question of error metric is a great example of how Section 1 of the paper is a mix of extremely clear and very confusing writing. Beyond this one example, there's also a few times where the authors just introduce a term from the literature without any explanation. As someone not familiar with this particular term from the prior work, I found the following line pretty funny:
> This is due to the three main coreset properties: merge and reduce.

I have no idea why "merge and reduce" is three properties, and this is not explained in the introduction. The authors similarly do this with "$(\alpha,\beta)$-approximation", which ends up being super important in this paper, and it was only by digging in the prior work that I figured out that "$(\alpha,\beta)$-approximation" means _bicriteria approximation_. "Merge and reduce", and bicriteria approximation are both heavily used in this paper, and not clearly saying what these intuitively are in the introduction is a huge misstep that really hurts the legibility of the paper.

## Part 2: Describing how the Algorithms Work

Page 5 of this paper is a summary of how they construct the bicriteria approximation and the subsampled dataset that comprise their coreset. This page took me an immense amount of effort to understand _what the algorithms intuitively are doing_, and I don't have an intuition for why these algorithm construct a near-optimal coreset. **The clarity issues from the introduction could be fixed with a minor revision, but the issue of clarity on this page is more brutal.**

This page is full of high-level technical descriptions of what the various parts of various algorithms do. I'm not going to belabor a huge list of points that confused me, but I will give a demonstrative example. I'll include broader frustrations about this writing in my list of typos & suggested edits.

For my example, consider the middle line from the first paragraph in the section "Algorithm 3: the merge-reduce step", which reads:
> This is by computing the following for every possible subset $C \subseteq B$ of size $\beta − 6k + 3$

I can understand that this algorithm is given a set $B$, and it can search over subsets of size $\beta - 6k +3$. I have no idea why $-6k+3$ is meaningful, what's it's used for in the analysis, if the numbers $6$ and $3$ are important, or anything else about it. This whole page is full of instructions whose value is completely lost on me.

### Part 2 again: The (hidden?) Strengths

The frustrating layer under this somewhat brutal writeup, was that as I progressed through page 5 of the paper, I could see there's a careful and clever design under all of this. The details seem precise. When I wanted to understand a claim in a bit more detail, there were easy-to-find formalisms in the appendix. I am emotionally convinced there's a cool result with a neat and careful construction under this all. I just haven't been given a correctness argument that I could review in the body of the paper.

I think the authors could reduce the formal specification of the algorithms on page 5. Then, remove the formal algorithm from the body of the paper, and replace it with an informal pseudocode that skips over implementation details and covers more high-level ideas. Then, with the recovered space, justify more clearly why the construction is accurate with high probability.

### Part 3: Experiments

The experiments are decently cool. It's a hard problem with an exponential dependence on the degree of the rational function, so the experiments are just stated for degree 2 rational functions. But, for this setting, the experiments are convincingly interesting, and show that the coresets can be pretty efficient for real data (figure 3 on page 9).


**Summary Of The Paper:**

This paper studies the problem of building a coreset for fitting rational functions to time series data. In particular, suppose $y_1, ..., y_n \in\mathbb{R}$ is a time series. Then, in rational function fitting, we want to find a rational function $r$ of degree $k$ such that $r(i) \approx y_i$. We formally measure error with the $\ell_1$ metric, the sum of absolute errors.

This paper isn't as much about solving this problem, as it is about _compressing_ the problem, so we want to find a small data structure such that all rational functions $r$ have their loss with respect to $y$ preserved within a multiplicative $(1\pm\varepsilon)$ factor. So, in order to approximately solve the full problem on all $n$ time points, it suffices to minimize over rational functions on this smaller coreset.

For constant failure probability, the paper builds such a coreset in $\tilde{O}(2^{O(k^2)} n^{O(\frac{k}{\log \log n})})$ time and it uses $\tilde{O}(\frac{1}{\varepsilon^2} \text{poly}(k) \log^{O(\log(k))}(n) )$ space. For constant degree $k$, this is $O(n^{1+o(1)})$ time and $O(
\frac{\text{polylog}(n)}{\varepsilon^2})$ space.

The coreset is constructed in two parts:
1. First, they use a bicriteria-approximation algorithm to partition the time domain into $O(\log n)$ subintervals, and approximately fit a rational function $r_i$ to each subinterval. _This is an approximation because each $r_i$ is an approximate minimizer, and is bicriteria because it returns $O(\log(n))$ rational functions instead of just one._
2. Then, since this bicriteria approximation is not accurate everyone, a randomized "sensitivity sampling" stage explicitly stores the time points where both (i) the bicriteria approximation is inaccurate and (ii) where a rational function could be able interpolate that point.

The loss of an arbitrary rational $r$ function against the full dataset is then approximated by sum of the loss of $r$ on the sampled points and the sum of the distance between $r$ and $r_i$ on the subintervals created by the bicriteria approximation. The authors say that most of the technical novelty and effort goes into building the bicriteria approximation.

**Summary Of The Review:**

The paper seems probably cool, and I want to like it, but the authors didn't give me good reason to think their claims are correct. As such, I reject the paper. A rewrite of the paper could definitely be published though (supposing the results are correct).

---

> ### Author Response · Authors · 2022-11-14
> **respond to initial review (part 1)**
>
> # We thank the reviewer for the review and glad that you have liked the direction of the paper.
> We will upload an updated version that would hopefully solve most of the writing problems, keeping your very detailed review in mind.
>
> # Regarding the comments in the Strength And Weaknesses:
>
> ## Part 1:
> It seems that there we had some miss communication at the part in question (From Auto-regression to Rational functions.).
>
> The different metric is the fitting to the generating function, and not to the original data.
> This solves the problem of "the $\ell_1$ loss would also just fit the last few data points from an exponentially growing time series.", since we apply the $\ell_1$ after the generating function and not the original data.
>
> We will update the writing to make this point more explicit.
>
> ## Part 2:
> We are glad that "When I wanted to understand a claim in a bit more detail, there were easy-to-find formalisms in the appendix."
>
> Regarding this specific part, indeed after rereading there was a significant place to improve the writing in this section.
> We will update a revision of the paper soon days keeping in mind your detailed Typos and suggestions that would significantly help improve the lacking writing here.
> Since this seemed in our eyes as your major complaint we would be very glad if you could re-read this in the updated version after we will upload the revision.
>
> We have implemented differently the suggestion at the last paragraph of "Part 2 again: The (hidden?) Strengths", where we included very high level intuition in the appendix.
>
> We would like to comment that since this part discuss the theoretical results that are of mathematical nature it is hard to properly explain the results without delving into the details; as commonly said "The God is in the details".
>
> **We would be glad to revise this section if it is still too technical in your eyes or this reference to the intuition in the appendix is not sufficient.**
> ### Part 2:
> We thank the reviewer of the uncommonly detailed list of suggested changes, which would help clarify the design that was more hidden at page 5 and hopefully solve the clarity problems at page 5.
> In summary, we agree with most of the suggestions and implemented them.
>
> The only points not yet implemented are point 9 and 34.
>
> We are unsure about point 34, it was written at Section 3.1.  **Results** what the error bars meant.
> We would be glad if you could clarify if the writing was unclear or if it should have been placed at a different place.
>
> **Point 9:**
>
> While there is validity to point 9, it seems that this would cause inconsistency with previous work on coresets.
>
> More specifically, in Algorithm 2 of Feldman et al. (2012) it was defined the projection of points to a function and not the other way around as suggested.
>
> **We would be glad to implement this point, however we wanted to validate beforehand considering the possible inconsistency with the previous work used.**

---

> > ### Author Response · Authors · 2022-11-14
> > **Response to initial review (part 2, list of the typos fixed with comments)**
> >
> > # List of the typos fixed with comments.
> > - (1.) Implemented.
> > - (2.) Implemented.
> > Regarding the question "why not just say 2-dimensional instead of d-dimensional?", indeed in our case 2-dimensional would suffice, however, in many instances coresets are for d-dimensional data and as such we wanted to give a more general version that correlates to the general coreset literature (since we begin at a general explanation).
> > **Does this seem over generalization in your eyes?**
> > - (3.) Indeed it seems that the flow was problematic. The meaning was supposed to be that we include this assumption from Rosman et al. (2014a) to solve this problem.
> > Indeed this should have been stated more explicitly, we fixed the writing here to be more explicit.
> > Hopefully the updated version would convey reasoning better, and we would be glad if you could check if it answered the question.
> > - (4.) Indeed this is a very good suggestion, which will help demonstrate the differences between the methods that we will be glad to implement and add to this to the Figure.
> > - (5.) Thank you for notifying us kindly, indeed we had a problem with those citations.
> > This problem was fixed and we have proof-read the citation for problems.
> > - (6.) We have changed the writing to be less strong.
> > - (7.) Indeed after rereading this comment can be very confusing and as such was removed.
> > We have wrote it originally since we initially aimed to solve the problem for the more complex models and the problem was obtained from simplifying these models, however, since we do not go into depth about those models (or our preliminary analysis of them) this is indeed very confusing.
> > - (8.) We have changed "union of k-dimensional" to "set of k-dimensional" as suggested.
> > - (10.) Indeed as commented by other reviewers this point was improperly motivated. We have included explanation before which we would be glad if you could validate that solved your concerns.
> > - (11.) Implemented.
> > - (12 +13.) Implemented, would be glad if you could validate that this wording is as you intended. If this is not the case we would be glad to change the wording.
> > - (14.) Indeed we had a problem with the writing here, we changed the writing to use the previously defined consecutive partition.
> > - (15.) Implemented.
> > - (16.) Thank you for noticing, indeed we should have used this definition more.
> > We have changed it here and at other parts of the paper to simplify the notation.
> > - (17.) Indeed, we should have either stated that $\delta$ is constant or added the dependency on it; we stated explicitly that $\delta$ is constant.
> > - (18.) Implemented.
> > - (19.) Implemented. While the discussion was on the call to Algorithm 2 at algorithm 1,  it is more logical to discuss Algorithm 2 separately. This paragraph was modified to accommodate this change.
> > - (20+ 21.) Implemented. Indeed this clarify it better.
> > - (22.) Implemented. Indeed clarifies the change.
> > - (23 + 24.) We have implemented the changes, we changed $C$ to $G$ to fix the mismatch from the previous paragraph and to be consistent with the notation at Algorithm 3.
> > - (25.) This notation here meant exactly what you have thought.
> > We would appreciate if you could recommend specific rewording to clarify it if there is still a need at your eyes.
> > - (26.) Implemented, indeed we should have added more explanation. We added that it is essentially computing the optimal solution. We would be glad if you could validate that it fixed the problem.
> > - (27.) The writing here was over complicated, the consecutive referred to consecutive partition, but it is better to avoid this problem by simply stating that the input signal is projected onto a bi-criteria approximation that would avoid this unnecessary reference.
> > - (28 +29.) Implemented, indeed it was messing.
> > - (30.) Implemented, indeed this notation is simpler and was implemented at other parts where we used to old notation.
> > - (31.) Fixed, it indeed should have been a different notation. Was fixed to be "Equation 1 at Definition 2".
> > - (32.) Fixed, indeed we should gave big-oh notation as we have done at the revision.
> > - (33.) Lemma 8's algorithm essentially states to compute the optimal solution and considering that the time dependency at our experiments is more than $O(n^4)$ it would be unfeasible to compare against it; all the experiments included n values at around 10,000, and at the synthetic experiments n ranges from $2^{12}$ to $2^{16}$.
> > - (35.) Implemented. We added the quotation marks at the quotation to make them more explicit.
> > - (36.) Implemented.
> > - (37.) Indeed we have used a too hard language.
> > We meant to convey that at some cases rational functions give better fitting than polynomials, which Figure 4 came to demonstrate.
> > Obviously this only holds for some cases and not for every case.
> > We have added a comment on this above Figure 4 to avoid such confusion.

---

> > > ### Comment · Reviewer_oFkV · 2022-11-17
> > > **Late Response -- sorry about that**
> > >
> > > Hey there
> > >
> > > Sorry I missed the discussion phase, but I'm not particularly convinced by the responses from the authors.
> > >
> > > I'll elaborate why below, but I'm unfortunately not clear if the authors will even be able to read this still...
> > >
> > > **I stay convinced this paper is a clear reject. I still don't get the core idea of the algorithm.**
> > >
> > > ---
> > >
> > > ## Core detailed response
> > >
> > > At a high level, I'm still unconvinced by the writing of this paper. In my review I gave some concrete demonstrative examples of some places that the writing threw me off, but the writing issues are endemic in the paper and not just constrained to the examples I gave.
> > >
> > > The writing might be a bit better after they made some edits to the paper, but I'm not about to reread a whole paper to check all the differences, and some of my major frustrations in the writing are still there. For instance, the writing on page 5 was a major sticking point to me -- it described an algorithm in a very technical way that was hard to parse, and the authors hadn't argued correctness in any way that I understood. Now, after the edit, page 5 certainly is better than it was, just still is a lengthy and unintuitive.
> > >
> > > In their rebuttal, the authors commented that
> > > > [The] theoretical results that are of mathematical nature [so] it is hard to properly explain the results without delving into the details; as commonly said 'The God is in the details'.
> > >
> > > I don't really agree with this view of writing and describing technical arguments in papers. I mean, yeah I'm sure it's hard, but it's critical. In my experience reading this paper, I truly felt that the technical detail presented made it much harder for me to understand the big picture of what's going on.
> > >
> > > I don't want to just say "thing bad" and reject a paper. So, I'll do my best to argue for two particular follow-up recommendations:
> > > 1. I asked people I knew if they could recommend me any papers that are shining examples of how to write deeply mathematical and technical results in a nice an interpretable way -- I was broadly pointed towards Speilman's writing, for instance [this paper](https://arxiv.org/pdf/0808.0163.pdf), which apparently is a strong example of presenting the intuition before/separately from the full rigorous ideas. Hopefully this can be a nice example of showing big technical ideas while preserving the devil in the details?
> > > 1. Upon reading the newest version of page 5 of the paper, I started to understand the underlying ideas _a bit_ more, and I'm kinda convinced now that a drawing a tree and labeling it's in-nodes and out-nodes might be a nice way to intuitively understand the shape and guarantee of the data structure. This feels kinda Figure 1, but with the relationship of the nodes expressed more directly and visually?
> > >
> > > ---
> > >
> > > ## Responses to the finer points
> > >
> > > I didn't go over allll of the typos, but I tried to hit the ones where the authors wanted the most input
> > >
> > > - My example concern stated in part 1, about fitting something other than the $\ell_1$ loss doesn't seem to have been addressed really.
> > > - I still have no idea what $-6k+3$ is all about. I didn't see this clarified anywhere.
> > > - [point 9] Yeah that's fair, it's a bit of a bind. My personal approach would be that if there's more than 2 prior works that use this language, I'd follow the prior work. If there's only 1 or 2 papers though, I'd change it and explicitly note after the definition that the prior work uses the language differently. This is clearly an arbitrary rule I made up, so do as y'all want.
> > > - [point 34] My bad, I completely missed that. Mia culpa.
> > > - [point 2] I would stick with 2d because that's the setting of this paper. Too general indeed.
> > > - [point 3] The two versions seem to have the same flow/clarity issue. If assuming integer time doesn't suffice for weighted coresets to exist, then why assume integer time at all?
> > > - [point 10] I don't see what language justifies this atypical way of writing down a rational function.
> > > - [point 16] But not in your abstract?
> > > - [point 25] Calling it a projection certainly makes that line a lot clearer. I don't know why you want to use the language of bicriteria approximation for this section though, instead of just saying something like "we evaluate B along the interval P". Honestly, I think a lot of the notation and language in this paper can be simplified if you replace this ordered-pair view of your input day with just saying that your given a sequence $y_1,...,y_n$ of data points. If your x-coordinates are always integers between 1 and n, why use ordered pairs at all _anywhere_ in this paper?

---

> > > > ### Author Response · Authors · 2022-11-18
> > > > **Response**
> > > >
> > > > ## We want to thank you for checking the updated page 5
> > > > We understand your sentiment.
> > > >
> > > > We want to clarify that we have not meant to ask to re-read the entire paper but only page 5 since the claims seemed focus on it and we have not gotten the impression of endemic writing issues from the initial comment, thank you for clarifying it.
> > > >
> > > > It seems that our cited comment had gone the wrong way, since we only wanted to mentioned that the task is challenging (as agreed by the reviewer in the comment) and not that it is uncritical as we believe the reviewer interpreted it as.
> > > >
> > > > We will take the comments to heart to improve the writing in the future, but wanted to mention that at first glance the abstract in the recommended paper seems to be heavy on the technical side; nonetheless, we will check it more thoroughly for the examples of the intuition mentioned (which seems there are indeed) in an aim to improve the writing.
> > > >
> > > > *We wanted to comment that since page 5 explains all the algorithms which (combined with their proofs) are the main contribution of the paper, we understand that it can be lengthy and unintuitive; the contrary would amount to the work being short and intuitive .*
> > > >
> > > > Nonetheless, **while complexity can be excused unclear writing should be fixed** and we would try to addressed it.
> > > >
> > > > ## Responses to the finer points:
> > > > We are glad that you have gone over the finer details.
> > > >
> > > > - It seems that there was some miscommunication, since the result is indeed for $\ell_1$ and the prior intuition aims to go to this point. There is a comment after problem 1 that aims to clarify that we consider $\ell_1$ as an example while we hope for generalization of the results to different norms.
> > > > - We will address this and add proper reference to the source of this value; essentially this value comes from the proof of Lemma 2 (more specifically Corollary 4, which was used at the proof), and bound the number of "unsatisfied" cells.
> > > > - [point 9] We do not recall instances where this notation was used at the work cited in the paper, but only our notation at significantly more than a single paper (we recall at least 4 papers which were cited). We gave this paper as an example since this was a major building point for the paper where the use of this notation was the closest to our use.
> > > > - [point 34] Understandable we simply wanted to clarify.
> > > > - [point 2] Was changed.
> > > > - [point 3] The time being ${1,\cdots,n}$ allows storage of the evaluation of of a function over it by simply storing the function; i.e., $\big(f(1),\cdots,f(n)\big)$ can be stored as $f$ with additional $[1,n]$ to express the range, while if the jumps are of varying sizes the differences need to be send/stored as well.
> > > > -  [point 10] Our notation is exactly as the "proper rational function" as explained at [1]; or essentially (under the condition that the free variable is one, that can be achieved by rewriting at the case it is non-zero) the "proper rational function" at [the Wikipedia page for Rational function](https://en.wikipedia.org/wiki/Rational_function), we are aware that it is not an appropriate scientific source but since the talk seemed to be about common use we believed it a good reference for a common use.
> > > > **Nonetheless, we will aim to clarify this in the paper.**
> > > > - [point 16] Indeed we have missed it and the use of $o(1)$ instead of the small values is better at the abstract, thank you for noticing. Was changed.
> > > > - [point 25] We will consider your suggestion. Regarding the use of the use of $y_1,\cdots,y_n$ we wanted to note that the $x$-values are not always 1 to n at many parts of paper, e.g., Algorithms 3 and 4 which correspond to a large portion of the novelty in our eyes. Indeed they can be rewritten to support this notation, but this rewrite does not seem to give a *free notation simplification* as might seems from the reviewer suggestion.
> > > >
> > > > [1] Chang, Feng-Cheng, and Harld Mott. "On the matrix related to the partial fraction expansion of a proper rational function." Proceedings of the IEEE 62.8 (1974): 1162-1163.‏

---

### Author Response · Authors · 2022-11-17
**Summary of reviews and discussion: concerns are addressed**

Dear Reviewers, ACs, and SACs.

We are glad that you have spent time reviewing our paper and providing feedback which helped us clarify parts that might confuse the readers.
A few days ago we have updated a revision of our manuscript following the reviews, along with detailed answers to the reviewer's questions and detailed implantation of the suggested changes.

Overall, we hope that our response and the new version were able to address the reviewer's concerns to the extent that will hopefully convince the reviewers to raise their scores.

**Since the discussion period ends soon, in case there are any lingering concerns we ask that the reviewers would express them.**

Thank you and we look forward to hearing back.

---

### Decision · Program_Chairs · 2023-01-20

**Decision:**

Reject

**Justification For Why Not Higher Score:**

Please see the above.


**Justification For Why Not Lower Score:**

N/A


**Metareview: Summary, Strengths And Weaknesses:**

The authors present a new algorithm for finding a “coreset for rational functions”, a small subset of a training set for which all rational functions of a certain degree have nearly the same loss as the original training set.  They also reported on experiments comparing the method with software provided with numpy and scipy.  Reviewers had concerns about clarity significant enough to doubt the correctness of the analysis, and these were not fully resolved.